# Towards the Causal Complete Cause of Multi-Modal Representation Learning

Jingyao Wang [1 2 *]  Siyu Zhao [2 *]  Wenwen Qiang [1 2]  Jiangmeng Li [1 2]
Changwen Zheng [1 2]  Fuchun Sun [1 3]  Hui Xiong [4]

## Abstract

Multi-Modal Learning (MML) aims to learn effective representations across modalities for accurate predictions. Existing methods typically focus on modality consistency and specificity to learn effective representations. However, from a causal perspective, they may lead to representations that contain insufficient and unnecessary information. To address this, we propose that effective MML representations should be causally sufficient and necessary. Considering practical issues like spurious correlations and modality conflicts, we relax the exogeneity and monotonicity assumptions prevalent in prior works and explore the concepts specific to MML, i.e., Causal Complete Cause ($C^3$). We begin by defining $C^3$, which quantifies the probability of representations being causally sufficient and necessary. We then discuss the causal identifiability of $C^3$ and introduce an instrumental variable to support identifying $C^3$ with non-exogeneity and non-monotonicity. Building on this, we conduct the $C^3$ measurement, i.e., $C^3$ risk. We propose a twin network to estimate it through (i) the real-world branch: utilizing the instrumental variable for sufficiency, and (ii) the hypothetical-world branch: applying gradient-based counterfactual modeling for necessity. Theoretical analyses confirm its reliability. Based on these results, we propose $C^3$ Regularization, a plug-and-play method that enforces the causal completeness of the learned representations by minimizing $C^3$ risk. Extensive experiments demonstrate its effectiveness.

[*]Equal contribution [1]Institute of Software Chinese Academy of Sciences, Beijing, China [2]University of the Chinese Academy of Sciences, Beijing, China [3]Tsinghua University, Beijing, China [4]Thrust of Artificial Intelligence, The Hong Kong University of Science and Technology (Guangzhou), China Department of Computer Science and Engineering, The Hong Kong University of Science and Technology Hong Kong SAR, China. Correspondence to: Wenwen Qiang <qiangwenwen@iscas.ac.cn>.

*Proceedings of the $42^{nd}$ International Conference on Machine Learning*, Vancouver, Canada. PMLR 267, 2025. Copyright 2025 by the author(s).

## 1. Introduction

The initial inspiration for artificial intelligence is to imitate human perceptions which are based on different modalities (Jordan & Mitchell, 2015; Mahesh, 2020), e.g., sight, sound, and touch. Each modality provides unique information with distinct statistical properties (Baltrušaitis et al., 2018; Wang et al., 2023c). A fundamental aspect of human perception is the ability to simultaneously integrate data from different modalities to understand the world (Xu et al., 2023). Multi-modal learning (MML) (Huang et al., 2021a; Ge et al., 2023) has emerged as a promising approach to emulate human sensory perception. It aims to learn good representations from multiple modalities that can achieve accurate decisions.

In this context, a key question arises: what defines a "good" MML representation? Existing methods typically define good MML representations from two perspectives: modality consistency and modality specificity (Zhang et al., 2023a; Ge et al., 2023; Radford et al., 2021; Fan et al., 2024; Dong et al., 2024). The former emphasizes extracting modality-shared semantics related to primary events within the MML task. This type of method (Liang et al., 2022a; Xia et al., 2024; Abdollahzadeh et al., 2021) aims to obtain unified representations by mapping features from different modalities into a common embedding space. In contrast, the second perspective believes modality specificity captures the distinct statistical properties of each modality, reflecting different aspects of the primary events. This type of method (Dong et al., 2024; Yang et al., 2022; Frost et al., 2015; Zhou et al., 2021) decomposes features within each modality into modality-specific and modality-shared components, learning all shared components while applying distance constraints on modality-specific features to enhance diversity.

However, from a causal perspective (Ahuja et al., 2020; Koyama & Yamaguchi, 2020), these methods may result in the learned representations being insufficient or unnecessary. Specifically, sufficiency indicates that the use of the representations will establish the label, while necessity indicates that the label becomes incorrect when the representations are absent (Pearl, 2009). **Figure 1** provides an example. If the MML model only focuses on causal sufficiency, it will lose important modality-specific semantics, affecting generalization; if the model only focuses on causal necessity, the

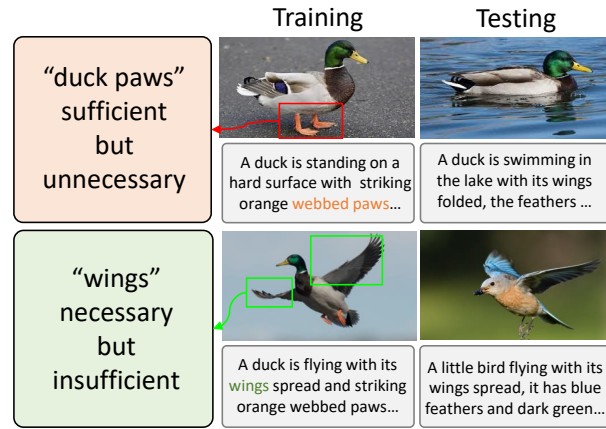

Figure 1: Example of causal sufficiency and necessity in "duck" classification task (See **Section 2** for more analyses).

decisions may be made incorrectly based on the background, affecting discriminability. Correspondingly, considering the aforementioned two types of MML methods: (i) those focused on modality consistency may cause the model to overlook important modality-specific semantics; (ii) those focused on modality specificity may incorrectly capture irrelevant information. Thus, existing MML methods (Dong et al., 2024; Lu et al., 2022) without causal constraints may fail to satisfy sufficiency and necessity, affecting model performance. The experiments in **Section 6.2** further prove this (**Figure 3** and **Table 1**): (i) the representations learned by existing methods have much lower correlation scores with sufficient and necessary causes than the proposed method with causal constraints; (ii) after constraining the learned representations, the performance of existing methods are significantly improved. The analyses in **Section 4** also emphasize the importance of causal sufficiency and necessity. Thus, inspired by (Pearl, 2009; Yang et al., 2024), we propose that a good MML representation must be both causally sufficient and necessary, i.e., causally complete.

Noticeably, in practical MML applications, constraining the causal sufficiency and necessity of learned representations remains challenging. Specifically, the related discussion in existing works (Yang et al., 2024; Pearl, 2009) is based on the exogeneity and monotonicity assumptions (**Definition D.1** and **Definition D.2** in **Appendix D.3**). Exogeneity refers to the scenario where the influence of the external intervention on the conditional distributions is negligible when the causal representation variable is exogenous relative to the label variable, while monotonicity illustrates the consistent, unidirectional effect on the label of the representation. However, in practice, non-trivial corner cases often arise: the inseparability of MML semantics leads to spurious correlations (**Figure 2**), and cross-modal conflicts combined with high-dimensional nonlinear interactions (Huang et al.,

2021b; Wang et al., 2024) undermine monotonicity. These challenges render the conditions of traditional causal sufficiency and necessity discussion inapplicable.

To address this, in this paper, we relax the above assumptions to explore the causal sufficiency and necessity in MML, ensuring the quality of the learned representation. Firstly, we propose the definition of causal sufficiency and necessity, i.e., causal complete causes ($C^3$), for MML based on (Pearl, 2009). It reflects the probability that the learned representation is causally complete by estimating the probability of label change after an intervention on the representation, given two conditions of observations. One condition is for sufficiency, and another is for necessity. Then, we analyze the causal identifiability of $C^3$, which allows us to quantify $C^3$ using the observable data in practice. Unlike previous studies, we propose an instrumental variable to ensure estimating the segmented effect of $C^3$ without spurious correlations, thereby relaxing the assumptions of exogeneity and monotonicity. Based on this, we propose the measurement of $C^3$ using the twin network, i.e., $C^3$ risk, where a low $C^3$ risk means that the learned representations are causally complete with high confidence. The challenge of $C^3$ measurement lies in eliminating the spurious correlations in the sufficiency evaluation and modeling the counterfactual data required for the necessity evaluation. The proposed twin network addresses this through (i) the real-world branch: using the proposed instrumental variable to eliminate spurious correlations in the representation, and (ii) the hypothetical-world branch: using provable gradient-based adjustments to model the counterfactuals. Through theoretical analyses, we prove the reliability of the twin network and provide the performance guarantee of the $C^3$ risk. Based on these theoretical results, we propose Causal Complete Cause Regularization ($C^3$R), a plug-and-play method to learn causal complete representations by constraining their $C^3$ risks.

**The main contributions are as follows:** (i) We propose the definition, identifiability, and measurement of the causal complete cause ($C^3$) concept for MML without the assumptions of exogeneity and monotonicity. These results provide an effective way, i.e., $C^3$ risk, to estimate the sufficiency and necessity of the learned representation for robust and accurate MML (**Section 3**). (ii) We theoretically demonstrate the effectiveness and reliability of the proposed $C^3$ risk. Inspired by this, we propose $C^3$R, which can be applied to any MML model to learn causal complete representations with low $C^3$ risk (**Sections 4 and 5**). (iii) Finally, we conduct extensive experiments on various datasets that prove the effectiveness and robustness of $C^3$R (**Section 6**).

## 2. Problem Analysis

**Problem Settings** Let $\mathcal{X}$, $\mathcal{Y}$, and $\mathcal{Z}$ represent the input space, label space, and latent space, respectively. Consider

a MML task involving data $(x, y) \in \mathcal{X} \times \mathcal{Y}$, where $x \in \mathcal{X}$ denotes the sample and $y \in \mathcal{Y}$ denotes the label. All the data are sampled from the same joint distribution $P_{XY}$ over $\mathcal{X} \times \mathcal{Y}$. The training dataset is defined as $\mathcal{D}_{tr} = \{(x_i, y_i)\}_{i=1}^{N}$, and the testing dataset is defined as $\mathcal{D}_{te}$. Here, each sample $x_i = \{x_i^{(1)}, \ldots, x_i^{(K)}\}$ contains $K$ modalities. The objective of MML is to obtain a robust model $f_\theta = \mathcal{W} \circ g$ that performs well on the unseen test dataset $\mathcal{D}_{te}$. Here, the $g : \mathcal{X} \to \mathcal{Z}$ is the feature extractor and $\mathcal{W} : \mathcal{Z} \to \mathcal{Y}$ is the classifier. The specific implementation is illustrated in **Appendix G**, and the table of notations is shown in **Table 4**.

From the perspective of data generation (Suter et al., 2019; Hu et al., 2022b), each sample is considered to be generated using a set of generating factors, e.g., color, shape, etc. In the context of MML, each sample contains generating factors from all modalities. From a causal perspective (Pearl, 2009), these generating factors can be divided into task-related factors, i.e., causal factors $F_c$, and task-independent factors, i.e., non-causal factors $F_s$. Among them, the causal factors are related to the task label, supporting accurate predictions (Deshpande et al., 2022). Thus, the goal of $f_\theta$ can be re-established as learning representations that contain all causal generating factors $F_c$ from the data without $F_s$.

**Example of Causal Sufficiency and Necessity** As shown in **Figure 1**, assume the goal of this MML task is to classify the "duck", the analysis is conducted under specific task and data conditions. The training data of this task contains three modalities, i.e., image, text, and audio, and all the modalities have the feature "duck paws". Then, the models that learn representations based on consistency may capture the features of "duck paws", and can establish the label "duck" based on "duck paws". However, the model may make errors in another "duck" scenario where the MML samples do not contain "duck paws", e.g., "duck swims on the lake" (upper right in **Figure 1**). This suggests that the learned representation contains *sufficient but unnecessary* causes: the label "duck" can be predicted using the current representation (Line 1 Left), but may not work in another scenario (Line 1 Right). Similarly, the models may also learn *necessary but insufficient* representations, e.g., the representations with feature "wings". Take the Line 2 of **Figure 1** as an example, "duck" must have "wings", but the samples with "wings" may also correspond to another label, e.g., "bird" (lower of **Figure 1**), leading to incorrect predictions. Under this context, a sufficient and necessary feature can be "flat duck bill" as its presence indicates "duck" and every "duck" sample includes it. Briefly, for *sufficient and necessary* causes, they ensure that the learned representations not only reflect the features targeted the "duck" label, but also the features that the "duck" label must have. Thus, a good representation must have both causal sufficiency and necessity, i.e., causal complete causes, for accurate and robust

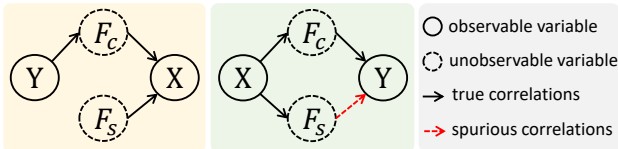

Figure 2: Structural Causal Model (SCM) for MML. **Left:** causal generating mechanism, **Right:** the learning process.

prediction. This is also the goal we explore in this study. More analyses are provided in **Appendix D.2**.

**Causal Analysis of MML** We construct a Structural Causal Model (SCM) for MML based on the causal generating mechanism (Suter et al., 2019; Deshpande et al., 2022) in **Figure 2 left**. In this SCM, Y and X denote the label variable and corresponding generated data variable in the MML task. $F_c$ and $F_s$ represent the distinct sets of generating factors that are causally and non-causally related to Y. Each generating factor corresponds to a semantic of the MML task, e.g., color, shape, background, modality indicator, etc. Since both $F_c$ and $F_s$ represent high-level knowledge of the data but only $F_c$ is causally related to Y, we could naturally define the MML task label variable Y as the cause of the $F_c$, i.e., Y $\to F_c$. There is no connection between $F_s$ and Y. Following (Hu et al., 2022a; Deshpande et al., 2022), the sample in each modality is generated simultaneously using all the generating factors, including factors caused from the label and unobservable variables, e.g., environmental effects. The unobservable variable results in the generating factors $F_s$, which are non-causally related to Y. Then, we get both $F_c \to X$ and $F_s \to X$. Thus, we obtain **Figure 2 left**. Based on this, we further construct an SCM to discuss the learning process of MML, as shown in **Figure 2 right**. It can be viewed as the inverse process of the causal generating mechanism. Based on **Figure 2 left**, an ideal MML model should only utilize causal generating factors $F_c$ and be invariant to any intervention on non-causal generating factors $F_s$. However, in practice, $F_c$ and $F_s$ in the learned representation may be coupled (Dong et al., 2024; Li et al., 2023). This leads to the model potentially learning based on non-causal $F_s$. There exist spurious correlations between $F_s$ and Y, i.e., we get the additional $F_s \to Y$ in **Figure 2 right**.

The presence of spurious correlations leads to inaccuracies in MML. Specifically, the learned MML representations may fall into four scenarios: (i) sufficient and necessary: contains all $F_c$; (ii) necessary but insufficient: contains part of $F_c$; (iii) sufficient but unnecessary: contains all $F_c$ with $F_s$; (iv) insufficient and unnecessary: only contains $F_s$. See **Appendix D** and **G** for more analyses. To ensure accurate and robust MML, in this paper, we aim to propose a methodology that constrains the model to learn representations that contain only causally sufficient and necessary information.

## 3. Causal Complete Cause

To access the learning of causal sufficient and necessary representations, in this section, we first provide the definition of causal sufficiency and necessity in MML, i.e., Causal Complete Cause ($C^3$). Next, we discuss the identifiability of $C^3$, which ensures the quantification of $C^3$ through observable data in practice, even in corner cases (non-monotonicity and non-exogeneity). Finally, we give the measurement of $C^3$, i.e., $C^3$ risk, to measure the probability of whether the learned MML representation is causally complete.

### 3.1. Definition of $C^3$

To learn representations of causal variable $F_c$ for MML that is with both causal sufficiency and necessity, based on (Pearl, 2009; Yang et al., 2024), we introduce the concept of the probability of *Causal Complete Cause* ($C^3$) as follows:

**Definition 3.1** (Probability of Causal Complete Cause ($C^3$)). Denote that the data and corresponding label variables of the given multi-modal data distribution are X and Y, while the variable of the learned MML representation is Z. Let the specific implementations of representation variable Z as $c$ and $\bar{c}$, where $c$ denotes the implementation that results in the accurate label prediction $Y = y$, and $\bar{c} \neq c$ denotes the implementation resulting in $Y \neq y$. The probability that Z is the causal complete cause of Y can be defined as:

$$C^3(\mathrm{Z}) := \underbrace{P(\mathrm{Y}_{do(\mathrm{Z}=c)} = y \mid \mathrm{Z} = \bar{c}, \mathrm{Y} \neq y)}_{\text{Sufficiency}} P(\mathrm{Z} = \bar{c}, \mathrm{Y} \neq y)$$
$$+ \underbrace{P(\mathrm{Y}_{do(\mathrm{Z}=\bar{c})} \neq y \mid \mathrm{Z} = c, \mathrm{Y} = y)}_{\text{Necessity}} P(\mathrm{Z} = c, \mathrm{Y} = y),$$

$$(1)$$

where $P(\mathrm{Y}_{do(\mathrm{Z}=c)} = y \mid \mathrm{Z} = \bar{c}, \mathrm{Y} \neq y)$ denotes the probability of $Y = y$ when force Z to be another specific implementation $c$ via do-operator $do(\mathrm{Z} = c)$, when giving observations $\mathrm{Z} = \bar{c}$ and $\mathrm{Y} \neq y$ with probability $P(\mathrm{Z} = \bar{c}, \mathrm{Y} \neq y)$. Similarly, the second term corresponds to the case that where the observations are $\mathrm{Z} = c$ and $\mathrm{Y} = y$, the probability of Y becomes incorrect when force $\mathrm{Z} = \bar{c}$.

**Definition 3.1** indicates that when Z with a high $C^3$ score, it has a high probability of being the causal complete cause of Y. The two terms in Eq.1 correspond to sufficiency and necessity, respectively. Specifically, the sufficiency term $P(\mathrm{Y}_{do(\mathrm{Z}=c)} = y \mid \mathrm{Z} = \bar{c}, \mathrm{Y} \neq y)$ means that even when $\mathrm{Z} = \bar{c}$ and $\mathrm{Y} \neq y$, intervening to set $\mathrm{Z} = c$ results in a high probability of $\mathrm{Y} = y$, indicating that $\mathrm{Z} = c$ has a sufficient causal effect on the occurrence of $\mathrm{Y} = y$. Correspondingly, for necessity term $P(\mathrm{Y}_{do(\mathrm{Z}=\bar{c})} \neq y \mid \mathrm{Z} = c, \mathrm{Y} = y)$, even when $\mathrm{Z} = c$ and $\mathrm{Y} = y$, intervening to set $\mathrm{Z} = \bar{c}$ results in $\mathrm{Y} \neq y$, indicating that $\mathrm{Z} = c$ is necessary for the occurrence of $\mathrm{Y} = y$. Under this condition, the learned representation can be divided into sufficient but unnecessary causes, neces-

sary but insufficient causes, sufficient and necessary causes, and insufficient and unnecessary causes (**Appendix D.1** for detailed analyses). We aim to constrain the $C^3$ score of the learned representation, i.e., extract sufficient and necessary causes, to achieve robust and accurate MML.

### 3.2. Causal Identifiability of $C^3$

Since it is difficult to obtain all samples in the multi-modal data distribution, especially in real systems, e.g., the counterfactual data in the definition of $C^3$ is difficult to obtain (Kusner et al., 2017; Morgan & Winship, 2015), calculating the probability of $C^3$ is still a challenging issue. To access the calculation of $C^3$ based on observable MML data, we discuss the causal identifiability of $C^3$ in this section.

Identifiability refers to the ability to uniquely infer causal effects from observable data under given assumptions (Pearl, 2009). For MML settings, it means the causal factors within the representation $Z$ can uniquely determine the label $Y$ from the sample $X$. To ensure reliable estimation, we constrain the $C^3$ score of the learned representation $Z$ extracted from $X$, ensuring its identifiability. In previous studies (Tian & Pearl, 2002; Yang et al., 2024), identifying causal probabilities typically assumes that statistical data are derived under exogeneity and monotonicity conditions (**Definition D.1** and **Definition D.2** in **Appendix D.3**). Exogeneity refers to where the external intervention's impact on the conditional distribution $P(\mathrm{Y}|\mathrm{Z})$ is negligible when the variable Z is exogenous to Y. This ensures the learned Z to recognize $F_c$ without being affected by $F_s$. Monotonicity, on the other hand, describes the consistent, unidirectional effect of the representation Z on Y. This ensures that the impact of the learned Z on the labeling decision will not be reversed as other factors change. However, in practice, non-trivial cases often arise: the inseparability of generating factors in the latent space leads to spurious correlations, i.e., the learned representations Z contain non-causal $F_s$ (**Figure 2**); Meanwhile, modality conflicts and the high-dimensional nonlinear interactions (Wang et al., 2024; Huang et al., 2021b) undermine monotonicity. These challenges render traditional identifiability conditions inapplicable. To address this, we relax the above assumptions and explore the identifiability of $C^3$, assessing the calculation of the $C^3$ score.

Consider the effects of spurious correlations, i.e., $P(\mathrm{Y}_{do(\mathrm{Z}=c)}) \neq P(\mathrm{Y} \mid \mathrm{Z} = c)$, we first relax the exogeneity assumption (with definition and analysis in **Appendix D.3**). Based on **Definition 3.1**, we extend Theorem 9.2.15 in (Pearl, 2009) to MML for $C^3$, obtaining:

**Theorem 3.2** (Causal Identifiability under Non-Exogeneity). *Given the MML model $f_\theta$, where the label variable Y is influenced by the causal factors $F_c$ and non-causal factors $F_s$. Assume $f_\theta$ satisfy the Positive Markovian assumption (Tian & Pearl, 2002), the probabilities of $C^3$ ($P(\mathrm{Y}_{do(\mathrm{Z}=c)} =$*

$y, Y_{do(Z=\bar{c})} \neq y))$ *is identifiable and can be estimated via:*

$$C^3(Z) = P(Y_{do(Z=c)}) - P(Y_{do(Z=\bar{c})}) \qquad (2)$$

*where $Y$ is monotonic relative to $Z$ with $Y_{do(Z=c)} = \bar{y} \wedge Y_{do(Z=c)} = y$ is false or $Y_{do(Z=\bar{c})} = y \wedge Y_{do(Z=c)} = \bar{y}$ is false, and the model $f_\theta$ satisfies local invertibility.*

**Theorem 3.2** establishes that, under the monotonicity condition, $C^3$ can be estimated using observable multi-modal data, thereby quantifying $C^3$. The local invertibility (Proposition D.3) states that the model can uniquely recover the distribution of $s$ from the conditional distribution of $Y$ given its parents $\text{Pa}(Y)$. Specifically, by modeling the effects of spurious correlations ($F_s$ on Y) as non-causal pathways in the SCM, interventions with the $do$-operator, such as $P(Y_{do(Z=c)})$ and $P(Y_{do(Z=\bar{c})})$, facilitates reliable causal effect estimation. For counterfactual terms ($Z = \bar{c}$) involved in the necessity probability, the model $f_\theta$ imposes locally reversible constraints to ensure identifiability, as discussed in (Galles & Pearl, 1998). Under these conditions, the probability of causal sufficiency $C^3_{su}$ and the probability of causal necessity $C^3_{ne}$ can be estimated as follows:

$$C^3_{su}(Z) = \frac{P(Y_{do(Z=c)}) - P(Y_{do(Z=\bar{c})})}{1 - P(Z=c)} \qquad (3)$$

$$C^3_{ne}(Z) = \frac{P(Y_{do(Z=c)}) - P(Y_{do(Z=\bar{c})})}{P(Z=c)} \qquad (4)$$

Next, according to the second paragraph of **Subsection 3.2**, considering the cross-modal conflicts with high-dimensional nonlinear interactions in practice, we relax the assumptions of monotonicity. The key idea lies in introducing an instrumental variable $V$ (Caner & Hansen, 2004) to estimate the piecewise effects of conditional distributions within the $C^3$ definition. Meanwhile, $V$ satisfies $V \perp\!\!\!\perp F_s$ with a controllable impact on $F_c$, i.e., $V \rightarrow F_c \rightarrow Y$. Then, we have:

**Theorem 3.3** (Causal Identifiability under Non-Monotonicity and Non-Exogeneity). *Given $f_\theta$ that learns the causal effect $Z \rightarrow Y$. The representation variable $Z = \Xi_c F_c + \Xi_s F_s$ where $\Xi_{(\cdot)}$ is the weight matrix. Introducing an instrumental variable $V$ satisfying $P(Y \mid Z, V) \approx P(Y \mid Z)$, we have:*

$$\begin{aligned} C^3(Z) = \int_v & \left[ P(Y = y \mid Z = c, V = v) \right. \\ & \left. - P(Y = y \mid Z = \bar{c}, V = v) \right] P(V = v)\, dv. \end{aligned} \qquad (5)$$

**Theorem 3.3** posits that the estimation of $C^3$ is achievable without the dependence on exogeneity and monotonicity, enabling the quantification of $C^3$ in the absence of counterfactual data. The detailed discussion is provided in **Appendix D.3**, and the proofs of the above theorems are provided in **Appendix A.1 and A.2**, respectively. According to this theorem, the practical estimation of $C^3$ hinges on how the instrumental variable $V$ is modeled and how the sufficiency and necessity terms of $C^3$ are constrained under $V$, also the main focus in the next subsection, i.e., measurement of $C^3$.

### 3.3. Measurement of $C^3$

Based on **Definition 3.1** and **Theorem 3.3**, we provide the measurement of $C^3$ in this section, i.e., $C^3$ risk, to estimate the $C^3$ score of the representation distribution $P(Z|X = x)$ inferred from X on the multi-modal data distribution $P_{XY}$. When the learned representation obtains less necessary and sufficient information, the $C^3$ risk will be higher. Specifically, we first model the instrumental variable $V$ based on **Theorem 3.3**. It aims to make the MML model learn representations Z that only contain the causal factors $F_c$ for decision-making while ensuring the identifiability of the $C^3$ score assessment (**Theorem 3.4**). Next, based on **Definition 3.1**, we provide the $C^3$ measurement of the learned Z, i.e., $C^3$ risk (**Theorem 3.5**). We employ a twin network (**Figure 5**) with a real-world branch (extract causal representation using the proposed instrumental variable) and a hypothetical-world branch (modeling counterfactual data with gradient-based adjustments), thereby modeling the sufficiency and necessity terms. **Appendix D.4** establishes the reliability of the proposed twin network.

Firstly, as shown in **Section 2**, we establish the model $f_\theta$ using the feature extractor $g$ and the linear classifier $\mathcal{W} : \mathbb{R}^d \rightarrow \mathcal{Y}$ on the overall representation $Z = g(X)$ ($Z = \Xi_c F_c + \Xi_s F_s$) to obtain the label $Y = \text{sign}(\mathcal{W}^\top g(X))$. To infer $F_c$ from observable data $x \sim \mathcal{X}$, we use the instrumental variable $V$. Here, we focus on how $V$ is modeled to constrain Z only contains $F_c$ for decision-making while satisfying the conditions in **Theorem 3.3**. We propose:

**Theorem 3.4** (Instrumental Variable $V$ in MML). *For multimodal input $X \in \mathbb{R}^{K \times d}$, representing $K$ modalities each with a feature dimension of $d$, instrumental variable $V$ is introduced to extract causal generating factors $F_c$ while reducing the influence of non-causal factors $F_s$. $V$ is generated from $X$ using self-attention mechanism with:*

$$\begin{aligned} V = \sum_{j=1}^{K} & \frac{\exp(s_{ij})}{\sum_{k=1}^{K} \exp(s_{ik})} X_j W_V, \\ & \text{where} \quad s_{ij} = \frac{q_i^\top k_j}{\sqrt{d}} - \alpha \|q_i - k_j\|^2. \end{aligned} \qquad (6)$$

*where $q_i = X_i W_Q$ and $k_j = X_j W_K$ are the query and key of $i$-th and $j$-th modalities' features, $W_Q, W_K, W_V \in \mathbb{R}^{d \times d}$ are linear projection matrices, mapping query, key and value respectively. Then, $V$ satisfies (i) $P(Z \mid V) \neq P(Z)$; (ii) $P(F_s \mid V) = P(F_s)$; and (iii) $P(Y \mid Z, V) = P(Y \mid F_c)$.*

In **Theorem 3.4**, we propose using the self-attention mechanism to model the instrumental variable $V$ for $C^3$. Specifically, the alignment score $s_{ij}$ is to capture inter-modal interactions for labels. Then, by strengthening the correlation between $V$ and Z, the learned representation Z is guided to approach $F_c$ while weakening the correlation with $F_s$. Furthermore, consider inter-modal misalignment, $\|q_i - k_j\|^2$

with weight $\alpha$ is introduced as a distance-based adjustment to diminish the impact of modalities with large feature distances on attention weights. It satisfies three conditions: (i) Relevance: $V$ manipulate $F_c$; (ii) Independence: $V$ is conditionally independent from $F_s$; and (iii) Exclusion: the influence of $V$ on Y is completely indirectly realized through $F_c$. The detailed proofs are provided in **Appendix A.3**.

Next, based on these results, we provide the measurement of $C^3$, i.e., $C^3$ risk. Specifically, we first utilize $V$ to constrain $f_\theta$, effectively distinguishing $F_c$ from $F_s$ and ensuring the identifiability of $C^3$ based on **Theorem 3.4**. Obtaining $Z_c$ that only contains $F_c$, we then define sufficiency risk (when $Z = c$, the label is not $y$) and necessity risk (when $Z = \bar{c}$, the label is $y$) to construct the $C^3$ risk based on **Definition 3.1** and Eq.5. The key challenge lies in the counterfactual data, i.e., $\overline{Z}_c : Z = \bar{c}$, is difficult to obtain (Kusner et al., 2017; Morgan & Winship, 2015). To address this, we propose a twin network comprising: (i) the real-world branch to obtain $\widehat{Z}_c$ with $V$, and (ii) the hypothetical-world branch to obtain $\overline{Z}_c$ under a provable gradient-based adjustment, ensuring $\overline{Z}_c$ stay close to the original observable distribution (**Appendix D.4**). These branches share network structures and parameters, maintaining a mirrored correspondence that enforces causal consistency (Pearl, 2009). Thus, we get:

**Theorem 3.5** ($C^3$ Risk). *Given $N$ observable samples $\{(x_i, y_i)\}_{i=1}^N$ drawn from the multi-modal distribution $P_{XY}$, and the model $f_\theta$ with feature extractor $g(\cdot)$ and linear classifier $\mathcal{W}$, we define the following twin-network:*

$$\widehat{Z}_{c,i} = g^*(x_i, v_i), \quad \overline{Z}_{c,i} = \widehat{Z}_{c,i} + \Delta_i,$$
$$\text{where} \quad \Delta_i = -\nabla_{\widehat{Z}_{c,i}} \ell\Big(\sigma(\mathcal{W}^\top \widehat{Z}_{c,i}), y_i\Big), \quad (7)$$
$$\text{s.t.} \quad \mu_{KL}(\overline{Z}_{c,i}, \widehat{Z}_{c,i}) \leq \varepsilon, \quad \forall i,$$

*where $g^*(x_i, v_i)$ is the calibrated $g$ with $v_i$ using Kullback–Leibler divergence $\mu_{KL}(\cdot)$, i.e., with loss $\mathcal{L}_v = \mathbb{E}_{(x,y) \sim P_{XY}} \mathbb{E}_{v_i \in V} \mu_{KL}(g(x), v_i)$, to estimate the causal representation $\widehat{Z}_{c,i}$ in the real-world branch, $\Delta_i$ arises from the gradient-based intervention with $v_i \sim V$ for the hypothetical world. If $f_\theta$ converges to an optimal solution under a Monte Carlo approximation of $V$, then by using:*

$$\widehat{R}^{C^3} = \frac{1}{N} \sum_{i=1}^N \Big[ \rho[\sigma(\mathcal{W}^\top \widehat{Z}_{c,i}) \neq y_i] + \rho[\sigma(\mathcal{W}^\top \overline{Z}_{c,i}) = y_i] \Big], \quad (8)$$

*the discrepancy between $\widehat{R}_N^{C^3}$ and the ideal causal complete quantity approaches zero with high probability. Here, $\sigma$ denotes the signum function, $\rho(\cdot)$ denotes an indicator function which is equal to 1 if the condition in $\rho(\cdot)$ is true.*

**Theorem 3.5** presents a method to measure the causal completeness of learned representations based on observable data. It leverages instrumental variables $V$ and a twin network to address the issues of spurious correlations in MML and the unavailability of counterfactual data in practice. The detailed proofs and analysis are provided in **Appendix A.4**.

# 4. Performance Guarantee with $C^3$ Risk

In this section, we conduct theoretical analyses to establish the connection between the proposed $C^3$ risk and MML generalization performance, proving its effectiveness.

Before discussing generalization, we first illustrate the empirical and expectation risks of the MML model $f_\theta$ on training and test data, i.e., $\widehat{R}^{C^3}(f_\theta)$ (Eq.8) and $R^{C^3}(f_\theta) = \mathbb{E}_{(x,y) \sim P_{XY}} \rho[\sigma(\mathcal{W}^\top \widehat{F}_{c,i}) \neq y_i] + \rho[\sigma(\mathcal{W}^\top \overline{F}_{c,i}) = y_i]$. Then, we provide the following performance guarantee:

**Theorem 4.1** (Performance Guarantee via $C^3$). *Denote the hypothesis set $\mathcal{H}$ of linear classifier $\mathcal{W} \in \mathcal{H} \subseteq \{h : \mathbb{R}^d \to \mathcal{Y}\}$, let $H = \mathrm{Pdim}(\{\ell_h : h \in \mathcal{H}\})$. Then with probability at least $1 - \delta$, and training samples that sampling from empirical distribution $\mathcal{D}_{tr}(x, y)$, we have*

$$R^{C^3}(f_\theta) \leq \widehat{R}^{C^3}(f_\theta) + M\mathfrak{R}(\mathcal{H}) + \sqrt{\frac{ln(1/\delta)}{2H}}. \quad (9)$$

*Here, $\mathfrak{R}(H)$ is the Rademacher complexity of the hypothesis class $H$, and $M$ is a finite constant.*

**Theorem 4.1** derives the upper bound on the generalization error of $C^3$ risk, relating its empirical error on the training data to its performance on unseen samples. Specifically, the generalization error is bounded by the sum of the empirical error, the Rademacher complexity of the hypothesis class of the linear classifier, and a confidence term that controls the probability of exceeding the bound. This result provides a theoretical performance guarantee for optimizing MML models with $C^3$ risk. See **Appendix A.5** for detailed proofs.

# 5. Learning Causal Complete Representations

Based on the above theoretical results, we propose a plug-and-play method, Causal Complete Cause Regularization ($C^3$R), which is built upon the $C^3$ risk to extract causal complete representations from observable multi-modal data.

Specifically, we first introduce the constraints of the instrumental variable $V$ (**Theorem 3.4**) to make the MML model $f_\theta$ learn causal representations while satisfying the identifiability in practice. Then, we minimize the $C^3$ risk of the learned representations (**Theorem 3.5**) to ensure the causal completeness of the learned representation, i.e., causal sufficiency and necessity. In summary, the objective of $C^3$R is a combination of the above two-step objectives, which can be embedded in various MML models. It can be expressed as:

$$\min_{f_\theta} \widehat{R}^{C^3} + \lambda_v \mathcal{L}_v + \lambda_{fe} \mathcal{L}_{fe} \quad (10)$$

where $\lambda_v$ and $\lambda_{fe}$ denotes the importance weights, the three terms: (i) $\widehat{R}^{C^3}$ (Eq.8) constrains the model to learn a causally complete representation, i.e., one that exhibits minimal sufficiency and necessity risk; (ii) $\mathcal{L}_v$, built upon

Table 1: Performance comparison when 50% samples are corrupted with Gaussian noise, i.e., zero mean with the variance of $N$. "(N, Avg.)" and "(N, Worst.)" denotes the average and worst-case accuracy. The best results are highlighted in **bold**.

| Method | NYU Depth V2 | | | | SUN RGB-D | | | | FOOD 101 | | | | MVSA | | | |
|---|---|---|---|---|---|---|---|---|---|---|---|---|---|---|---|---|
| | (0,Avg.) | (0,Worst.) | (10,Avg.) | (10,Worst.) | (0,Avg.) | (0,Worst.) | (10,Avg.) | (10,Worst.) | (0,Avg.) | (0,Worst.) | (10,Avg.) | (10,Worst.) | (0,Avg.) | (0,Worst.) | (10,Avg.) | (10,Worst.) |
| CLIP (Sun et al., 2023) | 69.32 | 68.29 | 51.67 | 48.54 | 56.24 | 54.73 | 35.65 | 32.76 | 85.24 | 84.20 | 52.12 | 49.31 | 62.48 | 61.22 | 31.64 | 28.27 |
| ALIGN (Jia et al., 2021) | 66.43 | 64.33 | 45.24 | 42.42 | 57.32 | 56.26 | 38.43 | 35.13 | 86.14 | 85.00 | 53.21 | 50.85 | 63.25 | 62.69 | 30.55 | 26.44 |
| MaPLe (Khattak et al., 2023) | 71.26 | 69.27 | 52.98 | 48.73 | 62.44 | 61.76 | 34.51 | 30.29 | 90.40 | 86.28 | 53.16 | 40.21 | 77.43 | 75.36 | 43.72 | 38.82 |
| CoOp (Jia et al., 2022a) | 67.48 | 66.94 | 49.43 | 45.62 | 58.36 | 56.31 | 39.67 | 35.43 | 88.33 | 85.10 | 55.24 | 51.01 | 74.26 | 73.61 | 42.58 | 37.29 |
| VPT (Jia et al., 2022a) | 62.16 | 61.21 | 41.05 | 37.81 | 54.72 | 53.92 | 33.48 | 29.21 | 83.89 | 82.00 | 51.44 | 49.01 | 65.87 | 64.98 | 32.79 | 29.21 |
| Late fusion (Wang et al., 2016) | 69.14 | 68.35 | 51.99 | 44.95 | 62.09 | 60.55 | 47.33 | 44.60 | 90.69 | 90.58 | 58.00 | 55.77 | 76.88 | 74.76 | 55.16 | 47.78 |
| ConcatMML (Zhang et al., 2021) | 70.30 | 69.42 | 53.20 | 47.71 | 61.90 | 61.19 | 45.64 | 42.95 | 89.43 | 88.79 | 56.02 | 54.33 | 75.42 | 75.33 | 53.42 | 50.47 |
| AlignMML (Wang et al., 2016) | 70.31 | 68.50 | 51.74 | 44.19 | 61.12 | 60.12 | 44.19 | 38.12 | 88.26 | 88.11 | 55.47 | 52.76 | 74.91 | 72.97 | 52.71 | 47.03 |
| ConcatBow (Zhang et al., 2023c) | 49.64 | 48.66 | 31.43 | 29.87 | 41.25 | 40.54 | 26.76 | 24.27 | 70.77 | 70.68 | 35.68 | 34.92 | 64.09 | 62.04 | 45.40 | 40.95 |
| ConcatBERT (Zhang et al., 2023c) | 70.56 | 69.83 | 44.52 | 43.29 | 59.76 | 58.92 | 45.85 | 41.76 | 88.20 | 87.81 | 49.86 | 47.79 | 65.59 | 64.74 | 46.12 | 41.81 |
| MMTM (Joze et al., 2020) | 71.04 | 70.18 | 52.28 | 46.18 | 61.72 | 60.94 | 46.03 | 44.28 | 89.75 | 89.43 | 57.91 | 54.98 | 74.24 | 73.55 | 54.63 | 49.72 |
| TMC (Han et al., 2020) | 71.06 | 69.57 | 53.36 | 49.23 | 60.68 | 60.31 | 45.66 | 41.60 | 89.86 | 89.80 | 61.37 | 61.10 | 74.88 | 71.10 | 60.36 | 53.37 |
| LCKD (Wang et al., 2023b) | 68.01 | 66.15 | 42.31 | 40.56 | 56.43 | 56.32 | 43.21 | 42.43 | 85.32 | 84.26 | 47.43 | 44.22 | 62.44 | 62.27 | 43.52 | 38.63 |
| UniCODE (Xia et al., 2024) | 70.12 | 68.74 | 44.78 | 42.79 | 59.21 | 58.55 | 46.32 | 42.21 | 88.39 | 87.21 | 51.28 | 47.95 | 66.97 | 65.94 | 48.34 | 42.95 |
| SimMMDG (Dong et al., 2024) | 71.34 | 70.29 | 45.67 | 44.83 | 60.54 | 60.31 | 47.86 | 45.79 | 89.57 | 88.43 | 52.55 | 50.31 | 67.08 | 66.35 | 49.52 | 44.01 |
| MMBT (Kiela et al., 2019) | 67.00 | 65.84 | 49.59 | 47.24 | 56.91 | 56.18 | 43.28 | 39.46 | 91.52 | 91.38 | 56.75 | 56.21 | 78.50 | 78.04 | 55.35 | 52.22 |
| QMF (Zhang et al., 2023c) | 70.09 | 68.81 | 55.60 | 51.07 | 62.09 | 61.30 | 48.58 | 47.50 | 92.92 | 92.72 | 62.21 | 61.76 | 78.07 | 76.30 | 61.28 | 57.61 |
| CLIP+$C^3$R | 76.54 | **75.12** | 56.73 | 52.90 | 62.31 | 58.71 | 41.59 | 37.52 | 92.93 | 91.80 | 59.77 | 57.54 | 69.61 | 68.64 | 39.58 | 35.89 |
| MaPLe+$C^3$R | 77.07 | 74.45 | 58.94 | 55.95 | 66.21 | 65.51 | 40.12 | 37.34 | 94.38 | 93.51 | 60.63 | 46.07 | 81.19 | 81.51 | 49.32 | 45.98 |
| Late fusion+$C^3$R | 73.26 | 71.62 | 57.21 | 50.98 | 64.84 | 63.25 | **53.35** | 50.43 | 94.09 | 92.24 | 65.27 | 59.02 | **83.77** | 79.79 | 62.14 | 52.50 |
| LCKD+$C^3$R | 77.14 | 75.12 | 50.11 | 47.98 | 60.97 | 60.14 | 47.23 | 46.21 | 90.89 | 90.14 | 54.48 | 51.16 | 66.78 | 65.67 | 49.28 | 42.84 |
| SimMMDG+$C^3$R | 75.32 | 74.61 | 49.99 | 47.22 | 65.50 | 64.58 | 52.69 | **51.70** | 92.24 | 91.14 | 57.32 | 53.56 | 73.62 | 71.01 | 51.65 | 51.07 |
| MMBT+$C^3$R | 73.74 | 71.82 | 54.35 | 52.57 | 61.47 | 59.99 | 48.42 | 46.07 | 94.25 | 93.90 | 60.41 | 60.11 | 82.76 | **81.64** | 62.12 | 58.93 |
| QMF+$C^3$R | **77.58** | 74.95 | **59.72** | **59.18** | **67.35** | **65.84** | 52.26 | 51.28 | **94.87** | **93.79** | **66.45** | **63.69** | 83.13 | 81.98 | **66.66** | **64.51** |

Table 2: Performance with missing modalities on BraTS. The brackets "()" indicate the effect changes after introducing $C^3$R. "•" and "○" indicate the availability and absence of the modality for testing. The best results are highlighted in **bold**.

| Modalities | | | | Enhancing Tumour | | | | | | Tumour Core | | | | | | Whole Tumour | | | | | |
|---|---|---|---|---|---|---|---|---|---|---|---|---|---|---|---|---|---|---|---|---|---|
| Fl | T1 | T1c | T2 | HMIS | HVED | RSeg | mmFm | LCKD | LCKD+$C^3$R | HMIS | HVED | RSeg | mmFm | LCKD | LCKD+$C^3$R | HMIS | HVED | RSeg | mmFm | LCKD | LCKD+$C^3$R |
| • | ○ | ○ | ○ | 11.78 | 23.80 | 25.69 | 39.33 | 45.48 | 49.81 (+4.33) | 26.06 | 57.90 | 53.57 | 61.21 | 72.01 | 76.65 (+4.64) | 52.48 | 84.39 | 85.69 | 86.10 | 89.45 | 91.62 (+2.17) |
| ○ | • | ○ | ○ | 10.16 | 8.60 | 17.29 | 32.53 | 43.22 | 49.13 (+6.01) | 37.39 | 33.90 | 47.90 | 56.55 | 66.58 | 72.18 (+5.60) | 57.62 | 49.51 | 70.11 | 67.52 | 76.48 | 82.39 (+5.91) |
| ○ | ○ | • | ○ | 62.02 | 57.64 | 67.07 | 72.60 | 75.65 | 80.50 (+4.85) | 65.29 | 59.59 | 76.83 | 75.41 | 83.02 | 88.06 (+5.04) | 61.53 | 53.62 | 73.31 | 72.22 | 77.23 | 81.93 (+4.70) |
| ○ | ○ | ○ | • | 25.63 | 22.82 | 28.97 | 43.05 | 47.19 | 54.13 (+6.94) | 57.20 | 54.67 | 57.49 | 64.20 | 70.17 | 77.32 (+7.15) | 80.96 | 79.83 | 82.24 | 81.15 | 84.37 | 90.78 (+6.41) |
| • | • | ○ | ○ | 10.71 | 27.96 | 32.13 | 42.96 | 48.30 | 54.16 (+4.86) | 41.12 | 61.14 | 60.68 | 65.91 | 74.58 | 79.83 (+5.25) | 64.62 | 85.71 | 88.24 | 87.06 | 89.97 | 93.63 (+3.66) |
| • | ○ | • | ○ | 66.10 | 68.36 | 70.30 | 75.07 | 78.75 | 82.98 (+4.23) | 71.49 | 75.07 | 80.62 | 77.88 | 85.67 | 89.74 (+4.07) | 68.99 | 85.93 | 88.51 | 87.30 | 90.47 | 93.91 (+3.44) |
| • | ○ | ○ | • | 30.22 | 32.31 | 33.84 | 47.52 | 49.01 | 56.12 (+7.11) | 57.68 | 62.70 | 61.16 | 69.75 | 75.41 | 82.57 (+7.16) | 82.95 | 87.58 | 88.28 | 87.59 | 90.39 | 95.48 (+5.09) |
| ○ | • | • | ○ | 66.22 | 61.11 | 69.06 | 74.04 | 76.09 | 81.76 (+5.67) | 72.46 | 67.55 | 78.72 | 78.59 | 82.49 | 88.32 (+5.83) | 68.47 | 64.22 | 77.18 | 74.42 | 80.10 | 87.03 (+6.96) |
| ○ | • | ○ | • | 32.39 | 24.29 | 32.01 | 44.99 | 50.09 | 56.03 (+5.94) | 60.92 | 56.26 | 62.19 | 69.42 | 72.75 | 78.78 (+6.03) | 82.41 | 81.56 | 84.78 | 82.20 | 86.05 | 92.33 (+6.28) |
| ○ | ○ | • | • | 67.83 | 67.83 | 69.71 | 74.51 | 76.01 | 83.97 (+7.96) | 76.64 | 73.92 | 80.20 | 78.61 | 84.85 | 93.57 (+8.72) | 82.48 | 81.32 | 85.19 | 82.99 | 86.49 | 94.40 (+7.91) |
| • | • | • | ○ | 68.54 | 68.60 | 70.78 | 75.47 | 77.78 | 82.94 (+5.06) | 76.01 | 77.05 | 81.06 | 79.80 | 85.24 | 90.46 (+5.22) | 72.31 | 86.72 | 88.37 | 87.33 | 90.50 | 95.51 (+5.01) |
| • | • | ○ | • | 31.07 | 32.34 | 36.41 | 47.70 | 49.96 | 56.25 (+6.29) | 60.32 | 63.14 | 64.38 | 71.52 | 76.68 | 82.69 (+6.01) | 83.43 | 88.07 | 88.81 | 87.75 | 90.46 | 96.23 (+5.77) |
| • | ○ | • | • | 68.72 | 68.93 | 70.88 | 75.67 | 77.48 | 83.90 (+6.42) | 77.53 | 76.75 | 80.72 | 79.55 | 85.56 | 92.39 (+6.83) | 83.85 | 88.09 | 89.27 | 88.14 | 90.90 | 96.78 (+5.88) |
| ○ | • | • | • | 69.92 | 67.75 | 70.10 | 74.75 | 77.60 | 82.54 (+4.94) | 78.96 | 75.28 | 80.33 | 80.39 | 84.02 | 89.43 (+5.41) | 83.94 | 82.32 | 86.01 | 82.71 | 86.73 | 91.73 (+5.00) |
| • | • | • | • | 70.24 | 69.03 | 71.13 | 77.61 | 79.33 | 86.36 (+7.03) | 79.48 | 77.71 | 80.86 | 85.78 | 85.31 | 91.43 (+6.12) | 84.74 | 88.46 | 89.45 | 89.64 | 90.84 | 95.41 (+4.57) |

**Theorem 3.4**, employs the instrumental variable $V$ to guide the model toward the causal representation, corresponding to the real-world branch; (iii) $\mathcal{L}_{fe}$ stems from the counterfactual construction condition in Eq. 7, specifically $\mathcal{L}_{fe} = \frac{1}{N} \sum_{i=1}^{N} \mu_{\text{KL}}(\overline{Z}_{c,i}, \widehat{Z}_{c,i})$, ensuring that $\widehat{Z}_{c,i}$ remains within the feasible region. Thus, by minimizing the above objective, the $C^3$ score of the learned representation will be higher while maintaining the required conditions for practical implementations. Then, it makes the learned representation causally sufficient and necessary with high confidence.

# 6. Experiments

In this section, we conduct extensive experiments on various benchmark datasets to verify the effectiveness of $C^3$R. More details and experiments are provided in **Appendix E-H**.

## 6.1. Experimental Settings

**Datasets** We select six datasets: (i) scenes recognition on NYU Depth V2 (Silberman et al., 2012) and SUN RGBD (Song et al., 2015) with RGB and depth images; (ii) image-text classification on UPMC FOOD101 (Wang et al., 2015) and MVSA (Niu et al., 2016) with image and text; (iii) segmentation considering missing modalities on BraTS (Menze et al., 2014; Bakas et al., 2018) with Flair, T1, T1c, and T2; and (iv) synthetic MMLSynData (see **Appendix D.5**).

**Implementation Details** We use a three-layer MLP with activation functions (Clevert et al., 2016) as the representation learner. The hidden vector dimensions of each layer are specified as 64, 32, and 128, while the learned representation is 64. For optimization, we employ the Adam optimizer (Kingma & Ba, 2015) with Momentum and weight decay

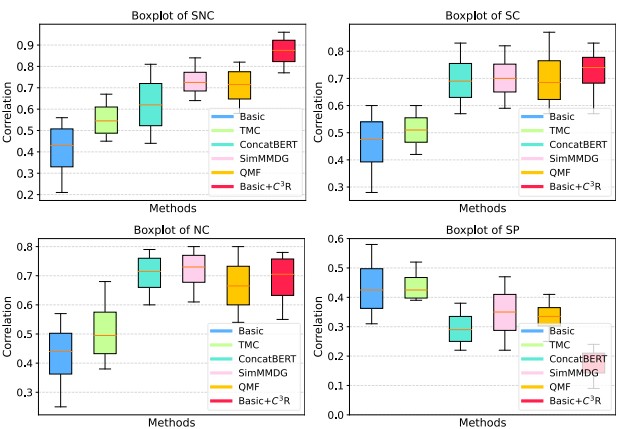

Figure 3: Evaluation for the property of learned representations (SNC, SC, NC, and SP). See **Appendix H** for details.

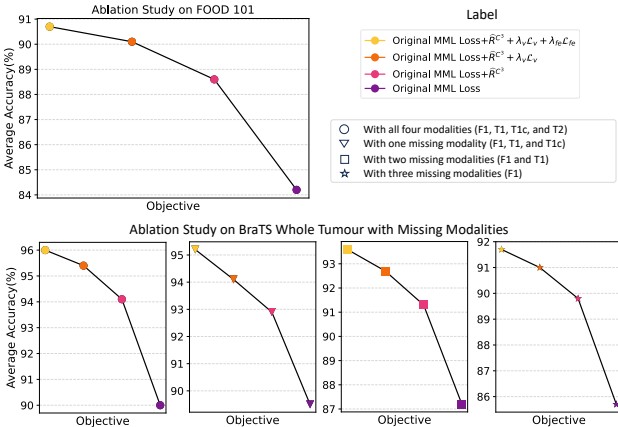

Figure 4: Ablation study of $C^3$R (performance when removing different regular terms). See **Appendix H** for details.

set at $0.8$ and $10^{-4}$. The initial learning rate is established at $0.1$, with the flexibility for linear scaling as required. Additionally, we use grid search to set the hyperparameters $\lambda_v = 0.75$ and $\lambda_{fe} = 0.4$. All experimental procedures are executed in five runs via NVIDIA RTX A6000 GPUs. Codes can be found in our Github repository[1].

### 6.2. Results

**Performance and robustness analysis** To evaluate the effectiveness of $C^3$R, we record the performance change across various MML baselines after introducing $C^3$R under Gaussian noise (for image modality) and blank noise (for text modality) following ([Han et al.](), [2020](); [Zhang et al.](), [2023d](); [Ma et al.](), [2021]()). We record both the average and worst-case accuracy. The results are shown in **Table 1**. We can observe that $C^3$R achieves stable improvements in both the average and worst-case accuracy. This proves the superior effectiveness and robustness of $C^3$R.

**When faces the problem of missing modalities** Considering the missing modality issues, we evaluate the performance of $C^3$R and several strong baselines ([Wang et al.](), [2023b](); [Bakas et al.](), [2018](); [Zhang et al.](), [2022b]()) on all 15 possible combinations of missing modalities on BraTS. From **Table 2**, we can observe that (i) $C^3$R brings significant performance improvements; (ii) $C^3$R can reduce the learning gap for the representations on different modal semantics, i.e., reducing the accuracy gap of learning on the difficult-to-identify Fl and T1 modalities and the easy T1c. This demonstrates the superiority of $C^3$R and the advantage of causally complete representation in missing modality issues.

**Learning causal complete representations** To evaluate $C^3$R's ability to extract causal complete causes, we conduct

[1]Codes can be found in [https://github.com/WangJingyao07/Multi-Modal-Base](https://github.com/WangJingyao07/Multi-Modal-Base)

experiments: (i) construct four types of MML data, i.e., sufficient and necessary (SNC), sufficient but unnecessary (SC), necessary but insufficient causes (NC), and spurious correlations (SP) following ([Yang et al.](), [2024]()) (see **Appendix H** for more details); then (ii) evaluate their correlation with the learned representation based on the distance metric ([Jones et al.](), [1995]()), i.e., the higher the score, the stronger the correlation. The results are shown in **Figure 3** (see **Appendix D** and **H** for details). The representations learned by $C^3$R have a higher correlation with SNC and lower with SP. This proves that $C^3$R can learn causal complete representations more effectively, while other methods are hard to achieve.

**Ablation Study** To evaluate the effect of each item in $C^3$R (Eq.10), we evaluate the performance change on LCKD+$C^3$R. We record the performance changes before and after the model introduced specific terms. We consider the standard and corner case (missing modality), and conduct experiments on FOOD 101 and BraTS. The results are shown in **Figure 4**. We can observe that each item plays an important role. See **Appendix H** for more experiments, including the analyses about model efficiency and parameter sensitivity. We provide a brief summary about all the experiments conducted in this paper, as shown in Table 3.

## 7. Related Work

Multi-modal learning aims to learn good representations through multiple modalities for accurate prediction. Recently, multiple methods ([Xu et al.](), [2023](); [Jia et al.](), [2022b](); [Wang et al.](), [2023c](); [Fan et al.](), [2024]()) have been proposed to solve MML tasks through modality consistency, e.g., tokenize diverse modalities into sequences and utilize Transformers for joint learning ([Bao et al.](), [2022](); [Wang et al.](), [2022]()), whereas CLIP ([Radford et al.](), [2021](); [Fan et al.](), [2024]()), ALIGN ([Jia et al.](), [2021]()), etc. employ distinct encoders for each modality and utilize contrastive loss to synchro-

nize features. These methods align features from different modalities into the same space to capture the primary events. Another type of work (Jiang et al., 2023; Dong et al., 2024; Wang et al., 2023a; Wu et al., 2020) proposed to additionally focus on modality specificity. They divide the MML features into modality-shared and modality-specific components and learn separately. However, all these methods may result in learning insufficient or unnecessary information. Meanwhile, they rely on strong assumptions like semantic alignment (Akbari et al., 2021; Wang et al., 2023c; Lu et al., 2022; Zhang et al., 2023b) and may be limited to ideal data (Ge et al., 2023; Zhang et al., 2023a), affecting model performance. For causal sufficiency and necessity, (Pearl, 2009) first proposed the related concepts, and (Yang et al., 2024) applied it to domain generalization with assumptions in an ideal environment. The differences between our methodology and previous causal-related works include problem settings, theorems, implementations, learning objectives, etc. (More comparison and analyses are provided in **Appendix D.6**). We explore the MML-specific concepts of causal completeness without strong assumptions, ensuring the effectiveness of the learned representations.

## 8. Conclusion

We explore the MML-specific concepts of causal sufficient and necessary without exogeneity and monotonicity assumptions. To measure whether the learned MML representation is causally complete, we present the definition, identifiability, and measurement of $C^3$ with theoretical support. Based on these results, we propose a plug-and-play method $C^3$R to promote MML learning causal complete representations. Extensive experiments demonstrate its effectiveness.

## Impact Statement

This paper presents work whose goal is to advance the field of Machine Learning and Multi-Modal Learning. There are many potential societal consequences of our work, none of which we feel must be specifically highlighted here.

## Acknowledgements

The authors would like to thank the anonymous reviewers for their valuable comments. This work was supported in part by the National Natural Science Foundation of China (Grant No. 62406313), in part by the National Key R&D Program of China (Grant No.2023YFF0725001), in part by the National Natural Science Foundation of China (Grant No.92370204), in part by the guangdong Basic and Applied Basic Research Foundation (Grant No.2023B1515120057) in part by Guangzhou-HKUST (GZ) Joint Funding Program (Grant No.2023A03J0008), Education Bureau of Guangzhou Municipality.

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

# Appendix

This supplementary material provides results for additional experiments and details to reproduce our results that could not be included in the paper submission due to space limitations.

- **Appendix A** provides proofs and further theoretical analysis of the theory in the text.

- **Appendix B** provides the pseudo-code of incorporating $C^3$R into the MML models.

- **Appendix C** provides the detailed notations and the corresponding definitions for this work.

- **Appendix D** provides discussions of how to better understand $C^3$ score, and the three parts of learned invariant representations, i.e., sufficient but unnecessary, necessary but insufficient, and sufficient and necessary causes.

- **Appendix E** provides the additional details of the benchmark datasets.

- **Appendix F** provides the additional details of the baselines for comparision.

- **Appendix G** provides the additional details of the implementation details.

- **Appendix H** provides the full results and additional experiments for the evaluation of $C^3$R.

Note that before we illustrate the details and analysis, we provide a brief summary about all the experiments conducted in this paper, as shown in Table 3.

Table 3: Illustration of the experiments conducted in this work. Note that all experimental results are obtained after five rounds of experiments.

| Experiments | Location | Results |
|---|---|---|
| Performance and robustness analysis | Section 6.2 and Appendix H.1 | Table 1, Table 6, and Table 7 |
| Performance comparison when faces the problem of missing modalities | Section 6.2 and Appendix H.2 | Table 2 |
| Performance of learning causal complete representations | Section 6.2, Appendix D.5, and Appendix H.3 | Figure 3 and Figure 6 |
| Ablation Study about the effect of each item in the $C^3$R objective | Section 6.2 and Appendix H.4 | Figure 4 |
| Experiment of model efficiency | Appendix H.4 | Figure 7 |
| Experiment of parameter sensitivity | Appendix H.4 | Figure 8 and Figure 9 |
| Visualization of Causal Complete Cause | Appendix H.5 | Figure 10 and Table 8 |
| Performance When Facing Noise | Appendix H.6 | Table 9 |
| Role of Distance Loss | Appendix H.7 | Figure 11 |

# A. Proofs

In this section, we provide the proofs of (i) the $C^3$ identifiability under non-exogeneity (**Theorem 3.2**), (ii) the $C^3$ identifiability under non-exogeneity and non-monotonicity (**Theorem 3.3**) (iii) the modeling of instrumental variable $Z$ in MML (Conditions in **Theorem 3.4**), (iv) $C^3$ risk with twin network (**Theorem 3.5**), where the reliability and completeness of the twin network is demonstrated in **Appendix D.4**, and (v) the performance guarantee for $C^3$ risk (**Theorem 4.1**).

## A.1. Proofs of Theorem 3.2

*Proof.* Based on Definition 3.1, $C^3(Z)$ is defined as:

$$C^3(Z) := P\left(Y_{do(Z=c)} = y \mid Z = \bar{c}, Y \neq y\right) \cdot P(Z = \bar{c}, Y \neq y)$$

$$+ P\left(Y_{do(Z=\bar{c})} \neq y \mid Z = c, Y = y\right) \cdot P(Z = c, Y = y)$$

Expand the conditional probabilities into joint probabilities over marginal probabilities:

$$P\left(Y_{do(Z=c)} = y \mid Z = \bar{c}, Y \neq y\right) = \frac{P\left(Y_{do(Z=c)} = y, Z = \bar{c}, Y \neq y\right)}{P\left(Z = \bar{c}, Y \neq y\right)}$$

$$P\left(Y_{do(Z=\bar{c})} \neq y \mid Z = c, Y = y\right) = \frac{P\left(Y_{do(Z=\bar{c})} \neq y, Z = c, Y = y\right)}{P\left(Z = c, Y = y\right)}$$

Substituting these into $C^3(Z)$:

$$C^3(Z) = P\left(Y_{do(Z=c)} = y, Z = \bar{c}, Y \neq y\right) + P\left(Y_{do(Z=\bar{c})} \neq y, Z = c, Y = y\right)$$

Considering that the monotonicity assumption states:

$$Y_{do(Z=\bar{c})} = y \Rightarrow Y_{do(Z=c)} = y$$

This means there are no individuals for whom $Y$ is $y$ under $Z = \bar{c}$ but changes to $\neq y$ when $Z = c$. Therefore:

$$P\left(Y_{do(Z=\bar{c})} = y \wedge Y_{do(Z=c)} \neq y\right) = 0$$

This eliminates the second term in $C^3(Z)$:

$$P\left(Y_{do(Z=\bar{c})} \neq y, Z = c, Y = y\right) = 0$$

Thus, the simplified $C^3(Z)$ becomes:

$$C^3(Z) = P\left(Y_{do(Z=c)} = y, Z = \bar{c}, Y \neq y\right)$$

Considering the causal model, $Y_{do(Z=c)} = y$ can be decomposed into two parts:

- Individuals for whom $Y \neq y$ when $Z = \bar{c}$, but $Y = y$ when intervened to $Z = c$.
- Individuals for whom $Y = y$ when $Z = \bar{c}$, and $Y$ remains $y$ when intervened to $Z = c$.

Thus:

$$P\left(Y_{do(Z=c)} = y\right) = C^3(Z) + P\left(Y_{do(Z=c)} = y, Y_{do(Z=\bar{c})} = y\right)$$

According to the monotonicity assumption:

$$Y_{do(Z=\bar{c})} = y \Rightarrow Y_{do(Z=c)} = y$$

Therefore:

$$P\left(Y_{do(Z=c)} = y, Y_{do(Z=\bar{c})} = y\right) = P\left(Y_{do(Z=\bar{c})} = y\right)$$

Substituting this into the previous equation:

$$P\left(Y_{do(Z=c)} = y\right) = C^3(Z) + P\left(Y_{do(Z=\bar{c})} = y\right)$$

Solving for $C^3(Z)$ from the above equation:

$$C^3(Z) = P\left(Y_{do(Z=c)} = y\right) - P\left(Y_{do(Z=\bar{c})} = y\right)$$

This is precisely the formula presented in the theorem.

## A.2. Proofs of Theorem 3.3

*Proof.* Under spurious correlations and non-monotonicity, one cannot simply take

$$P(Y \mid Z = c) - P(Y \mid Z = \bar{c})$$

as a valid measure of causal effect. The key idea lies in introducing an instrumental variable $V$ (Caner & Hansen, 2004) to estimate conditional distributions and piecewise effects. An instrumental variable $V$ helps "decouple" $F_c$ from unobserved $F_s$, allowing us to recover or approximate the true intervention distribution $P(Y_{do(Z)})$.

Specifically, if

$$P(Y \mid Z, V) = P(Y \mid Z),$$

then $V$ is conditionally independent of $Y$ given Z. By stratifying over $V$, we can estimate the conditional interventional distribution and aggregate (integrate) across values of $V$ to obtain the true causal effect.

Recall the key idea in causal inference:

$$P(Y_{do(Z=c)} = y) \;=\; \int_v P\big(Y_{do(Z=c)} = y \,\big|\, V = v\big) \, P(V = v) \, \mathrm{d}V.$$

Since $V$ is assumed to provide no direct influence on $Y$ aside from through $F_c$, we can drop the do-operator in the conditional:

$$P(Y_{do(Z=c)} = y) \approx \int_v P(Y = y \mid Z = c, V = v) \, P(V = v) \, \mathrm{d}v$$

$$= \int_v P(Y = y \mid Z = c) \, P(V = v) \, \mathrm{d}z \quad (\text{if } P(Y \mid Z, V) = P(Y \mid Z)).$$

Likewise,

$$P(Y_{do(Z=\bar{c})} = y) = \int_v P(Y = y \mid Z = \bar{c}, V = v) \, P(V = v) \, \mathrm{d}v.$$

Recall the definition of $C^3$:

$$C^3(Z) \;:=\; P(Y_{do(Z=c)} = y) \;-\; P(Y_{do(Z=\bar{c})} = y).$$

By substituting the integrals,

$$C^3(Z) = \int_v P(Y = y \mid Z = c, V = v) \, P(V = v) \, \mathrm{d}v$$

$$- \int_v P(Y = y \mid Z = \bar{c}, V = v) \, P(V = v) \, \mathrm{d}v$$

$$= \int_z \Big[ P(Y = y \mid Z = c, V = v) - P(Y = y \mid Z = \bar{c}, V = v) \Big] P(V = v) \, \mathrm{d}v.$$

Thus, we get:

$$C^3(Z) = \int_v \Big[ P(Y = y \mid Z = c, V = v) - P(Y = y \mid Z = \bar{c}, V = v) \Big] P(V = v) \, \mathrm{d}v.$$

Hence, even under non-monotonicity and non-exogeneity, if there exists an appropriate instrumental variable $V$ satisfying $P(Y \mid Z, V) = P(Y \mid Z)$, we can stratify on $V$ and integrate out $P(V = v)$ to recover an identifiable estimate for $C^3(Z)$.

## A.3. Analyses of Theorem 3.4

*Proof.* In a multimodal learning setting, we denote the causal generating factors across modalities as $F_c$ and the label-irrelevant interference factors as $F_s$, and our goal is to construct an auxiliary variable $z$ via a self-attention mechanism. After training, this $z$ should sufficiently aggregate information from $F_c$, weaken as much as possible the influence of $F_s$, and avoid providing any extraneous intervention pathway for the label $Y$. Briefly, in this subsection, we illustrate how $Z$ achieve the three conditions in Theorem 3.4, (i) $P(Z \mid V) \neq P(Z)$; (ii) $P(F_s \mid V) = P(F_s)$; and (iii) $P(Y \mid Z, V) = P(Y \mid F_c)$.

Building upon the earlier stage, we now provide more detailed formulaic explanations, especially regarding how gradients are computed.

First, we start from the classical self-attention formula. Let

$$q_i = X_i\,W_Q, \quad k_j = X_j\,W_K, \quad v_j = X_j\,W_V,$$

where $X_i \in \mathbb{R}^d$ represents the input features of the $i$-th modality, and $W_Q, W_K, W_V \in \mathbb{R}^{d \times d}$ are learnable linear projection parameters. To generate the auxiliary variable $z$ for the $i$-th modality, we define the scoring function:

$$s_{ij} \;=\; \frac{q_i^\top k_j}{\sqrt{d}} \;-\; \alpha\,\|\,q_i - k_j\,\|^2,$$

and then apply a Softmax to obtain the attention weights

$$\alpha_{ij} \;=\; \frac{\exp\!\big(s_{ij}\big)}{\sum_{m=1}^{K} \exp\!\big(s_{im}\big)}.$$

Subsequently, we construct

$$z \;=\; \sum_{j=1}^{K} \alpha_{ij}\,(v_j)$$

by taking the weighted sum. In this formulation, $F_c$ can be viewed as the "common dimension" across all modalities that are strongly correlated with the label, whereas $F_s$ corresponds to each modality's internal style or noise dimension. In order for $v$ to primarily collect $F_c$ while excluding $F_s$, we rely on the objective-driven gradient updates during training. If we let $\theta = (W_Q, W_K, W_V)$ denote all projection matrices and possibly other network parameters, then each iteration's update rule can typically be abstracted as

$$\theta \;\longleftarrow\; \theta \;-\; \eta \sum_{(i,j)} \frac{\partial \mathcal{L}}{\partial s_{ij}}\,\frac{\partial s_{ij}}{\partial \theta},$$

where $\eta$ is the learning rate, and $\mathcal{L}$ is the model's overall error or loss function used to measure how well the network predicts the label $Y$ on the current batch (or satisfies other constraints, such as preferences for causal representations). By applying the chain rule, we can further expand

$$\frac{\partial \mathcal{L}}{\partial s_{ij}} \;=\; \frac{\partial \mathcal{L}}{\partial \alpha_{ij}}\,\frac{\partial \alpha_{ij}}{\partial s_{ij}} \;+\; \sum_{p \neq j} \frac{\partial \mathcal{L}}{\partial \alpha_{ip}}\,\frac{\partial \alpha_{ip}}{\partial s_{ij}},$$

where $\frac{\partial \alpha_{ij}}{\partial s_{ij}}$ and $\frac{\partial \alpha_{ip}}{\partial s_{ij}}$ reflect the competitive relationships of attention distribution across different modalities, and $\frac{\partial \mathcal{L}}{\partial \alpha_{ij}}$ corresponds to how the specific attention weight impacts the task or other constraints, whether positively or negatively. Next, we consider $\frac{\partial s_{ij}}{\partial \theta}$. Since

$$s_{ij} \;=\; \frac{q_i^\top k_j}{\sqrt{d}} \;-\; \alpha\,\|\,q_i - k_j\,\|^2 \;=\; \frac{(X_i W_Q)^\top (X_j W_K)}{\sqrt{d}} \;-\; \alpha\,\|(X_i W_Q) - (X_j W_K)\|^2,$$

its gradient with respect to $\theta$ can be further decomposed into

$$\frac{\partial s_{ij}}{\partial W_Q}, \quad \frac{\partial s_{ij}}{\partial W_K}, \quad \frac{\partial s_{ij}}{\partial W_V}.$$

Note that $v_j = X_j W_V$ appears only in the weighted sum, not in the scoring function, so $\frac{\partial s_{ij}}{\partial W_V} = 0$. However, in generating the final output $v$, $v_j$ still influences $\frac{\partial \mathcal{L}}{\partial \alpha_{ij}}$, so $\frac{\partial \mathcal{L}}{\partial v_j}$ will likewise propagate back to $W_V$. When the network repeatedly performs this type of gradient backpropagation, if certain $(i, j)$ features are only close in terms of $F_s$ and do not help real label prediction, then $\frac{\partial \mathcal{L}}{\partial s_{ij}}$ tends to be negative or very small; conversely, if $(i, j)$ align well in terms of $F_c$ and bring better predictions, $\frac{\partial \mathcal{L}}{\partial s_{ij}}$ will be overall positive or large, encouraging the score to increase. Consequently, after many iterations of gradient descent, the scores $s_{ij}$ corresponding to $F_c$ are continuously reinforced, shrinking $\|q_i - k_j\|$ further and boosting

the dot product $\frac{q_i^\top k_j}{\sqrt{d}}$, so that $\alpha_{ij}$ is raised to a higher level. Meanwhile, pairs that initially had high scores only due to $F_s$ resemblance will gradually get suppressed by the long-term gradient updates, lowering $\alpha_{ij}$ and reducing their contribution to Z.

By a similar logic, in meeting the exclusion requirement, if some $(i, j)$ combination can potentially bypass $F_c$ and directly influence label $Y$, but fails to get consistent positive reinforcement in the loss function (or even triggers negative regularization), then the result of backpropagation will be to suppress such shortcut-related attention weights, preventing them from accumulating through repeated updates. If $F_c$ alone suffices to determine $Y$, then additional influences on the label from such detours are not favored by the loss, so the parameters shift against this overstepping, ultimately making the learned attention distribution pay little heed to noncausal pathways.

In this manner, once training converges, the instrumental variable $V$ comes to reflect common features aligned with $F_c$, which explains why observing $V$ updates our inference about $F_c$ and ensures $P(Z \mid V) \neq P(Z)$. Meanwhile, couplings with $F_s$, which represent mere style or noise, are gradually eliminated from the scoring, leaving $z$ nearly independent of $F_s$. Lastly, given Z, $V$ does not convey additional pathways to $Y$, since detouring around $F_c$ has no sustainable benefit in the gradient. From the perspective of the gradient update formula, each positive or negative adjustment applies directly to $s_{ij}$ and $\alpha_{ij}$, ultimately shaping $z$ into a representation that strongly consolidates the causal features while masking non-causal interferences and refraining from interfering directly with the label.

Hence, by the end of training, we naturally obtain an auxiliary variable $V$ that satisfies the following properties: it is positively correlated with $F_c$, largely insensitive to $F_s$, and does not offer extra intervention for label $Y$. The formulaic derivation and gradient-based explanation clearly illustrate why simply introducing the above self-attention scoring and relying on gradient-driven optimization can distill a robust representation of Z with minimal interference in a multimodal setting, i.e., achieving (i) $P(Z \mid V) \neq P(V)$; (ii) $P(F_s \mid V) = P(F_s)$; and (iii) $P(Y \mid Z, V) = P(Y \mid F_c)$.

### A.4. Proofs of Theorem 3.5

*Proof.* The main proofs can be divided into three steps: (i) from Markov property to the validity of local intervention; (ii) reliability of the twin network: why gradient-based intervention + KL constraint can approximate a causal intervention; and (iii) justification of $c^3$ risk and corresponding error analysis.

Before establishing the main proof, we begin by illustrating the background of the problem and the statement of the Theorem. Specifically, we operate under a multimodal distribution $P_{XY}$, with an observable sample set $\{(x_i, y_i)\}_{i=1}^N$, $(x_i, y_i) \sim P_{XY}$, where $x_i$ here denotes a concatenation/joint form of features from multiple modalities, and $y_i$ is the label. We then consider a learning model $f_\theta = \mathcal{W} \circ g(\cdot)$, where $g(\cdot)$ is a feature extraction network and $\mathcal{W}$ a linear classifier. In order to approximate the causal representation $Z_c$ during training, we introduce a calibration or regularization loss in $g$. In particular, for each sample $x_i$, we draw a random variable $v_i \sim V$, which can be regarded as an auxiliary or instrumental distribution, and define $\widehat{Z}_{c,i} = g^*(x_i, v_i)$, where $g^*$ denotes the modified feature extractor incorporating the calibration/regularization that approximates the causal representation.

Next, for gradient-based intervention with KL constraint, in this causal representation space $\widehat{Z}_{c,i}$, we introduce a small perturbation $\Delta_i$, setting

$$\Delta_i = -\nabla_{\widehat{Z}_{c,i}} \ell\big(\sigma(\mathcal{W}^\top \widehat{Z}_{c,i}), y_i\big),$$

thereby defining the second branch of the twin network:

$$\overline{Z}_{c,i} = \widehat{Z}_{c,i} + \Delta_i.$$

To ensure that this local intervention does not diverge too far from $\widehat{Z}_{c,i}$, we impose the Kullback–Leibler (KL) divergence constraint:

$$\mu_{\mathrm{KL}}\big(\overline{Z}_{c,i}, \widehat{Z}_{c,i}\big) \leq \varepsilon,$$

which means we only allow gradient interventions within a small neighborhood, ensuring that $\overline{Z}_{c,i}$ remains probabilistically similar to $\widehat{Z}_{c,i}$. Then, we define the empirical quantity

$$\widehat{R}^{C^3} = \frac{1}{N} \sum_{i=1}^N \Big[\rho\big(\sigma(\mathcal{W}^\top \widehat{Z}_{c,i}) \neq y_i\big) + \rho\big(\sigma(\mathcal{W}^\top \overline{Z}_{c,i}) = y_i\big)\Big],$$

where $\rho(\cdot)$ is an indicator function, taking value 1 when the condition is true and 0 otherwise, and $\sigma(\cdot)$ is the signum function. In concise terms, $\widehat{R}^{C^3}$ simultaneously counts the classification errors in the real branch $\widehat{Z}_{c,i}$ and the instances where the intervention branch $\overline{Z}_{c,i}$ incorrectly retains the original label $y_i$ instead of flipping it.

Theorem 3.5 states: If $f_\theta$ converges to an optimal solution under the Monte Carlo approximation $\{v_i \sim V\}$, and for each sample $i$ the perturbation $\Delta_i$ satisfies the above KL constraint, then with high probability in the large-sample limit, $\widehat{R}^{C^3}$ converges to the true causal complete quantity.

**Part I** Formalizing the Markov property (Frydenberg, 1990; Blumenthal, 1957), we assume in the multimodal context that the causal representation $Z_c$ satisfies a Markov property: given $Z_c$, the label $Y$ is independent of other interference variables $F_s$, symbolically $(Y \perp F_s) \mid Z_c$, or in causal graph terms, back-door paths (Pearl, 2009) are blocked by $Z_c$. When we perform an intervention on $Z_c$ (be it global or local), it does not compromise the remaining independence conditions involving $F_s$. If, for sample $x_i$, we feed it into $g^*$ to obtain $\widehat{Z}_{c,i}$ as the real-world branch, and then apply a small gradient perturbation to get $\overline{Z}_{c,i}$ as the hypothetical-world branch, the Markov assumption ensures that modifying $Z_c$ alone will not inadvertently bring $F_s$ into play or disrupt other independent factors. Consequently, the concept of "twin" is feasible at the formulaic level. The difference between $\widehat{Z}_{c,i}$ and $\overline{Z}_{c,i}$ precisely gauges how the label $Y$, if it truly depends only on $Z_c$, should respond to a local change.

**Part II** Let the loss function

$$\ell\Big(\sigma(\mathcal{W}^\top \widehat{Z}_{c,i}), \, y_i\Big)$$

be denoted by

$$L_i(\widehat{Z}_{c,i}) = \ell\Big(\sigma(\mathcal{W}^\top \widehat{Z}_{c,i}), \, y_i\Big).$$

Then

$$\Delta_i = -\nabla_{\widehat{Z}_{c,i}} \, L_i(\widehat{Z}_{c,i})$$

is a small step in the negative gradient direction on the $\widehat{Z}_{c,i}$ space. We define

$$\overline{Z}_{c,i} = \widehat{Z}_{c,i} + \Delta_i.$$

To keep this perturbation local, we require

$$\mu_{\mathrm{KL}}\big(\overline{Z}_{c,i}, \, \widehat{Z}_{c,i}\big) \, \leq \, \varepsilon,$$

thereby bounding the scale of $\|\Delta_i\|$ so that $\overline{Z}_{c,i}$ remains in the KL-neighborhood of $\widehat{Z}_{c,i}$.

Viewing $Z_c$ as directly influencing $Y$, changing $Z_c$ to $\overline{Z}_{c,i}$ in a causal model is akin to imposing $do(Z_c = \overline{Z}_{c,i})$. In practice, performing a complete change in $Z_c$ for a multimodal environment can be too difficult, or insufficient data may exist to estimate it (Liang et al., 2022b). By enacting a local gradient $\Delta_i$ while ensuring $\overline{Z}_{c,i}$ and $\widehat{Z}_{c,i}$ remain within finite KL distance, we can approximate a mild causal intervention and test whether the model is sensitive to label changes that should happen. If the model truly captures the correct causal relationship, then it should detect and reflect these label flips. The reliability and completeness of the twin network is proved and discussed in Appendix D.4.

**Part III** Recall the proposed $C^3$ risk, we have:

$$\widehat{R}^{C^3} = \frac{1}{N} \sum_{i=1}^{N} \Big[ \rho\big(\sigma(\mathcal{W}^\top \widehat{Z}_{c,i}) \neq y_i\big) + \rho\big(\sigma(\mathcal{W}^\top \overline{Z}_{c,i}) = y_i\big) \Big].$$

For brevity, set

$$h_i = \rho\big(\sigma(\mathcal{W}^\top \widehat{Z}_{c,i}) \neq y_i\big) + \rho\big(\sigma(\mathcal{W}^\top \overline{Z}_{c,i}) = y_i\big),$$

Then, we get:

$$\widehat{R}^{C^3} = \frac{1}{N} \sum_{i=1}^{N} h_i.$$

In theory, if we had a fully correct causal model, we could define an ideal quantity

$$R^{C^3} = \mathbb{E}_{(x,y)\sim P_{XY}} \left[ \rho(\cdots) \right],$$

where the $\rho(\cdots)$ events reflect correct label under the real branch? and incorrect label under the intervention branch? etc. Because the Markov property allows local interventions on $F_c$ alone and the Monte Carlo set $\{v_i\}$ helps calibrate $g^*$, we expect that, as $N \to \infty$,

$$\left| \widehat{R}_N^{C^3} - R^{C^3} \right| \to 0 \quad \text{(with high probability)}.$$

Based on this, we can decomposing the error, i.e., define:

$$\delta_i = h_i - h_i^{\text{ideal}},$$

where $h_i^{\text{ideal}}$ represents the indicator's value under the truly causal complete scenario for sample $i$. Then

$$\widehat{R}^{C^3} - R^{C^3} = \frac{1}{N} \sum_{i=1}^{N} \delta_i.$$

To show that this difference converges to zero as $N \to \infty$, one needs to argue that under the KL constraint and the Markov assumption, $\delta_i$ can be made very small on average, and its variance is controlled. Applying Hoeffding's inequality or Markov's inequality on the i.i.d. samples $(x_i, y_i, v_i)$ then ensures convergence with high probability.

When $\|\Delta_i\|$ is small and $\mu_{\text{KL}}(\overline{Z}_{c,i}, \widehat{Z}_{c,i}) \le \varepsilon$, the local intervention around $\widehat{Z}_{c,i}$ will not drastically alter the sample's label distribution. In particular, assuming $f_\theta$ is well-optimized, two things can happen:

- If $\sigma(\mathcal{W}^\top \widehat{Z}_{c,i})$ was correct, then a small gradient shift $\sigma(\mathcal{W}^\top \overline{Z}_{c,i})$ likely stops predicting the same label (assuming it is designed to flip it).

- Conversely, if $\widehat{Z}_{c,i}$ was incorrect, then the adjusted $\overline{Z}_{c,i}$ can likely correct the label.

Since this agrees with how a genuinely causal intervention would flip or not flip the label, $\delta_i$ can be suppressed to a very small expected value (Arslan et al., 2015; Piotroski & So, 2012). With i.i.d. sampling and concentration inequalities, we get:

$$\frac{1}{N} \sum_{i=1}^{N} \delta_i \to 0 \quad \text{(in probability)},$$

implying $\widehat{R}_N^{C^3} \to R^{C^3}$. Hence, it shows that in a multimodal setting under the Markov assumption, performing a controlled gradient-based intervention on $\widehat{Z}_{c,i}$ to obtain $\overline{Z}_{c,i}$ based on Hoeffding inequality, and defining $\widehat{R}^{C^3}$ from these two branches (real vs. intervention), yields an estimate converging to the genuine causal complete quantity in the large-sample limit. Thus, the proof of Theorem 3.5 is established:

$$\widehat{R}_N^{C^3} \longrightarrow R^{C^3} \quad \text{as } N \to \infty, \text{with high probability}.$$

### A.5. Proofs of Theorem 4.1

*Proof.* We will need the following definition for this proof:

$$\rho_{\theta,h}(x,y) = \min_{y'} \left( h(x,y) - h(x,y') + \theta 1_{y'=y} \right), \tag{11}$$

where $\theta > 0$ is an arbitrary constant. Observe that $\mathbb{E}\left[ 1_{\rho_h(x,y)\le 0} \right] \le \mathbb{E}\left[ 1_{\rho_{\theta,h}(x,y)\le 0} \right]$ since the inequality $\rho_{\theta,h}(x,y) \le \rho_h(x,y)$ holds for all $(x,y) \in X \times y$:

$$\rho_{\theta,h}(x,y) = \min_{y'} \left( h(x,y) - h(x,y') + \theta 1_{y'=y} \right) \tag{12}$$

$$\le \min_{y'\neq y} \left( h(x,y) - h(x,y') + \theta 1_{y'=y} \right) \tag{13}$$

$$= \min_{y'\neq y} \left( h(x,y) - h(x,y') \right) = \rho_h(x,y) \tag{14}$$

where the inequality follows from taking the minimum over a smaller set. Now, let $\widetilde{\mathcal{H}} = \{(x, y) \mapsto \rho_{\theta,h}(x, y) : h \in \mathcal{H}\}$ and $\widetilde{\mathcal{H}} = \left\{ \Phi_\rho \circ \widetilde{h} : \widetilde{h} \in \widetilde{\mathcal{H}} \right\}$. With probability at least $1 - \delta$, for all $h \in \mathcal{H}$,

$$\mathbb{E}\left[\Phi_\rho\left(\rho_{\theta,h}(x, y)\right)\right] \leq \frac{1}{m} \sum_{i=1}^{m} \Phi_\rho\left(\rho_{\theta,h}\left(x_i, y_i\right)\right) + 2\mathfrak{R}_m(\tilde{\mathcal{H}}) + \sqrt{\frac{\log \frac{1}{\delta}}{2m}} \tag{15}$$

Since $1_{u \leq 0} \leq \Phi_\rho(u)$ for all $u \in \mathbb{R}$, the generalization error is a lower bound on the left-hand side, and we can write:

$$R^{C^3}(f_\theta) \leq \frac{1}{m} \sum_{i=1}^{m} \Phi_\rho\left(\rho_{\theta,h}\left(x_i, y_i\right)\right) + 2\mathfrak{R}_m(\widetilde{\mathcal{H}}) + \sqrt{\frac{\log \frac{1}{\delta}}{2m}}. \tag{16}$$

Fixing $\theta = 2\rho$, we observe that $\Phi_\rho\left(\rho_{\theta,h}\left(x_i, y_i\right)\right) = \Phi_\rho\left(\rho_h\left(x_i, y_i\right)\right)$. Indeed, either $\rho_{\theta,h}\left(x_i, y_i\right) = \rho_h\left(x_i, y_i\right)$ or $\rho_{\theta,h}\left(x_i, y_i\right) = 2\rho \leq \rho_h\left(x_i, y_i\right)$, which implies the desired result. Furthermore, $\mathfrak{R}_m(\widetilde{\mathcal{H}}) \leq \frac{1}{\rho}\mathfrak{R}_m(\widetilde{\mathcal{H}})$ since $\Phi_\rho$ is a $\frac{1}{\rho}$-Lipschitz function. Therefore, for any $\delta > 0$, with probability at least $1 - \delta$, for all $h \in \mathcal{H}$:

$$R^{C^3}(f_\theta) \leq \widehat{R}^{C^3}(f_\theta) + \frac{2}{\rho}\mathfrak{R}_m(\tilde{\mathcal{H}}) + \sqrt{\frac{\log \frac{1}{\delta}}{2m}}, \tag{17}$$

and to complete the proof it suffices to show that $\mathfrak{R}_m(\widetilde{\mathcal{H}}) \leq 2k\mathfrak{R}_m\left(\Pi_1(\mathcal{H})\right)$.

Here $\mathfrak{R}_m(\widetilde{\mathcal{H}})$ can be upper-bounded as follows:

$$\mathfrak{R}_m(\tilde{\mathcal{H}}) = \frac{1}{m} \mathop{\mathbb{E}}_{S,\sigma}\left[\sup_{h \in \mathcal{H}} \sum_{i=1}^{m} \sigma_i \left(h\left(x_i, y_i\right) - \max_{y}\left(h\left(x_i, y\right) - 2\rho 1_{y=y_i}\right)\right)\right] \tag{18}$$

$$\leq \frac{1}{m} \mathop{\mathbb{E}}_{S,\sigma}\left[\sup_{h \in \mathcal{H}} \sum_{i=1}^{m} \sigma_i h\left(x_i, y_i\right)\right] + \frac{1}{m} \mathop{\mathbb{E}}_{S,\sigma}\left[\sup_{h \in \mathcal{H}} \sum_{i=1}^{m} \sigma_i \max_{y}\left(h\left(x_i, y\right) - 2\rho 1_{y=y_i}\right)\right]. \tag{19}$$

Now we bound the first term above. Observe that

$$\frac{1}{m} \mathop{\mathbb{E}}_{\sigma}\left[\sup_{h \in \mathcal{H}} \sum_{i=1}^{m} \sigma_i h\left(x_i, y_i\right)\right] = \frac{1}{m} \mathop{\mathbb{E}}_{\sigma}\left[\sup_{h \in \mathcal{H}} \sum_{i=1}^{m} \sum_{y \in \dagger} \sigma_i h\left(x_i, y\right) 1_{y_i = y}\right] \tag{20}$$

$$\leq \frac{1}{m} \sum_{y \in \dagger} \mathop{\mathbb{E}}_{\sigma}\left[\sup_{h \in \mathcal{H}} \sum_{i=1}^{m} \sigma_i h\left(x_i, y\right) 1_{y_i = y}\right] \tag{21}$$

$$= \sum_{y \in \dagger} \frac{1}{m} \mathop{\mathbb{E}}_{\sigma}\left[\sup_{h \in \mathcal{H}} \sum_{i=1}^{m} \sigma_i h\left(x_i, y\right) \left(\frac{\epsilon_i}{2} + \frac{1}{2}\right)\right] \tag{22}$$

where $\epsilon_i = 2 \cdot 1_{y_i = y} - 1$. Since $\epsilon_i \in \{-1, +1\}$, we have that $\sigma_i$ and $\sigma_i \epsilon_i$ admit the same distribution and each of the terms of the right-hand side can be bounded as follows:

$$\frac{1}{m} \mathop{\mathbb{E}}_{\sigma}\left[\sup_{h \in \mathcal{H}} \sum_{i=1}^{m} \sigma_i h\left(x_i, y\right) \left(\frac{\epsilon_i}{2} + \frac{1}{2}\right)\right] \tag{23}$$

$$\leq \frac{1}{2m} \mathop{\mathbb{E}}_{\sigma}\left[\sup_{h \in \mathcal{H}} \sum_{i=1}^{m} \sigma_i \epsilon_i h\left(x_i, y\right)\right] + \frac{1}{2m} \mathop{\mathbb{E}}_{\sigma}\left[\sup_{h \in \mathcal{H}} \sum_{i=1}^{m} \sigma_i h\left(x_i, y\right)\right] \tag{24}$$

$$\leq \widehat{\mathfrak{R}}_m\left(\Pi_1(\mathcal{H})\right) \tag{25}$$

Thus, we can write $\frac{1}{m}\mathbb{E}_{S,\sigma}\left[\sup_{h \in \mathcal{H}} \sum_{i=1}^{m} \sigma_i h\left(x_i, y_i\right)\right] \leq k\mathfrak{R}_m\left(\Pi_1(\mathcal{H})\right)$. To bound the second term, we derive

$$\frac{1}{m} \mathop{\mathbb{E}}_{S,\sigma}\left[\sup_{h \in \mathcal{H}} \sum_{i=1}^{m} \sigma_i \max_{y}\left(h\left(x_i, y\right) - 2\rho 1_{y=y_i}\right)\right] \leq \sum_{y \in \dagger} \frac{1}{m} \mathop{\mathbb{E}}_{S,\sigma}\left[\sup_{h \in \mathcal{H}} \sum_{i=1}^{m} \sigma_i\left(h\left(x_i, y\right) - 2\rho 1_{y=y_i}\right)\right], \tag{26}$$

and since Rademacher variables are mean zero, we observe that

$$\mathbb{E}_{S,\sigma}\left[\sup_{h\in\mathcal{H}}\sum_{i=1}^{m}\sigma_i\left(h\left(x_i,y\right)-2\rho 1_{y=y_i}\right)\right] = \mathbb{E}_{S,\sigma}\left[\sup_{h\in\mathcal{H}}\left(\sum_{i=1}^{m}\sigma_i h\left(x_i,y\right)\right)-2\rho\sum_{i=1}^{m}\sigma_i 1_{y=y_i}\right] \tag{27}$$

$$= \mathbb{E}_{S,\sigma}\left[\sup_{h\in\mathcal{H}}\sum_{i=1}^{m}\sigma_i h\left(x_i,y\right)\right] \leq \mathfrak{R}_m\left(\Pi_1(\mathcal{H})\right), \tag{28}$$

which completes the proof.

## B. Pseudo-Code

The pseudo-code of incorporating $C^3$R into the MML models is shown in Algorithm 1.

---

**Algorithm 1** Pseudo-Code of MML with $C^3$R

---

**Input**: MML data distribution $P_{XY}$; Randomly initialize MML model $f_\theta$ with feature extractor $g$ and a linear classifier $\mathcal{W}$; Hyperparameters $\lambda_v$ and $\lambda_{fe}$
**Output**: MML model $f_\theta$
1: Sample training datasets $\mathcal{D}_{tr} = \{(x_j^i, y_j^i)\}_{j=1}^{N}$ with $N$ samples from $P_{XY}$         ▷ Data Sampling
2: **for** each iteration **do**
3:     Calculate the multi-modal representation via $Z_i = g(x)$
4:     Obtain causal $\widehat{Z}_{c,i} = g^*(Z_i, v_i)$ from $Z_i$ based on the $V$ via Eq.6    ▷ Extract Causal Representation for Sufficiency
5:     Obtain counterfactual $\overline{Z}_{c,i} = \widehat{Z}_{c,i} + \Delta_i$ via Eq.7       ▷ Model Counterfactual Representation for Necessity
6:     Update the MML model $f_\theta$ using the original MML loss with $C^3$R in Eq.10       ▷ Update MML Model
7: **end for**
8: **return** solution

---

## C. Table of Notations

We list the definitions of all notations from the main text in Table 4.

## D. Discussion

### D.1. More Details of $C^3$ Definition

Recalling **Definition 3.1**, the learned representation of existing MML models can be divided into:

- Sufficient but Unnecessary Causes: $P(\mathrm{Y}_{do(\mathrm{Z}=c)} = y \mid \mathrm{Z} = \bar{c}, \mathrm{Y} \neq y) > 0$ and $P(\mathrm{Z} = \bar{c} \mid \mathrm{Y} = y) > 0$.
- Necessary but Insufficient Causes: $P(\mathrm{Y}_{do(\mathrm{Z}=\bar{c})} = y) = 0$ and $P(\mathrm{Y}_{do(\mathrm{Z}=c)} \neq y \mid \mathrm{Z} = \bar{c}, \mathrm{Y} \neq y) > 0$.
- Sufficient and Necessary Causes: $P(\mathrm{Y}_{do(\mathrm{Z}=c)} = y \mid \mathrm{Z} = \bar{c}, \mathrm{Y} = \bar{y}) > 0$ and $P(\mathrm{Y}_{do(\mathrm{Z}=\bar{c})} = y) = 0$.
- Insufficient and Unnecessary Causes: $P(\mathrm{Y}_{do(\mathrm{Z}=c)} = y \mid \mathrm{Z} = \bar{c}, \mathrm{Y} \neq y) = 0$ and $P(\mathrm{Y}_{do(\mathrm{Z}=\bar{c})} = y) = 0$

For the four parts of Z, we provide the detailed illustration:

- Sufficient but unnecessary: Z results in effect Y, yet the presence of Y does not definitively imply that Z is the cause;

- Necessary but insufficient: the occurrence of effect Y confirms that the cause is Z, but Z alone is not guaranteed to produce Y;

- Sufficient and necessary: the presence of effect Y invariably indicates cause Z, and the presence of Z invariably results in Y.

- Insufficient and unnecessary: Even if Z exists, it does not guarantee that Y will occur. Similarly, even if Y occurs, it cannot be determined that Z is the cause of Y because there are other possible causes.

Table 4: The definitions of notations.

| Notations | Definition |
|---|---|
| | **Notations of Data** |
| $\mathcal{X}, \mathcal{Y}$, and $\mathcal{Z}$ | The input space, label space, and latent space |
| $(x, y) \in \mathcal{X} \times \mathcal{Y}$ | The sample $x \in \mathcal{X}$ and the label $y \in \mathcal{Y}$ |
| $P_{XY}$ | The joint distribution over $\mathcal{X} \times \mathcal{Y}$ |
| $\mathcal{D}_{tr} = \{(x_i, y_i)\}_{i=1}^{N}$ | The training dataset with $N$ samples |
| $\mathcal{D}_{te} = \{(x_i, y_i)\}_{i=1}^{N_{te}}$ | The test dataset with $N_{te}$ samples |
| $x_i = \{x_i^{(1)}, \ldots, x_i^{(K)}\}$ | Each sample, which contains $K$ modalities |
| | **Notations of Model** |
| $f_\theta = \mathcal{W} \circ g$ | The MML model |
| $g : \mathcal{X} \to \mathcal{Z}$ | The feature extractor of MML model $f_\theta$ |
| $\mathcal{W} : \mathcal{Z} \to \mathcal{Y}$ | The classifier of MML model $f_\theta$ |
| | **Notations of Variables** |
| X, Y | Variables of the sample and corresponding label |
| Z | Variable of the learned representation, i.e., $Z = \Xi_c F_c + \Xi_s F_s$ |
| $F_c, F_s$ | Variables of the causal and non-causal generating factors |
| $\Xi_c, \Xi_s$ | Weight matrices of $F_c$ and $F_s$ that make up $Z$ |
| V | The instrumental variable satisfies $V \perp\!\!\!\perp F_s$ with a controllable impact $V \to F_c \to Y$ |
| | **Notations of $C^3$ and $C^3$R Learning Objective** |
| $\widehat{R}^{C^3}$ | $C^3$ risk |
| $\mathcal{L}_v$ | The loss based on the instrumental variable $V$, designed to guide the model toward the causal representation |
| $\mathcal{L}_{fe}$ | The feature extraction loss term based on counterfactual constructions, ensuring remain within the feasible region |
| $\lambda_v, \lambda_{fe}$ | The weights for $\mathcal{L}_v$ and $\mathcal{L}_{fe}$ |
| $\widehat{Z}_{c,i}$ | Real extraction version of Z for sample $x_i$ where the condition is $c$ (real-world branch) |
| $\overline{Z}_{c,i}$ | Counterfactual version of Z for sample $x_i$ where the condition is $\bar{c}$ (hypothetical-world branch) |
| $\Delta_i$ | Gradient-based adjustment for counterfactual data |
| $\mu_{\text{KL}}(\cdot)$ | Kullback-Leibler divergence |
| $\sigma$ | Signum function |
| $\rho(\cdot)$ | Indicator function which is equal to 1 if the condition in $\rho(\cdot)$ is true. |
| $s_{ij}$ | The alignment scores between the $i$-th and $j$-th modalities |
| $W_Q, W_K, W_V$ | Linear projection matrices for query, key, and value in the modeling of $V$ |
| $q_i$ | Query vector for the $i$-th modality |
| $k_j$ | Key vector for the $j$-th modality |
| $\alpha$ | Weighting factor for distance-based adjustment $\|q_i - k_j\|^2$ in the modeling of $V$ |

For the derivation of Eq.3 and Eq.4 $C^3(Z)$ can be decomposed into sufficiency and necessity components. Specifically, following (Pearl, 2009): (i) when $Z \neq c$ occurs with probability $1 - P(Z = c)$, its contribution reflects the sufficiency effect $C_{su}^3(Z)$ required for $Z = c$; (ii) when $Z = c$ occurs with probability $P(Z = c)$, its contribution reflects the necessity effect $C_{ne}^3(Z)$ of $Z = \bar{c}$ on Y. Therefore, we have $C^3(Z) = \left(1 - P(Z = c)\right) C_{su}^3(Z) + P(Z = c) C_{ne}^3(Z)$. By normalizing $C^3(Z)$ according to their respective weights, we obtain:

$$C_{su}^3(Z) = \frac{C^3(Z)}{1 - P(Z = c)} = \frac{P\left(Y_{do(Z=c)}\right) - P\left(Y_{do(Z=\bar{c})}\right)}{1 - P(Z = c)} = \frac{P(Y_{do(Z=c)}) - P(Y_{do(Z=\bar{c})})}{1 - P(Z = c)} \tag{29}$$

$$C_{ne}^3(Z) = \frac{C^3(Z)}{P(Z = c)} = \frac{P\left(Y_{do(Z=c)}\right) - P\left(Y_{do(Z=\bar{c})}\right)}{P(Z = c)} = \frac{P(Y_{do(Z=c)}) - P(Y_{do(Z=\bar{c})})}{P(Z = c)} \tag{30}$$

## D.2. Example to Understand $C^3$ Score

We have provided an example of causal sufficiency and necessity in MML tasks in **Section 2**. Inspired by (Yang et al., 2024), we also provide specific numerical calculations to further explain this example. This task aims to identify the label of "duck" using three modalities, i.e., image, text, and audio. Using this example, we further use probability to explain causal sufficiency and necessity. The illustration of this example is shown in **Figure 1**.

**Example of Causal Sufficiency**    If the learned representation is represented by the variable Z (taking binary values 1 or 0), we use it to predict the label of whether being "duck". If the learned representation contains the information of "duck paws", it is a sufficient but unnecessary cause because the MML data containing "duck paws" must have a "duck". However, a sample with"duck" might not contain "duck paws", e.g., duck swim in the lake. Assuming that $P(Y_{do(Z=1)} = 1) = 1$ and $P(Y_{do(Z=0)} = 0) = 0.5$, $P(Y = 1) = 0.75$, $P(Z = 1, Y = 1) = 0.5$, $P(Z = 0, Y = 0) = 0.25$, $P(Z = 0, Y = 1) = 0.25$. Then, following (Pearl, 2009), for the probability of sufficiency and necessity, we obtain:

$$\begin{cases} P(Y_{do(Z=1)} = 1|Y = 0, Z = 0) = \frac{P(Y_{do(Z=1)}=1)-P(Y=1)}{P(Y=0,Z=0)} = \frac{1-0.75}{P(Y=1,Z=1)} = 1 \\ P(Y_{do(Z=0)} = 0|Y = 1, Z = 1) = \frac{P(Y=1)-P(Y_{do(Z=0)}=1)}{P(Y=1,Z=1)} = \frac{0.75-0.5}{P(Y=1,Z=1)} = 0.5 \end{cases} \tag{31}$$

where the first line represents the probability of sufficiency and the second line represents the probability of necessity. Thus, the learned representation contains "laugh (positive)-good (positive)-rising tone (positive)" has a probability of being the sufficient but unnecessary cause.

**Example of Causal Necessity**    If the learned representation contains the information of "wings" Z (taking values 1 and 0, where 1 means "wings"), to predict Y ("duck" 1, other label 0). Since a "duck" must have "wings" but if give a sample with "bird", it may also have "wings". Assuming that $P(Y_{do(Z=1)} = 1) = 0.5$ and $P(Y_{do(Z=0)} = 0) = 1$, $P(Y = 1) = 0.25$, $P(Z = 1, Y = 1) = 0.25$, $P(Z = 0, Y = 0) = 0.5$, $P(Z = 0, Y = 1) = 0.25$. Then, for the probability of sufficiency and necessity, we obtain:

$$\begin{cases} P(Y_{do(Z=1)} = 1|Y = 0, Z = 0) = \frac{P(Y_{do(Z=1)}=1)-P(Y=1)}{P(Y=0,Z=0)} = 0.5 \\ P(Y_{do(Z=0)} = 0|Y = 1, Z = 1) = \frac{P(Y=1)-P(Y_{do(Z=0)}=1)}{P(Y=1,Z=1)} = 1 \end{cases} \tag{32}$$

where the first line represents the probability of sufficiency and the second line represents the probability of necessity. Thus, the learned representation has a probability of being the necessary but insufficient cause.

If we only focus on causal sufficient causes, we will lose important modality-specific information and affect generalizability; if we only focus on causal necessary causes, then decisions will be made incorrectly based on background knowledge, affecting discriminability. For sufficient and necessary causes, the probability of sufficiency or necessity should be 1. Together they ensure that the MML model can learn a representation that not only reflects different modal information, i.e., contains the semantics of all three modalities, but also targets primary events, i.e., the "duck" label.

### D.3. Discussion of Identifiability

For detailed analyses, in this subsection, we provide the definitions of exogeneity and monotonicity in MML settings, and illustrate the importance of the identifiability proposed in **Subsection 3.2**.

Firstly, we propose the definitions of exogeneity and monotonicity for $C^3$ based on (Pearl, 2009) as follows:

**Definition D.1** (Exogeneity). Variable Z is exogenous relative to variable Y if and only if the Y would potentially respond to conditions $c$ or $\bar{c}$ is independent of the actual value of Z. The intervention probability is identified by conditional probability $P(Y_{do(Z=c)} = y) = P(Y = y|Z = c)$.

Exogeneity (**Definition D.1**) means that when Z is exogenous to $Y$, external interventions (e.g., environmental disturbances or other unobserved factors) have a negligible impact on the conditional distribution of Z and $Y$. In other words, Z as a causal variable is not easily influenced by external environments or noise, ensuring that its causal effect on $Y$ remains stable across different modalities (image, text, audio, etc.). This assumption ensures the learned representation Z only contains causal generating factors $F_c$ without spurious correlations caused by $F_s$. It requires the model to retain relatively reliable predictive or inference capability even under cross-modal or cross-domain distribution shifts.

**Definition D.2** (Monotonicity). Variable $Y$ is monotonic relative to Z if and only if $Y_{do(Z=c)} = \bar{y} \wedge Y_{do(Z=c)} = y$ is false or $Y_{do(Z=\bar{c})} = y \wedge Y_{do(Z=c)} = \bar{y}$ is false, where $\bar{y} \neq y$. For probabilistic formulations, $P(Y_{do(Z=c)} = y, Y_{do(Z=\bar{c})} \neq y) = 0$ or $P(Y_{do(Z=c)} \neq y, Y_{do(Z=\bar{c})} = y) = 0$.

Monotonicity (**Definition D.2**) illustrates the consistent, unidirectional effect on Y of representation $Z$. If Z increases (or decreases) along some dimension, $Y$ correspondingly and consistently increases (or decreases). Such a monotonic constraint helps the model capture Z's causal effect on $Y$ more effectively, so that even if different modalities depict Z slightly differently, the overall causal direction remains consistent, improving the learned representation's generalization across modalities and scenarios. The assumption ensures that changes in all data variables have the same trend effect on the results. It simplifies the causal structure and is a common assumption for complex scenarios such as MML.

However, in practical MML settings, non-trivial cases often arise: for instance, the inseparability of causal and non-causal semantics leads to spurious correlations (as depicted in Figure 2), and cross-modal semantic conflicts combined with high-dimensional nonlinear interactions undermine monotonicity. These challenges render traditional identifiability conditions inapplicable. More specifically, enforcing exogeneity and monotonicity during model training can introduce multiple issues. First, it may oversimplify or deviate from real-world distributions, leading to large errors. As **Figure 2 shows**, real-world $F_c$ often cannot be perfectly resistant to external interventions, and the spurious correlations between $F_s$ and $Y$ will break exogeneity. In such cases, forcing exogeneity can lead to the model learning spurious correlations, undermining its applicability and accuracy. Likewise, if monotonicity does not hold in reality (e.g., certain modalities exhibit nonlinear or time-varying relationships with Z), the model may be biased or misjudge the direction of Z's effect on $Y$. Moreover, some systems have feedback loops in which changes in $Y$ can affect $F_c$ or other variables (e.g., user behavior and recommendation outcomes on social media), and imposing strictly one-way causal or monotonic relationships can ignore these dynamic interactions and distort the model. Additionally, from a performance perspective, if exogeneity and monotonicity assumptions are rigidly applied during training, the model may fail to generalize once deployed to new environments or tasks that do not meet these assumptions. Such "hard" constraints may lead the model to over- or underestimate the actual contribution of $F_c$ to $Y$, resulting in decision risks. Therefore, the discussion on identifiability in **Subsection 3.2** is crucial to ensure accuracy and generalizability of MML models in real-world applications.

### D.4. Reliability and Completeness of the Twin Network

Considering that the intervention value $\bar{c}$ does not necessarily come from the same distribution as Z (Pearl, 2009), we define the intervention variable $\bar{Z}$ has the same range as Z, where $\bar{c}$ comes from its distribution $P(\bar{Z}|X = x)$. Correspondingly, the estimated distribution is defined as $P(Z|X = x)$ and $P(\bar{Z}|X = x)$. Generally, the $C^3$ risk is formally defined as:

$$R^{C^3} := \mathbb{E}_{(x,y)\sim P_{XY}}\Big[\underbrace{\mathbb{E}_{c\sim P(Z|X=x)}\rho[\sigma(\mathcal{W}^\top c) \neq y]}_{\text{Sufficiency}} + \underbrace{\mathbb{E}_{\bar{c}\sim P(\bar{Z}|X=x)}\rho[\sigma(\mathcal{W}^\top \bar{c}) = y]}_{\text{Necessity}}\Big], \tag{33}$$

where $\mathcal{W}$ is the above invariant predictor, $\rho(\cdot)$ denotes an indicator function which is equal to 1 if the condition in $\rho(\cdot)$ is true, otherwise equals 0. However, as described in the main text (**Subsection 3.3**), it is difficult to directly obtain the evaluation of necessity, since the counterfactual data $\bar{c}$ is hard to acquire. Therefore, we employ a twin network to separately model the real-world branch for sufficiency evaluation and the hypothetical-world branch for necessity evaluation. The twin network for $C^3$ estimation in MML is shown in **Figure 5**. Then, we get:

$$\widehat{R}^{C^3} = \frac{1}{N}\sum_{i=1}^{N}\Big[\rho[\sigma(\mathcal{W}^\top \widehat{Z}_{c,i}) \neq y_i] + \rho[\sigma(\mathcal{W}^\top \overline{Z}_{c,i}) = y_i]\Big],$$

$$\text{where} \quad \widehat{Z}_{c,i} = g^*(x_i, v_i), \quad \overline{Z}_{c,i} = \widehat{Z}_{c,i} - \nabla_{\widehat{Z}_{c,i}}\ell\Big(\sigma(\mathcal{W}^\top \widehat{Z}_{c,i}), y_i\Big), \tag{34}$$

This equation satisfies $\mu_{\text{KL}}(\overline{Z}_{c,i}, \widehat{Z}_{c,i}) \leq \varepsilon, \quad \forall i$, also illustrated in **Theorem 3.5**. In this subsection, we discuss and prove the reliability and completeness of the proposed twin network.

The core idea of the twin network is to generate two representation branches for the same input $x$, which are close but slightly different, thereby modeling the real world and the hypothetical world in parallel. In this work, we aim to use the twin network to separately model causal sufficiency and necessity, corresponding to the real world and the hypothetical world: (i) the real world evaluates whether the extracted representation, i.e., $Z = c$, can yield the ground-truth label; (ii) the hypothetical world evaluates whether the extracted representation, under the intervention $Z = \bar{c}$, can still yield the ground-truth label. The key challenge lies in modeling counterfactual data $Z = \bar{c}$.

Note that the true counterfactual effect involves unobserved outcomes, making direct modeling difficult (Pearl, 2009). Existing work typically adopts the "minimal change" principle to estimate counterfactuals, which is proven to align with

true counterfactual effects (Pearl, 2009; Kusner et al., 2017): to study the causal effect of a variable or factor on the outcome, one should avoid making large modifications to other irrelevant factors. Instead, only the parts most sensitive to the outcome should be adjusted, ensuring that the original instance and its counterfactual counterpart remain as similar as possible, apart from the change in the target variable. For instance, Chapters III–V in (Wachter et al., 2017) demonstrate that by making only minor adjustments to the treatment variable while preserving the distribution of other covariates, the counterfactual samples maintain the original data's characteristics, ensuring accurate counterfactual effect estimates. Based on these results, we leverage gradient intervention to satisfy the "minimal change" principle for counterfactual effect estimation with theoretical guarantees. Inspired by this, we propose to first use instrumental variables to extract the causal representation $\widehat{Z}_c$ from input $x$ in the real-world branch. Simultaneously, in the hypothetical world, we compute $\overline{Z}_c = \widehat{Z}_c + \Delta$ along the gradient direction $\Delta$, constructing a relative "small-step" change strictly in the dimensions relevant to the causal mechanism. This approach allows $\overline{Z}_c$ to serve as an approximate counterfactual representation for the same input. The core idea lies in: (i) constraining the gradient direction to ensure that counterfactual modeling does not deviate from the true implementation Z in the latent space (Berger & Gostiaux, 2012); (ii) applying minimal changes to satisfy the "minimal change" principle of counterfactual modeling. Under the Markov assumption (Frydenberg, 1990; Blumenthal, 1957), once $\widehat{Z}_c$ is given, any disturbance variables will be independent of Y. Therefore, if $\widehat{Z}_c$ needs to flip, it is sufficient to make small-step modifications to $\widehat{Z}_c$ until the label changes, aligning with the essence of counterfactual reasoning: if I slightly modify $\widehat{Z}_c$, will Y remain the same or change? As long as these small perturbations are enough to alter the model's output, they effectively simulate the counterfactual condition of the prediction.

To assess whether the generated counterfactual data adhere to the principle for accurate estimation, we develop a distribution consistency test with Wasserstein distance $D_w$, i.e., whether the distribution of the covariates matches that of the original data. The lower $D_w$ shown below proves that our method satisfies the principle. Besides, we also conduct a toy experiment on LCKD and NYU Depth V2 to demonstrate credibility. We follow (Galles & Pearl, 1998) to make manually curated examples and select the recently proposed transport-based counterfactual modeling method (De Lara et al., 2024) as another baseline. Table 5 shows the advantages of our method, i.e., superior accuracy and lowest computational cost.

Table 5: Model performance and calculation overhead

| Model | Accuracy | Calculation Overhead |
|---|---|---|
| $C^3$R | 77.6 | $1\times$ |
| manually curated example | 71.2 | $4.1\times$ |
| Transport-based | 77.2 | $2.3\times$ |

We further provide an illustration of the twin network in **Figure 5**. The left (orange) region contains nodes $X, F_c, F_s, Y$ to represent the causal structure of the real world, while the right (purple) region contains $X^*, F_c^*, F_s^*, Y^*$ to indicate a hypothetical (counterfactual) scenario. Both branches connect to the overhead nodes $D, V$, where $D$ denotes the variable for data generation and $V$ denotes the instrumental variable, meaning that they share certain background and content variable but can diverge in some causal nodes. Following (Pearl, 2009), in causal inference or counterfactual analysis, if we aim to compare how the world actually is (the real scenario) with how the world would be under certain interventions (the hypothetical/counterfactual scenario), we naturally create two similar branches in a diagram, sharing part of the background or parent nodes (such as $D, V$) but intervening in certain causal nodes ($F_c^*$ or $X^*$). The twin network shows two conditions of the same model.

Based on the above analyses, we propose the following propositions to expand the ability to identify causal effects:

**Proposition D.3** (Local Invertibility). *Let $M$ be a twin network that operates on exogenous/background variables s. Suppose that in every local neighborhood of $M$, the mapping $P(Y \mid \mathrm{Pa}(Y)) \longmapsto P(s)$ is injective, i.e. from the conditional distribution of Y given its parents $\mathrm{Pa}(Y)$, one can uniquely solve for the distribution $P(s)$. Then, any counterfactual statement in $M$ is uniquely identifiable from the observational data.*

**Proposition D.3** tells us that, in each neighborhood, specifying $\mathrm{Pa}(Y) = v$ enforces a unique distribution of $s$. So once we move $\mathrm{Pa}(Y)$ from the real-world value to an intervened value $v'$ (the $do(x')$ operation), that new local condition forces a unique $s$. If, hypothetically, local invertibility did not hold, multiple distinct distributions over $s$ could give the same conditional $P(Y \mid \mathrm{Pa}(Y) = v)$, leading to multiple solutions for the counterfactual. But since local invertibility excludes

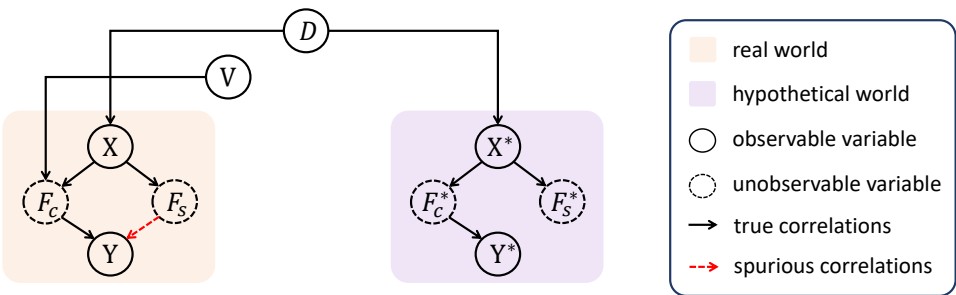

Figure 5: Twin Network for $C^3$ estimation in MML.

that possibility, the solution for each neighborhood is unique—hence identifiability. Thus, under local invertibility, any local or global counterfactual query in the twin network can be computed from observational data.

*Proof.* A general counterfactual statement $\alpha$ in causal inference (for example, If $X$ had been $x'$, then $Y$ would have been $y'$.) often demands integrating over all possible exogenous-variable configurations $s$. Formally, evaluating

$$P(\alpha) \;=\; \int P(\alpha \,|\, s)\, P(s)\, \mathrm{d}s$$

in principle calls for specifying how $s$ is distributed and how it changes under different local conditions.

If in a neighborhood $\mathcal{N}$, we know the conditional distribution $P\big(Y \mid \mathrm{Pa}(Y) = v\big)$ for some value $v$, the local invertibility assumption states that precisely one distribution over $s$ leads to that conditional. Symbolically,

$$P\big(Y \,|\, \mathrm{Pa}(Y) = v\big) \;\mapsto\; \{\, P(s) \,\}_{\mathrm{unique}}.$$

Hence, specifying $\mathrm{Pa}(Y) = v$ fixes how $s$ must be distributed.

In computing a counterfactual probability, e.g. $P(Y_x = y')$, one typically modifies certain parent values $\mathrm{Pa}(Y) \to v'$ and then integrates out $s$. Because local invertibility provides a single assignment to $s$ in the local region, we obtain a unique predicted outcome distribution. There is no ambiguity about which exogenous states correspond to that local condition.

Since any condition on $\mathrm{Pa}(Y)$ in the local domain yields a unique exogenous distribution, any if–then statement about local changes in $\mathrm{Pa}(Y)$ can be pinned down. As long as the twin network $M$ remains locally invertible in all relevant neighborhoods, we can piece together all such local analyses to conclude that any finite or infinite combination of counterfactual statements is identifiable.

**Proposition D.4** (Markov Identifiability). *If the twin network $M$ is semi-Markovian, and furthermore satisfies graphical criteria (Pearl, 2009), then the quantities $C^3$, $C_{su}^3$, and $C_{ne}^3$ are all identifiable within such a causal model, and the causal effect can be uniquely determined from the topology of the graph.*

**Proposition D.4** shows that whenever the graph meets the semi-Markovian plus the cited conditions, the causal effect and the individual-level necessity/sufficiency measures are identified from the topology alone. Accoding to (Norton, 2009), if the confounders appear in specific ways (with certain back-door or front-door paths, or no cyclical bridging in the unobserved structure), we can factor the joint distribution into purely observational data pieces or partial adjustments. This also extends to multi-world queries needed by $C^3$, $C_{su}^3$, and $C_{ne}^3$. Meanwhile, if there were multiple expressions consistent with the same graph, we would lose identifiability (Tangirala, 2018). But the theorems ensure a single final expression for each of $C^3$, $C_{su}^3$, and $C_{ne}^3$, establishing that these cross-world probabilities cannot be ambiguous.

### D.5. Multi-modal Representation Learning on Synthetic Data

To assess the effectiveness of the proposed $C^3$R in capturing the critical causal information that serves as both sufficient and necessary causes, we have constructed synthetic data that encompasses four distinct categories of variables following (Yang et al., 2024), i.e., sufficient and necessary causes (SNC), sufficient but unnecessary causes (SC), necessary but insufficient causes (NC), and spurious correlations (SP). Among them, the first three are the information contained in the invariant

representation currently learned in standard cases, while the last one is artificially constructed and used for evaluation in practice.

Specifically, we first construct different multi-modal data distributions, then generate labels from the known data distributions, and finally constrain their correlations to achieve the construction of different causes. We assume the task contains three modalities and construct the following generating functions. Note that the first three correspond to the three parts of the invariant representation learned by the current MML methods, i.e., sufficient but unnecessary, necessary but insufficient, and sufficient and necessary causes, while the last, i.e., spurious correlations, is for evaluation.

**Sufficient and Necessary Cause (SNC)**  Each modality of the sufficient and necessary cause is generated according to a Bernoulli distribution following (Yang et al., 2024), i.e., $\text{SNC}_i \sim \mathcal{B}(\xi_a), a = 1, 2, 3$, where $\xi$ represents the Bernoulli distribution parameters corresponding to different modalities, e.g., $\text{SNC}_i \sim \mathcal{B}(0.5), \xi_a = 0.5$. The data label $y_i$ is generated based on the sufficient and necessary cause, where $y_i = \text{SNC}_i \circ \mathcal{B}(\xi'_a)$, e.g., $\text{SNC}_i \sim \mathcal{B}(0.15)$ when $\xi_a = 0.5$. Since Y is generated from each modality corresponding $\text{SNC} \in \{0, 1\}$. The probability of the sufficient and necessary cause is $P(\text{Y} = 0|do(\text{SNC} = 0)) + P(\text{Y} = 1|do(\text{SNC} = 1))$.

**Sufficient but Unnecessary Cause (SC)**  Sufficiency indicates that the presence of a representation aids in establishing the accuracy of the label, i.e., $\text{SC} \rightarrow \text{Y}$. According to the definition of $C^3$, when SF is defined as $f_{\text{SC}}(\text{SNC})$, the distribution of intervention $P(\text{Y}|do(\text{SC} = \text{SC}_i))$ is determined by the conditional distribution $P(\text{Y}|\text{SC} = \text{SC}_i)$, where $P(\text{Y}|do(\text{SC} = \text{SC}_i)) = \int P(\text{Y}|do(\text{SNC}))P(\text{SNC}|f_{\text{SC}}(\text{SNC}) = \text{SC}_i)d\,\text{SNC} = \int P(\text{Y}|\text{SNC})P(\text{SNC}|f_{\text{SC}}(\text{SNC}) = \text{SC}_i)d\,\text{SNC}$. Even Y is only generated from SNC, the sufficient cause SC is exogenous relative to Y. Meanwhile, according to the identifiability results of $C^3$, the sufficient but unnecessary cause $\text{SC} \in \{0, 1\}$ in synthetic data has the same probability with SNC, expressed as $P(\text{Y} = 1|do(\text{SC} = 1)) = P(\text{Y} = 1|do(\text{SNC} = 1))$. However, it has a lower probability of $P(\text{Y} = 0|do(\text{SC} = 0))$ compared to $P(\text{Y} = 0|do(\text{SNC} = 0))$. To determine the value of $\text{SC}_i$, we use a transformation function $f_{\text{SC}} : \{0, 1\} \rightarrow \{0, 1\}$ to derive $\text{SC}_i$ from the sufficient and necessary cause value $\text{SNC}_i$ for each modality. The generation process of $\text{SC}_i$ can be expressed as: (i) $\text{SC}_i = \mathcal{B}(\xi_a)$ when $\text{SNC}_i = 0$; and (ii) $\text{SC}_i = \text{SNC}_i$ when $\text{SNC}_i = 1$.

**Necessary but Insufficient Cause (NC)**  Necessity indicates that the label becomes invalid when the representations are absent, i.e., $\text{NC} \leftarrow \text{Y}$, which reflects the general situation of different modal information, i.e., contains the semantics of all modalities. Similarly, we get: $P(\text{Y}|do(\text{NC} = \text{NC}_i)) = \int P(\text{Y}|do(\text{SNC}))P(\text{SNC}|f_{\text{NC}}(\text{SNC}) = \text{NC}_i)d\,\text{SNC} = \int P(\text{Y}|\text{SNC})P(\text{SNC}|f_{\text{NC}}(\text{SNC}) = \text{NC}_i)d\,\text{SNC}$. Based on the definition and identifiability outcomes of $C^3$, the cause that is insufficient but necessary exhibits an equivalent probability to $P(\text{Y} = 0|do(\text{NC} = 0))$ as $P(\text{Y} = 0|do(\text{SNC} = 0))$, yet it diminishes the likelihood of $P(\text{Y} = 1|do(\text{NC} = 1))$ compared to $P(\text{Y} = 1|do(\text{SNC} = 1))$. To determine the value of NC, we employ a transformation function $f_{\text{NC}} : \{0, 1\} \rightarrow \{0, 1\}$ to derive $\text{NC}_i$ from both sufficient and necessary cause $\text{SNC}_i$. The process of generating $\text{NC}_i$ is outlined below, and $\text{NC}_i$ serves as the cause of $y$: $\text{NC}_i = f_{\text{NC}}(\text{SNC}_i) := \text{SNC}_i * \mathcal{B}(\xi_b), b = 1, 2, 3$ where $\xi_b$ equals 0.5, 0.7, and 0.9, respectively.

**Spurious Correlations (SP)**  Spurious correlations indicate data in the learned representation that is irrelevant to the decision, e.g., background information and noise. Although the learned invariant representations do not contain noise information and false correlations are not separately considered in the scenarios involved in this study because the model satisfies the corresponding identifiability through the four constraints corresponding to exogeneity and monotonicity, we still construct relevant data to examine whether false correlations are learned in reality. We introduce an additional variable that exhibits spurious correlation with both the sufficient and necessary causes. The level of spurious correlation is determined by a parameter denoted as $\omega$. The generation process for this variable is defined as $\text{SP}_i = \omega * \text{SNC}_i * \mathbf{1}_d + (1 - \omega)\mathcal{N}(0, 1)$, where $\mathcal{N}(0, 1)$ represents a Gaussian distribution, $\mathbf{1}_d$ represents a vector of ones of dimension $d$, and $d$ is set to 5 in the synthetic generation process. As $\omega$ increases, the strength of the spurious correlation within the data sample intensifies. For our synthetic experiments, we choose $s = 0.1$ and $s = 0.7$ to explore different levels of spurious correlation. Note that the SP data here is artificially constructed and has been used for evaluation, and it involves the corner case, e.g., anti-causal and confounder, which has been discussed for evaluation.

**Construction of MMLSynData for Evaluation**  After obtaining the above functions, we now introduce how to generate the synthetic dataset for evaluation (the experiments of "Learning causal complete representations" in **Subsection 6.2**) following (Yang et al., 2024). Firstly, we introduce a nonlinear transformation to generate the multi-modal samples $x$ from the variables $\{\text{SNC}_i, \text{SC}_i, \text{NC}_i, \text{SP}_i\}$, where each sample consists three modalities (functions with different parameters).

Initially, we create a temporary vector with Gaussian noise, i.e., $v = [\text{SNC}_i * \mathbf{1}_d, \text{SC}_i * \mathbf{1}_d, \text{NC}_i * \mathbf{1}_d, \text{SP}_i * \mathbf{1}_d] + \mathcal{N}(0, 0.4)$, where $\mathbf{1}_d$ represents a vector of ones of dimension $d$, and $\mathcal{N}(0, 0.4)$ denotes the Gaussian noise with mean 0 and variance 0.4. Next, following (Dandekar et al., 2018), we use two functions, i.e., $v_1(v)$ subtracts 0.5 from each element of $v$ if $v$ is greater than 0, otherwise, it sets the output to 0; and $v_2(v)$ adds 0.5 to each element of $v$ if $v$ is less than 0, otherwise, it sets the output to 0. Finally, the vector $v$ is generated using the sigmoid function applied element-wise to the product of $v_1(v)$ and $v_2(v)$, i.e., $v = \text{sigmoid}(v_1(v) \cdot v_2(v))$. For the training phase, we generate 1,000 multi-modal samples for training, while generating 200 samples for evaluation. We call this MML synthetic dataset (MMLSynData).

### D.6. Comparison and Uniqueness Discussion

"Causal sufficiency and necessity" is an important concept in causal theory, which is proposed in Book "Causality" (Pearl, 2009). Recently, (Yang et al., 2024) applied it to domain generalization. To clearly highlight the novelty of our work and differentiate it from existing studies, we outline the key distinctions.

**Uniqueness of the $C^3$ Concept and Modeling** We introduce the concept of Probability of Causal Complete Cause ($C^3$), which serves as a foundational element in our framework for MML. For definition and extension: $C^3$ quantifies the likelihood that a representation Z is both a necessary and sufficient cause for the label Y. It extends the PNS concept of (Pearl, 2009) to accommodate the complexities of MML, involving distinct feature extraction and fusion mechanisms, while (Yang et al., 2024) is for domain generalization. They both based on the concept of (Pearl, 2009) but focus on different problems. Causal Identifiability: We propose identifiability of $C^3$ in real-world MML systems, relaxing the strong assumptions of Exogeneity and Monotonicity that previous studies (Zhang et al., 2024; von Kügelgen, 2024; Yang et al., 2024) relies on. We consider additional conditions specific to MML, such as handling missing modalities and confounders, thereby enhancing the robustness of causal inference in multimodal settings. Meanwhile, we introduce instrumental variables to ensure the identifiability of the model even in corner cases, e.g., in the presence of spurious correlations and noise. Next, for measurement and modeling, we introduce the $C^3$ risk as a metric to estimate the $C^3$ score of multimodal representations on unseen test datasets. We directly model the sufficiency and necessity of the learned representations via a provable twin network, while (Yang et al., 2024) replacing the necessity with counterfactual modeling via a theoretical upper bound based on monotonicity; (Schölkopf et al., 2021) presents a review for causal inference; (Sun et al., 2025; Yao et al., 2024; Ahuja et al., 2023) mainly focus on identifiability under different settings and problems, e.g., weak supervision and partial observability, instead of the concept of causal sufficiency and necessity; (Wang & Jordan, 2024) aim to construct measures of non-spuriousness and disentanglement, where the exploitation of causal necessity and sufficiency concept is to align it with non-spuriousness and "invoke" the corresponding measure in (Pearl, 2009) to construct the measure of non-spuriousness. Our framework is able to adaptively capture complex patterns and potential relationships in the data without relying on pre-set assumptions. Meanwhile, it can construct a more complex model structure without being limited by the high-dimensional and nonlinear conditions.

**Constraining Causal Completeness** Our methodology for enforcing causal sufficiency differs significantly from that of previous theory or causal-related works (Yang et al., 2024; Kim et al., 2024; Sun et al., 2025; Yao et al., 2024; Ahuja et al., 2023). For the objective function, as outlined in Section 5, our $C^3R$ objective incorporates both effectiveness and reliability. The effectiveness component addresses causal completeness of the learned representation, with the generalization guarantee, while the reliability component ensures mitigates the effects of spurious correlations and the accuracy of counterfactual modeling (for necessity) through additional constraints. It is inspired from the proposed measurement. The objective of (Yang et al., 2024) relies on the proposed generalization bounds, where the foundation is different. For implementation and experimental Settings: Our approach employs distinct feature extractors for each modality, followed by an MLP layer for fusion, tailored to the diverse data distributions inherent in MML. This contrasts with (Yang et al., 2024), which utilizes a shared feature extractor suitable for domain generalization across different domains. Furthermore, our experimental results demonstrate that $C^3R$ consistently improves performance across various MML scenarios.

In summary, while both our work and previous works (Yang et al., 2024; Sun et al., 2025; Yao et al., 2024; Ahuja et al., 2023; Brehmer et al., 2022) draw inspiration from key concepts in causal theory, they diverge significantly in their problem settings, motivations, theoretical foundations, optimization strategies, empirical validations, etc. Our introduction of causal sufficiency and necessity tailored for MML, the novel $C^3$ concept, and the comprehensive $C^3R$ framework collectively advance the field of MML beyond the scope of existing domain generalization approaches.

# E. Benchmark Datasets

In this section, we briefly introduce all datasets used in our experiments. In summary, the benchmark datasets can be divided into four categories: (i) scenes recognition on two datasets, i.e., NYU Depth V2 (Silberman et al., 2012) and SUN RGBD (Song et al., 2015) datasets, with two modalities, i.e., RGB and depth images; (ii) image-text classification on two datasets, i.e., UPMC FOOD101 (Wang et al., 2015) and MVSA (Niu et al., 2016) datasets, with two modalities, i.e., image and text; (iii) segmentation when consider missing modalities on the BraTS dataset (Menze et al., 2014; Bakas et al., 2018) with four modalities, i.e., Flair, T1, T1c, and T2; and (iv) MMLSynData mentioned in **Appendix D.5** which is an MML synthetic dataset that used to evaluate whether learned representations contain causal sufficiency and necessity. The composition of the data set is as follows:

- **NYU Depth V2** (Silberman et al., 2012) is an indoor scene dataset captured by New York University using Microsoft Kinect's RGB and depth cameras. It includes 1,449 labeled RGB and depth images across 464 distinct indoor scenes from three cities, along with 407,024 unlabeled images.

- **SUN RGBD** (Song et al., 2015) is a scene understanding dataset released by the Vision & Robotics Group at Princeton University. It comprises 10,335 RGB-D images of indoor scenes, which are pairs of color and depth images. These images were captured using four different types of 3D cameras, including the Intel RealSense, Asus Xtion, Kinect v1, and Kinect v2. Each image in the dataset has been meticulously annotated with 2D polygonal segmentation and 3D bounding boxes.

- **UPMC FOOD101** (Wang et al., 2015) is a comprehensive food recognition dataset consisting of 101,000 images across 101 categories. Each category features 750 training images and 250 test images. Notably, the images are stored in JPEG format and are uniformly resized to a maximum dimension of 512 pixels.

- **MVSA** (Niu et al., 2016) is a multimodal biometric dataset that includes a variety of biometric samples such as fingerprints, iris, face, and hand shapes. It is designed to support research in the fields of biometric recognition and security applications. The MVSA dataset typically comprises a rich collection of samples with image and text modalities.

- **BraTS** (Menze et al., 2014; Bakas et al., 2018) aims to segment specific areas within brain tumors, which are identified as the enhancing tumor (ET), the tumor core (TC), and the entire tumor (WT). The dataset is composed of 3D multi-modal MRI scans of the brain, featuring modalities such as Flair (Fl), T1, T1 contrast-enhanced (T1c), and T2, all of which come with corresponding ground-truth segmentations. It includes a training set of 285 cases and an evaluation set of 66 cases. While the ground-truth annotations for the training cases are accessible to the public, those for the validation set remain undisclosed.

- **MMLSynData** aims to analyze whether the learned representations contain causal sufficiency and necessity. It contains four types of data, i.e., sufficient and necessary causes (SNC), sufficient but unnecessary causes (SC), necessary but insufficient causes (NC), and spurious correlations (SP). Each type of data uses 250 MML samples for training and 50 groups for evaluation. The construction details and corresponding functions are described in **Appendix D.5**.

# F. Baselines

For comprehensive evaluation of the proposed $C^3R$, we select 5 types of comparison baselines for evaluation, which covers almost all types of MML baselines including (i) large model and foundation model for MML, i.e., CLIP (Sun et al., 2023), ALIGN (Jia et al., 2021), CoOp (Jia et al., 2022a), MaPLe (Khattak et al., 2023), and VPT (Jia et al., 2022a); (ii) classic MML methods, i.e., RGB (Zhang et al., 2023c), Depth (Zhang et al., 2023c), Late fusion (Wang et al., 2016), ConcatMML (Zhang et al., 2021), and AlignMML (Wang et al., 2016); (iii) strong unimodal baselines and the corresponding multi-modal methods, i.e., Bow (Zhang et al., 2023c), Img (Zhang et al., 2023c), BERT (Zhang et al., 2023c), ConcatBow (Zhang et al., 2023c), and ConcatBERT (Zhang et al., 2023c); (iv) recently proposed and SOTA MML methods, i.e., MMTM (Joze et al., 2020) and TMC (Han et al., 2020), LCKD (Wang et al., 2023b), UniCODE (Xia et al., 2024), SimMMDG (Dong et al., 2024), MMBT (Kiela et al., 2019), and QMF (Zhang et al., 2023c); and (v) MML methods that specifically designed for missing modalities, i.e., HMIS (Havaei et al., 2016), HVED (Dorent et al., 2019), RSeg (Chen et al., 2019), mmFm (Zhang et al., 2022b), and LCKD (Wang et al., 2023b).

Table 6: Full results (with error bars) of scenes recognition performance on NYU Depth V2 (Silberman et al., 2012) and SUN RGBD (Song et al., 2015) with RGB and depth images. "(N, Avg.)" and "(N, Worst.)" denotes the average and worst-case accuracy. The results show the error of accuracy by executing the experiments randomly 3 times on 40 randomly selected hyperparameters. The best results are highlighted in **bold**.

| Method | NYU Depth V2 | | | | SUN RGB-D | | | |
|---|---|---|---|---|---|---|---|---|
| | (0,Avg.) | (0,Worst.) | (10,Avg.) | (10,Worst.) | (0,Avg.) | (0,Worst.) | (10,Avg.) | (10,Worst.) |
| CLIP (Sun et al., 2023) | 69.32±0.35 | 68.29±0.36 | 51.67±0.42 | 48.54±0.36 | 56.24±0.51 | 54.73±0.31 | 35.65±0.47 | 32.76±0.54 |
| ALIGN (Jia et al., 2021) | 66.43±0.36 | 64.33±0.32 | 45.24±0.47 | 42.42±0.38 | 57.32±0.52 | 56.26±0.36 | 38.43±0.42 | 35.13±0.52 |
| MaPLe (Khattak et al., 2023) | 71.26±0.32 | 69.27±0.35 | 52.98±0.45 | 48.73±0.37 | 62.44±0.54 | 61.76±0.33 | 34.51±0.42 | 30.29±0.54 |
| CoOp (Jia et al., 2022a) | 67.48±0.34 | 66.94±0.32 | 49.43±0.44 | 45.62±0.32 | 58.36±0.50 | 56.31±0.36 | 39.67±0.41 | 35.43±0.55 |
| VPT (Jia et al., 2022a) | 62.16±0.36 | 61.21±0.31 | 41.05±0.47 | 37.81±0.32 | 54.72±0.54 | 53.92±0.32 | 33.48±0.44 | 29.81±0.51 |
| RGB (Zhang et al., 2023c) | 63.33±0.35 | 62.54±0.32 | 45.46±0.47 | 42.20±0.38 | 56.78±0.51 | 56.51±0.36 | 42.94±0.45 | 41.02±0.54 |
| Depth (Zhang et al., 2023c) | 62.65±0.37 | 61.01±0.34 | 44.13±0.48 | 35.93±0.32 | 52.99±0.55 | 51.32±0.32 | 35.63±0.40 | 33.07±0.54 |
| Late fusion (Wang et al., 2016) | 69.14±0.33 | 68.35±0.32 | 51.99±0.49 | 44.95±0.32 | 62.09±0.58 | 60.55±0.38 | 47.33±0.46 | 44.60±0.57 |
| ConcatMML (Zhang et al., 2021) | 70.30±0.32 | 69.42±0.36 | 53.20±0.47 | 47.71±0.31 | 61.90±0.52 | 61.19±0.37 | 45.64±0.41 | 42.95±0.50 |
| AlignMML (Wang et al., 2016) | 70.31±0.36 | 68.50±0.37 | 51.74±0.42 | 44.19±0.39 | 61.12±0.52 | 60.12±0.33 | 44.19±0.47 | 38.12±0.51 |
| Bow (Zhang et al., 2023c) | 61.38±0.35 | 59.23±0.36 | 37.98±0.42 | 34.24±0.38 | 54.37±0.50 | 54.11±0.34 | 39.07±0.47 | 36.43±0.53 |
| Img (Zhang et al., 2023c) | 43.27±0.33 | 42.96±0.34 | 29.27±0.46 | 28.53±0.37 | 36.28±0.54 | 35.26±0.36 | 21.32±0.40 | 20.31±0.53 |
| BERT (Zhang et al., 2023c) | 65.31±0.36 | 63.23±0.32 | 38.64±0.47 | 36.45±0.33 | 57.98±0.52 | 56.74±0.35 | 42.51±0.41 | 38.53±0.57 |
| ConcatBow (Zhang et al., 2023c) | 49.64±0.36 | 48.66±0.34 | 31.43±0.45 | 29.87±0.30 | 41.25±0.54 | 40.54±0.32 | 26.76±0.49 | 24.27±0.58 |
| ConcatBERT (Zhang et al., 2023c) | 70.56±0.34 | 69.83±0.36 | 44.52±0.46 | 43.29±0.34 | 59.76±0.52 | 58.92±0.34 | 45.85±0.42 | 41.76±0.54 |
| MMTM (Joze et al., 2020) | 71.04±0.36 | 70.18±0.34 | 52.28±0.42 | 46.18±0.33 | 61.72±0.56 | 60.94±0.31 | 46.03±0.44 | 44.28±0.57 |
| TMC (Han et al., 2020) | 71.06±0.34 | 69.57±0.32 | 53.36±0.42 | 49.23±0.39 | 60.68±0.54 | 60.31±0.30 | 45.66±0.46 | 41.60±0.50 |
| LCKD (Wang et al., 2023b) | 68.01±0.31 | 66.15±0.34 | 42.31±0.45 | 40.56±0.38 | 56.43±0.56 | 56.32±0.32 | 43.21±0.49 | 42.43±0.54 |
| UniCODE (Xia et al., 2024) | 70.12±0.37 | 68.74±0.32 | 44.78±0.48 | 42.79±0.39 | 59.21±0.55 | 58.55±0.36 | 46.32±0.47 | 42.21±0.57 |
| SimMMDG (Dong et al., 2024) | 71.34±0.32 | 70.29±0.31 | 45.67±0.41 | 44.83±0.39 | 60.54±0.50 | 60.31±0.37 | 47.86±0.43 | 45.79±0.56 |
| MMBT (Kiela et al., 2019) | 67.00±0.35 | 65.84±0.34 | 49.59±0.41 | 47.24±0.38 | 56.91±0.51 | 56.18±0.36 | 43.28±0.47 | 39.46±0.51 |
| QMF (Zhang et al., 2023c) | 70.09±0.30 | 68.81±0.34 | 55.60±0.42 | 51.07±0.34 | 62.09±0.50 | 61.30±0.36 | 48.58±0.46 | 47.50±0.58 |
| CLIP+$C^3$R | 76.54±0.89 (+7.22) | 75.12±1.20 (+6.83) | 56.73±0.83 (+5.06) | 52.90±1.03 (+4.36) | 62.31±1.51 (+6.07) | 58.71±0.91 (+3.98) | 41.59±1.19 (+5.94) | 37.52±0.78 (+4.76) |
| ALIGN+$C^3$R | 71.92±0.96 (+5.49) | 70.33±0.91 (+6.00) | 52.59±1.42 (+7.35) | 51.26±1.58 (+8.84) | 62.99±1.36 (+5.67) | 61.95±1.06 (+5.69) | 46.08±1.72 (+7.65) | 41.95±1.02 (+6.82) |
| MaPLe+$C^3$R | 77.07±1.16 (+5.81) | 74.45±1.23 (+5.18) | 58.94±1.17 (+5.96) | 55.95±1.41 (+7.22) | 66.21±1.85 (+3.77) | 65.51±1.23 (+3.75) | 40.12±1.51 (+5.61) | 37.34±1.03 (+7.05) |
| Late fusion+$C^3$R | 73.26±1.05 (+4.12) | 71.62±0.77 (+3.27) | 57.21±0.85 (+5.22) | 50.98±1.51 (+6.03) | 64.84±1.25 (+2.75) | 63.25±1.32 (+2.70) | **53.35±0.81** (+6.02) | 50.43±1.60 (+5.83) |
| ConcatMML+$C^3$R | 75.97±1.51 (+5.67) | 75.95±0.90 (+6.53) | 60.32±1.43 (+7.12) | 55.02±0.83 (+7.31) | 67.17±1.20 (+5.27) | 66.73±1.30 (+5.54) | 52.28±1.48 (+6.64) | 50.42±1.11 (+7.47) |
| Bow+$C^3$R | 65.77±0.93 (+4.39) | 65.22±1.42 (+5.99) | 44.82±1.64 (+6.84) | 42.88±1.04 (+8.64) | 58.15±1.41 (+4.78) | 57.52±0.67 (+3.41) | 47.37±1.11 (+8.30) | 43.85±1.42 (+7.42) |
| LCKD+$C^3$R | 77.14±1.62 (+9.13) | **75.12±1.62** (+8.97) | 50.11±1.59 (+1.80) | 47.98±0.99 (+7.42) | 60.97±1.40 (+4.54) | 60.14±0.79 (+3.82) | 47.23±1.65 (+4.02) | 46.21±1.65 (+3.78) |
| UniCODE+$C^3$R | 76.52±1.42 (+6.40) | 74.39±1.70 (+5.65) | 51.51±1.41 (+6.73) | 48.09±1.05 (+5.3) | 65.78±1.44 (+6.57) | 64.49±1.63 (+5.94) | 51.42±1.52 (+5.10) | 49.70±1.66 (+7.49) |
| SimMMDG+$C^3$R | 75.32±1.29 (+3.98) | 74.61±1.00 (+4.32) | 49.99±1.24 (+4.32) | 47.22±1.60 (+2.65) | 65.50±1.66 (+4.96) | 64.58±1.44 (+4.27) | 52.69±1.57 (+4.83) | **51.70±1.21** (+5.91) |
| MMBT+$C^3$R | 73.74±1.26 (+6.74) | 71.82±1.22 (+5.98) | 54.35±1.51 (+4.76) | 52.57±1.58 (+5.33) | 61.47±1.67 (+4.56) | 59.99±1.15 (+3.81) | 48.42±1.01 (+5.14) | 46.07±1.44 (+6.61) |
| QMF+$C^3$R | **77.58±1.37** (+7.49) | 74.95±1.32 (+6.14) | **59.72±1.60** (+4.12) | **59.18±1.35** (+8.11) | **67.35±1.12** (+5.26) | **65.84±1.26** (+4.54) | 52.26±1.59 (+3.68) | 51.28±1.83 (+3.78) |

# G. Implementation Details

For the basic MML model, we follow the commonly used structure mentioned in (Zhang et al., 2023d; Xu et al., 2023) or the corresponding official code. For the model architecture of the causal representations learner, we use a three-layer Multilayer Perceptron (MLP) neural network with activation functions designed following (Clevert et al., 2016). The dimensions of the hidden vectors of each layer are specified as 64, 32, and 128. It can be embedded after the feature extractor of any MML model, ensuring the causal completeness of the learned representations by learning a learnable matrix based on Eq.10 that is consistent with the size of the representations. Moving on to the optimization process, we employ the Adam optimizer to train our model. Momentum and weight decay are set to $0.8$ and $10^{-4}$, respectively. The initial learning rate for all experiments is established at 0.1, with the flexibility for linear scaling as required. Additionally, we use grid search to set the hyperparameters $\lambda_v = 0.75$, and $\lambda_{fe} = 0.4$. Note that we specially construct corresponding ablation experiments for the selection of these parameters, as described in **Appendix H**. Experimental results show that the model can maintain relatively stable performance on various data sets using different hyperparameter settings.

For evaluation, the training dataset is randomly split as training and validation datasets, the hyperparameters are selected on the validation dataset, which maximizes the performance of the validation dataset. The overall accuracy results are evaluated on the test dataset rather than the validation dataset. All experimental procedures are executed using NVIDIA RTX A6000 GPUs, and all experimental results are obtained on the basis of five rounds of experiments.

# H. Full Results and Additional Experiments

In this section, we provide the full results and analyses of the experiments in **Section 6** and additional experiments which can only be supplemented in the appendix due to space limitations. Specifically, we first provide full results and additional details of "Performance and Robustness Analysis" (the first experiment in **Subsection 6.2** with **Table 1**), including performance on all MML baseline methods and more analysis conclusions (**Appendix H.1**). Then, we provide the additional details

Table 7: Full results (with error bars) of image-text classification performance on UPMC FOOD101 (Wang et al., 2015) and MVSA (Niu et al., 2016) with image and text. "(N, Avg.)" and "(N, Worst.)" denotes the average and worst-case accuracy. The results show the error of accuracy by executing the experiments randomly 3 times on 40 randomly selected hyperparameters. The best results are highlighted in **bold**.

| Method | FOOD 101 | | | | MVSA | | | |
|---|---|---|---|---|---|---|---|---|
| | (0,Avg.) | (0,Worst.) | (10,Avg.) | (10,Worst.) | (0,Avg.) | (0,Worst.) | (10,Avg.) | (10,Worst.) |
| CLIP (Sun et al., 2023) | 85.24±0.31 | 84.20±0.34 | 52.12±0.54 | 49.31±0.43 | 62.48±0.33 | 61.22±0.37 | 31.64±0.58 | 28.27±0.54 |
| ALIGN (Jia et al., 2021) | 86.14±0.32 | 85.00±0.33 | 53.21±0.52 | 50.85±0.47 | 63.25±0.36 | 62.69±0.32 | 30.55±0.56 | 26.44±0.59 |
| MaPLe (Khattak et al., 2023) | 90.40±0.32 | 86.28±0.37 | 53.16±0.56 | 40.21±0.47 | 77.43±0.32 | 75.36±0.30 | 43.72±0.56 | 38.82±0.52 |
| CoOp (Jia et al., 2022a) | 88.33±0.30 | 85.10±0.32 | 55.24±0.56 | 51.01±0.48 | 74.26±0.33 | 73.61±0.34 | 42.58±0.56 | 37.29±0.58 |
| VPT (Jia et al., 2022a) | 83.89±0.31 | 82.00±0.35 | 51.44±0.52 | 49.01±0.47 | 65.87±0.36 | 64.98±0.37 | 32.79±0.50 | 29.21±0.59 |
| RGB (Zhang et al., 2023c) | 83.54±0.30 | 82.43±0.37 | 54.32±0.56 | 52.32±0.45 | 69.28±0.32 | 69.12±0.31 | 51.43±0.55 | 47.26±0.54 |
| Depth (Zhang et al., 2023c) | 81.37±0.35 | 81.21±0.33 | 46.29±0.57 | 43.57±0.42 | 67.52±0.39 | 66.76±0.34 | 43.77±0.51 | 38.68±0.53 |
| Late fusion (Wang et al., 2016) | 90.69±0.30 | 90.58±0.31 | 58.00±0.52 | 55.77±0.47 | 76.88±0.33 | 74.76±0.38 | 55.16±0.56 | 47.78±0.52 |
| ConcatMML (Zhang et al., 2021) | 89.43±0.34 | 88.79±0.32 | 56.02±0.52 | 54.33±0.48 | 75.42±0.33 | 75.33±0.30 | 53.42±0.51 | 50.47±0.53 |
| AlignMML (Wang et al., 2016) | 88.26±0.30 | 88.11±0.38 | 55.47±0.57 | 52.76±0.42 | 74.91±0.36 | 72.97±0.34 | 52.71±0.55 | 47.03±0.52 |
| Bow (Zhang et al., 2023c) | 82.50±0.36 | 82.32±0.34 | 41.95±0.54 | 41.41±0.48 | 48.79±0.34 | 35.45±0.32 | 41.57±0.50 | 32.18±0.53 |
| Img (Zhang et al., 2023c) | 64.62±0.32 | 64.22±0.39 | 33.03±0.58 | 32.67±0.47 | 64.12±0.37 | 62.04±0.36 | 45.00±0.56 | 39.31±0.50 |
| BERT (Zhang et al., 2023c) | 86.46±0.37 | 86.42±0.35 | 43.88±0.58 | 43.56±0.42 | 75.61±0.30 | 74.76±0.33 | 47.41±0.55 | 45.86±0.51 |
| ConcatBow (Zhang et al., 2023c) | 70.77±0.35 | 70.68±0.34 | 35.68±0.58 | 34.92±0.41 | 64.09±0.37 | 62.04±0.35 | 45.40±0.56 | 40.95±0.54 |
| ConcatBERT (Zhang et al., 2023c) | 88.20±0.36 | 87.81±0.32 | 49.86±0.54 | 47.79±0.40 | 65.59±0.38 | 64.74±0.36 | 46.12±0.56 | 41.81±0.51 |
| MMTM (Joze et al., 2020) | 89.75±0.35 | 89.43±0.39 | 57.91±0.52 | 54.98±0.46 | 74.24±0.38 | 73.55±0.34 | 54.63±0.50 | 49.72±0.56 |
| TMC (Han et al., 2020) | 89.86±0.33 | 89.80±0.34 | 61.37±0.52 | 61.10±0.43 | 74.88±0.30 | 71.10±0.31 | 60.36±0.55 | 53.37±0.54 |
| LCKD (Wang et al., 2023b) | 85.32±0.36 | 84.26±0.34 | 47.43±0.52 | 44.22±0.43 | 62.44±0.30 | 62.27±0.34 | 43.52±0.59 | 38.63±0.54 |
| UniCODE (Xia et al., 2024) | 88.39±0.36 | 87.21±0.35 | 51.28±0.52 | 47.95±0.41 | 66.97±0.39 | 65.94±0.33 | 48.34±0.58 | 42.95±0.54 |
| SimMMDG (Dong et al., 2024) | 89.57±0.38 | 88.43±0.34 | 52.55±0.57 | 50.31±0.42 | 67.08±0.35 | 66.35±0.39 | 49.52±0.58 | 44.01±0.52 |
| MMBT (Kiela et al., 2019) | 91.52±0.37 | 91.38±0.36 | 56.75±0.55 | 56.21±0.40 | 78.50±0.34 | 78.04±0.39 | 55.35±0.52 | 52.22±0.57 |
| QMF (Zhang et al., 2023c) | 92.92±0.32 | 92.72±0.35 | 62.21±0.58 | 61.76±0.40 | 78.07±0.39 | 76.30±0.31 | 61.28±0.55 | 57.61±0.50 |
| CLIP+$C^3$R | 92.93±1.04 (+7.69) | 91.80±1.41 (+7.60) | 59.77±1.38 (+7.65) | 57.54±1.61 (+8.23) | 69.61±1.00 (+7.13) | 68.64±0.86 (+7.42) | 39.58±1.52 (+7.94) | 35.89±1.73 (+7.62) |
| ALIGN+$C^3$R | 90.91±0.95 (+4.77) | 90.13±1.21 (+5.13) | 58.74±1.78 (+5.53) | 57.96±1.78 (+7.11) | 68.71±1.51 (+5.46) | 67.21±1.19 (+4.52) | 37.26±1.25 (+6.71) | 33.60±1.48 (+7.16) |
| MaPLe+$C^3$R | 94.38±0.99 (+3.98) | 93.51±1.55 (+7.20) | 60.63±1.59 (+7.47) | 46.07±0.69 (+5.86) | 81.19±0.50 (+3.76) | 81.51±0.94 (+6.15) | 49.32±0.90 (+5.60) | 45.98±1.24 (+7.16) |
| Late fusion+$C^3$R | 94.09±0.80 (+3.40) | 92.24±0.57 (+1.66) | 65.27±1.51 (+7.27) | 59.02±0.83 (+3.25) | **83.77**±0.66 (+6.89) | 79.79±0.85 (+5.03) | 62.14±0.78 (+6.98) | 52.50±0.98 (+4.72) |
| ConcatMML+$C^3$R | 94.48±0.73 (+5.05) | 94.36±1.20 (+5.57) | 60.91±1.23 (+4.89) | 59.46±1.16 (+5.13) | 79.95±0.68 (+4.53) | 78.84±0.98 (+3.51) | 59.36±1.48 (+5.94) | 57.66±1.04 (+7.19) |
| Bow+$C^3$R | 86.61±1.29 (+4.11) | 89.31±0.52 (+6.99) | 46.44±0.87 (+4.49) | 48.62±1.06 (+7.21) | 54.89±1.00 (+6.10) | 43.16±0.83 (+7.71) | 47.93±1.07 (+6.36) | 37.67±0.71 (+5.49) |
| LCKD+$C^3$R | 90.89±1.22 (+5.57) | 90.14±1.29 (+5.88) | 54.48±1.21 (+7.05) | 51.16±0.80 (+6.94) | 66.78±0.67 (+4.34) | 65.67±0.94 (+3.40) | 49.28±1.41 (+5.76) | 42.84±1.35 (+4.21) |
| UniCODE+$C^3$R | 91.76±0.69 (+3.37) | 89.74±0.96 (+2.53) | 54.79±0.63 (+3.51) | 52.14±0.68 (+4.19) | 70.49±0.81 (+3.52) | 67.96±1.00 (+2.02) | 52.56±1.10 (+4.22) | 47.55±0.83 (+4.60) |
| SimMMDG+$C^3$R | 92.24±1.07 (+2.67) | 91.14±1.36 (+2.71) | 57.32±1.53 (+4.77) | 53.56±0.84 (+3.25) | 73.62±1.22 (+6.54) | 71.01±0.94 (+4.66) | 51.65±0.80 (+2.13) | 51.07±0.74 (+7.06) |
| MMBT+$C^3$R | 94.25±0.72 (+2.73) | 93.90±1.02 (+2.52) | 60.41±0.71 (+3.66) | 60.11±1.01 (+3.90) | 82.76±0.50 (+4.26) | **81.64**±0.68 (+3.60) | 62.12±0.81 (+6.77) | 58.93±1.30 (+6.71) |
| QMF+$C^3$R | **94.87**±1.00 (+1.95) | **93.79**±0.68 (+1.07) | **66.45**±1.63 (+4.24) | **63.69**±1.38 (+1.93) | 83.13±0.94 (+5.06) | 81.98±0.94 (+5.68) | **66.66**±0.81 (+5.38) | **64.51**±1.08 (+6.90) |

and analysis of "When faces the problem of missing modalities" (the second experiment in **Subsection 6.2** with **Table 2**). Next, we provide the full results and more analysis of "Learning causal complete representations" (the third experiment in **Subsection 6.2** with **Figure 3**), including the detailed settings, the visualization and more analysis of the correlation between different methods and different causal causes under different spurious degrees. Finally, we provide the full results and additional experiments of the ablation study, including the experiments about model efficiency, trade-off performance, and parameter sensitivity.

## H.1. Full Results and Additional Details of Performance and Robustness Analysis

Due to space limitations, we provide part of the experimental results in **Table 1** of the main text, which contains typical methods of all categories of baselines mentioned in **Appendix F**. In **Table 6 and Table 7**, we provide comparison baselines for all baselines. Specifically, Table 6 provides the experiments about scene recognition on NYU Depth V2 (Silberman et al., 2012) and SUN RGBD (Song et al., 2015) datasets. **Table 7** provides the experiments about image-text classification on UPMC FOOD101 (Wang et al., 2015) and MVSA (Niu et al., 2016) datasets. From the results, we can observe that (i) $C^3$R achieves stable improvements in both the average and worst-case accuracy on almost all the comparison baselines, including both the foundation model and all types of MML baselines; and (ii) regardless of the data scale, $C^3$R can bring obvious and stable performance improvements, with an average increase of more than 5%.. This proves the superior effect and robustness of $C^3$R.

## H.2. Full Results and Additional Details of Performance with Missing Modalities

Due to the complexity of data in real systems, MML methods face the dilemma of missing modalities (Zhao et al., 2021; Zhang et al., 2022a). This problem severely affects model performance, especially when important modes are missing. Therefore, in order for the model to have good practicality, it is very critical to be able to maintain stable performance in the face of missing modes. To evaluate the performance of the proposed $C^3$R when facing missing modalities, we constructed

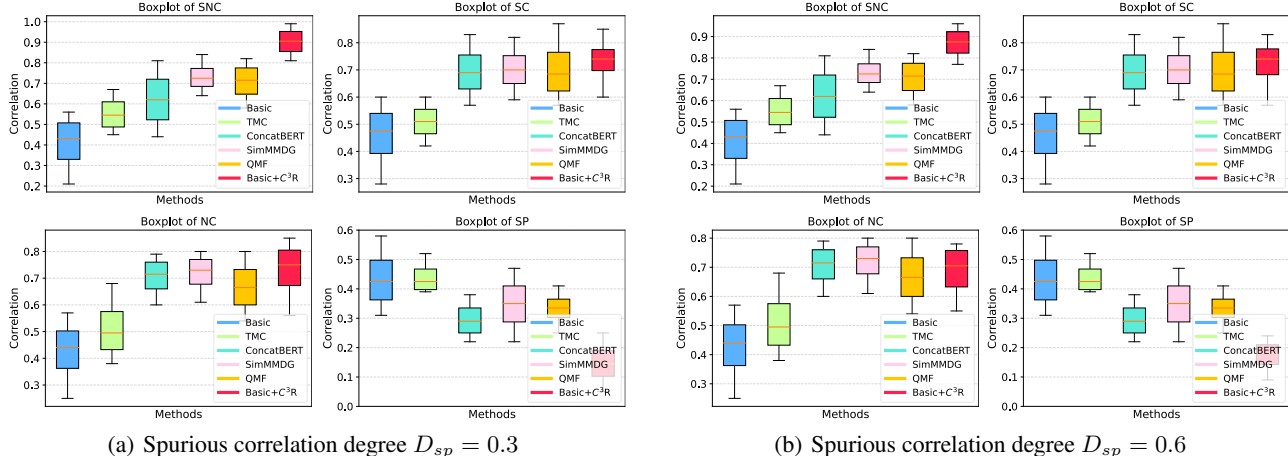

Figure 6: Evaluation for the property of learned representations (identification for SNC, SC, NC, and SP) with different spurious correlation degree $D_{sp}$.

comparative experiments on all 15 possible combinations of missing modalities on BraTS. Specifically, based on BraTS, we construct fifteen combinations of different modes following (Wang et al., 2023b). Then, we recorded the performance of $C^3$R and several strong baseline models after five runs.

From the results illustrated in **Table 2**, we can observe that (i) using F1 and T1c, the model performs much better than others on Enhancing Tumour. Likewise, T1c and Flair from the Tumour Core and Whole Tumour contributed the most. (ii) The model has been significantly improved after the introduction of $C^3$R, especially when important modes are missing, e.g., F1 on Enhancing Tumour. The results prove that our method can improve the performance of the model in the scenario of missing modality and improve the stability.

### H.3. Full Results and Additional Details of Causal Complete Evaluation

In order to verify whether our method $C^3$R actually extracts causal necessary and sufficient representations, we construct a synthetic data set called MMLSynData (as mentioned in **Appendix D.5**) to conduct comparative experiments. Specifically, we first built MMLSynData, which is a synthetic data set for MML scenarios containing four types of data, i.e., sufficient and necessary causes (SNC), sufficient but unnecessary causes (SC), necessary but insufficient causes (NC), and spurious correlations (SP). Each category contains 250 sets of training data and 50 sets of test data. Next, based on the above 1200 sets of data, we set different degrees of spurious correlation $D_{sp}$ for comparative experiments, including $D_{sp} = 0.3$ and $D_{sp} = 0.6$. It is worth noting that the result shown in **Figure 3** in the text is $D_{sp} = 0.6$. Then, we choose SOTA and classic MML methods to compare with the basic MML framework after the introduction of $C^3$R, where $basic$ represents a simple MML learning framework based on the Conv4 backbone network and a classifier following (Wang et al., 2016). We record their correlation with four different types of data in MMLSynData.

The results of $D_{sp} = 0.3$ and $D_{sp} = 0.6$ are shown in **Figure 6**. Combined with **Figure 3**, the results show that compared with other methods, the correlation with real data (such as SN, SF, and NC) is higher after the introduction of $C^3$R, and the correlation with spurious correlation (SP) has a lower score. For example, when $D_{sp} = 0.3$, we obtain the average distance correlations of SNC, SC, NC, and SP as 0.91, 0.71, 0.69, 0.13 respectively. Furthermore, when we set $D_{sp}$ to a larger value of 0.6, the distance correlation between the MML method and false information is almost unchanged after introducing $C^3$R, while other methods are difficult to achieve this. The results show that when there are more spurious correlations in the data, $C^3$R still tends to capture valid information from the real data to extract sufficient and necessary causes.

### H.4. Full Results and Additional Details of Ablation Study

To evaluate the effect of the model and understand how $C^3$R works well, we constructed a series of experiments, including (i) the effect of each item in the $C^3$R objective as shown in **Subsection 6.2**; (ii) trade-off performance about the model efficiency and accuracy after introducing $C^3$R; (iii) parameter sensitivity about the three hyperparameters in the $C^3$R

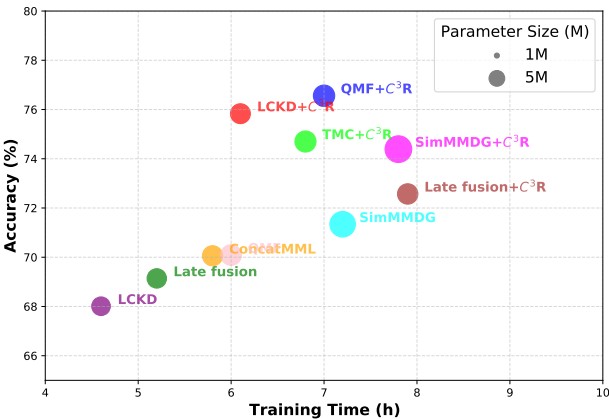

Figure 7: Trade-off Performance of different methods on (0,Avg.) NYU Depth V2.

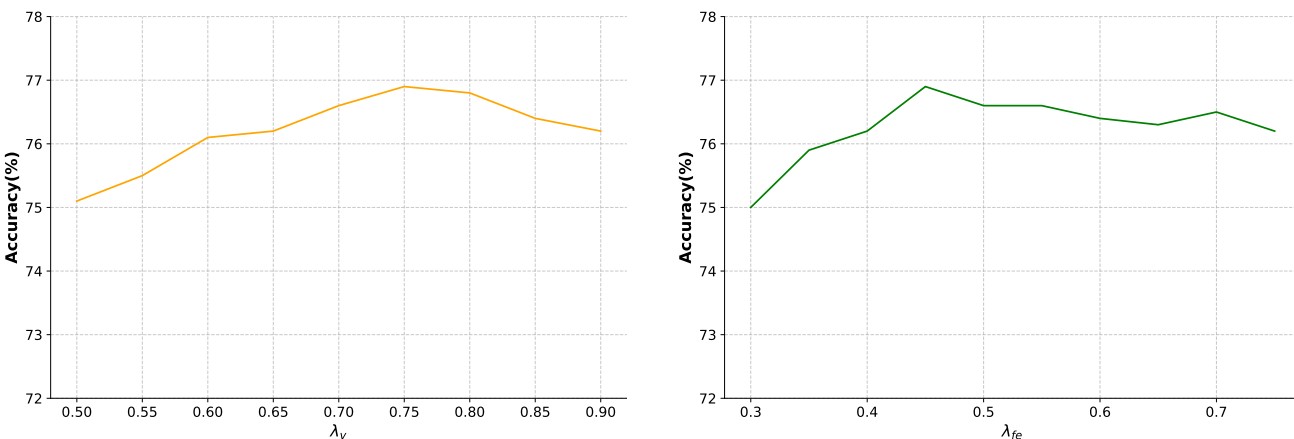

Figure 8: Parameter sensitivity about $\lambda_v$ on (0,Avg.) NYU Depth V2.

Figure 9: Parameter sensitivity about $\lambda_{fe}$ on (0,Avg.) NYU Depth V2.

objective, i.e., $\lambda_v$ and $\lambda_{fe}$. In this section, we provide details about the latter two experiments and results, where the results of the first experiment are shown in **Figure 4**.

**Model efficiency**   Since $C^3$R is a plug-and-play module, in order to ensure its practicability, we explore the balance between model performance and efficiency. We compare the trade-off performance of multiple baselines before and after using our $C^3$R with the same Conv4 backbone. The results illustrated in **Figure 7** show that introducing $C^3$R achieves great performance with acceptable computational cost.

**Parameter sensitivity**   We determine the hyperparameters of the regularization term in the experiment based on the performance of the validation samples. Specifically, for each experimental scenario, we test the impact of different values of $\lambda_v$ and $\lambda_{fe}$ on model performance. The range of these values is set between $[0.3, 0.9]$. In each scenario, we first use grid search to screen the parameters with a difference of $0.05$. After screening the optimal interval, we screened the parameters with a difference of $0.01$ and recorded the final average results.

The results are shown in **Figures 8-9**. From the results, we observe that when $\lambda_v = 0.75$, and $\lambda_{fe} = 0.4$, the results are better which is also our choice. Meanwhile, the model effect does not change significantly under different parameters, which illustrates the stability of the parameters and the convenience of adjustment in reality.

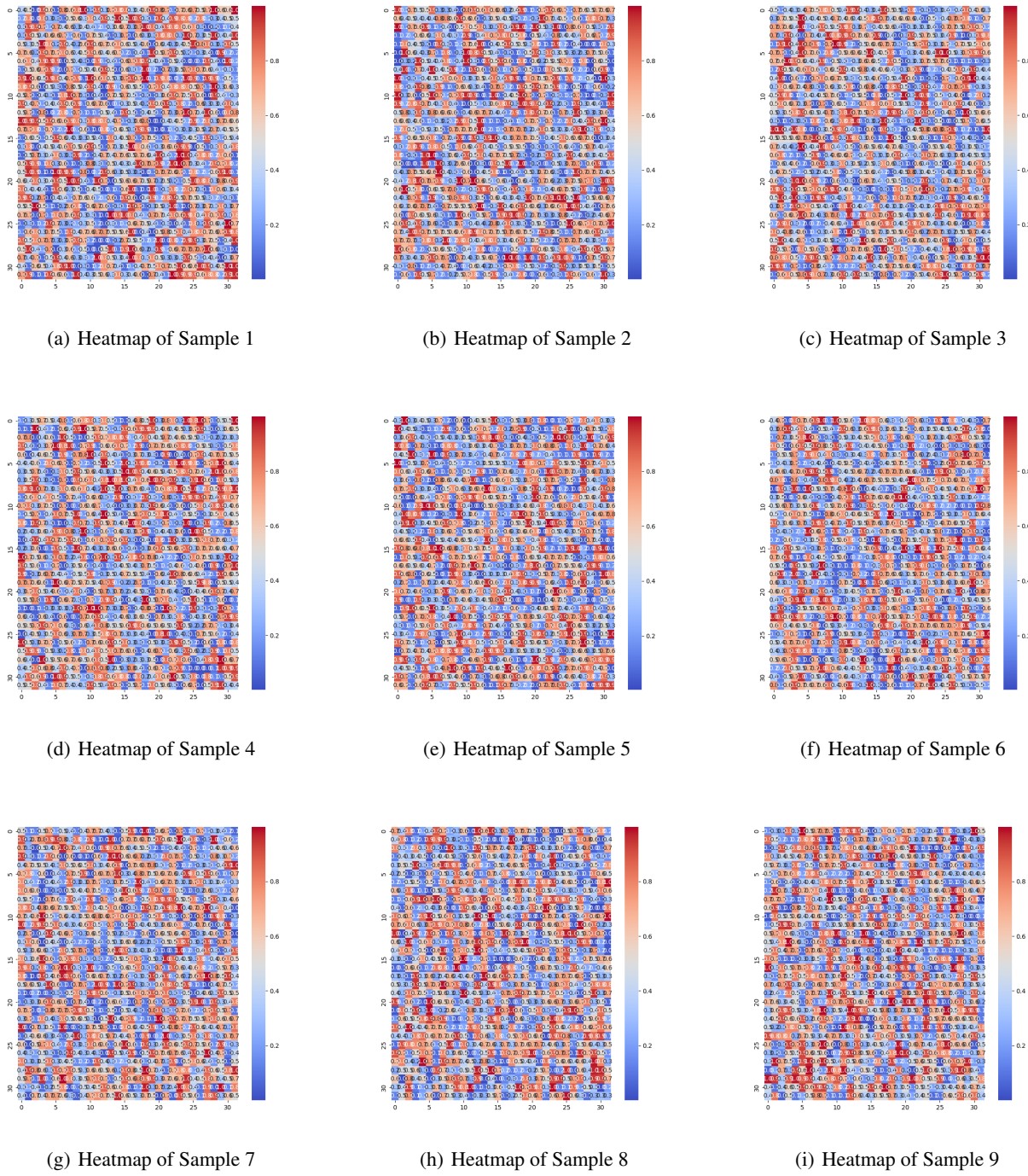

(a) Heatmap of Sample 1

(b) Heatmap of Sample 2

(c) Heatmap of Sample 3

(d) Heatmap of Sample 4

(e) Heatmap of Sample 5

(f) Heatmap of Sample 6

(g) Heatmap of Sample 7

(h) Heatmap of Sample 8

(i) Heatmap of Sample 9

Figure 10: Nine sets of visualization results on the BraTS dataset. For qualitative demonstration, we used the model after five iterations to calculate the importance weights of the resized $(32 \times 32)$ sampled data and obtained the above heat map. In the heat map, the score of each area represents the probability that it belongs to a causal complete cause.

Table 8: Effect of the extracted representation. This experiment aims to analyze the practical guidance of $C^3R$ and how this representation mirrors the properties in the example. The "Original" means that the representation is directly from the extractor of the method; "Reduce" means we reduce the weights of high-weight elements which are more likely to be causal complete and directly input them into the frozen classification head; "Increase" means we increase the weights of low-weight representation elements to twice the original weights and normalize them.

| Method | Representation | Accuracy(%) |
|---|---|---|
| LCKD | Original | 79.30 |
| LCKD+$C^3R$ | Original | 85.39 |
| LCKD+$C^3R$ | Reduce | 83.73 |
| LCKD+$C^3R$ | Increase | 84.02 |

Table 9: Performance when faces noise on NYU Depth V2. We follow the same experimental setting in Section 6.2. We apply a mask to 30% of the data area. Then, we selected MMBT and QMF as the baselines to calculate the performance changes before and after the introduction of $C^3R$.

| Method | Accuracy(%) |
|---|---|
| MMBT | 62.89 |
| QMF | 64.72 |
| MMBT+$C^3R$ | 71.26 |
| QMF+$C^3R$ | 75.49 |

## H.5. Visualization of Causal Complete Cause

In addition to the examples of causal completeness provided in **Figure 1** and **Appendix D**, we construct more qualitative demonstrations for real datasets. Specifically, for the five datasets involved in the experiment, we randomly sample a set of training samples from each dataset and visualize, i.e., calculate the importance heatmap of sufficient and necessary causes based on the constructed $C^3R$. The results are shown in **Figure 10**.

Besides this, we conduct a series of visualization experiments to further elucidate the practical guidance provided by $C^3R$ and to examine how this representation reflects the underlying properties in the example. Utilizing the BraTS dataset as a case study, we consider four modalities, namely F1, T1, T1c, and T2, and adopt LCKD as the baseline method. The results are presented in **Table 8**. Specifically, we first randomly select five sets of multi-modal data and input them into LCKD to obtain both the results and the representations with the weight matrix. Subsequently, we employ the same settings to visualize the feature representations after integrating $C^3R$, observing that the weights are altered and the accuracy is significantly enhanced. We hypothesize that this improvement arises because $C^3R$ calculates the probability of representations belonging to causal factors and subsequently re-weights the matrix. To further validate this hypothesis, we reduce the weights of high-weight elements and directly input them into the frozen classification head, resulting in a decrease in accuracy. Conversely, when we amplify the weights of low-weight representation elements to twice their original values and normalize them, the accuracy also declines. Thus, this example demonstrates how $C^3R$ effectively guides the representation in these datasets, where elements with high weights correspond to the causal factors of the selected data.

## H.6. Performance When Facing Noise

One reason of why we conduct experiments of "Performance and robustness analysis" is to consider the impact of data noise and damage. The difference between the "worst result" and the "average result" under the same setting may be caused by noises. The results in **Table 1**, **Table 6**, and **Table 7** show that $C^3R$ steadily improves the effect of MML and greatly reduce the gap between the worst and average results. Meanwhile, we also considered more serious data defects such as missing modality, and the results (**Table 2**) reached similar conclusions. These results prove that (i) $C^3R$ is effective and robust, and (ii) noise will damage multi-modal representations but hard to affect causal representations learned by $C^3R$.

To comprehensively evaluate the effect of $C^3R$ in the presence of noise, we further provide a toy experiment to intuitively evaluate the impact of noise on $C^3R$. Specifically, we choose NYU Depth V2 as the benchmark dataset and apply a mask to

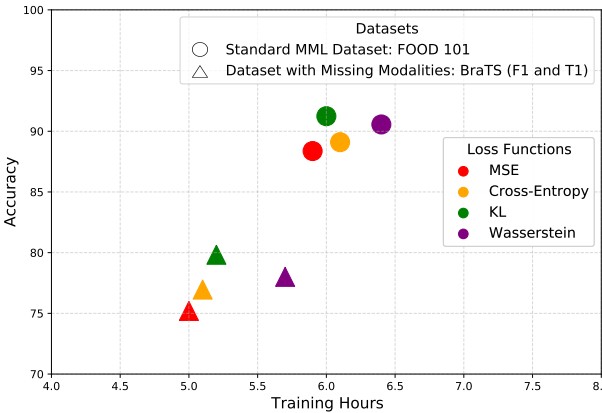

Figure 11: Trade-off Performance of with different distance loss functions.

30% of the data area. Then, we select MMBT and QMF as the baselines to calculate the performance changes of the models before and after the introduction of $C^3R$. The results are shown in **Table 9**. The results show that (1) noises affects the performance of the MML baselines with the degradation exceeding 5%; (2) even in the presence of noises, the performance of MML after the introduction of $C^3R$ only decreases by less than 2% compared with the original (**Table 1**), needless to say that is may be due to the mask blocking important feature areas. This further shows that $C^3R$ has good robustness.

### H.7. Role of Distance Loss

In this subsection, we examine the role of loss functions for distance in the objective of $C^3\text{R}$. The goal of this optimization is to evaluate the distance between $\widehat{Z}_{c,i}$ and $\overline{Z}_{c,i}$, with the choice of loss function directly affecting performance. We evaluate four common distance loss functions—Mean Squared Error (MSE) (Tsai et al., 2020), Cross-Entropy (De Boer et al., 2005), KL Divergence (Hershey & Olsen, 2007), and Wasserstein Distance (Panaretos & Zemel, 2019)—by analyzing their performance and training time on FOOD101. The loss functions are defined as follows:

**Mean Squared Error (MSE)** (Tsai et al., 2020) measures the average squared differences between predicted and true values across modalities. It is simple to compute but sensitive to differences between modalities:

$$\text{MSE}(\mathbf{y}, \hat{\mathbf{y}}) = \frac{1}{n} \sum_{i=1}^{n} \|\mathbf{y}_i - \hat{\mathbf{y}}_i\|^2,$$

where $\mathbf{y}$ and $\hat{\mathbf{y}}$ are the true and predicted values across modalities, $\|\cdot\|$ is the Euclidean distance, and $n$ is the sample size.

**Cross-Entropy** (De Boer et al., 2005) calculates the difference between true and predicted joint distributions, capturing uncertainty between modalities:

$$\text{CE}(\mathbf{y}, \hat{\mathbf{y}}) = -\sum_{i=1}^{n} \sum_{j=1}^{m} y_{ij} \log \hat{y}_{ij},$$

where $\mathbf{y}$ and $\hat{\mathbf{y}}$ denote the true and predicted joint probability distributions, and $m$ is the number of modalities.

**KL Divergence (Kullback-Leibler Divergence)** (Hershey & Olsen, 2007) measures the difference between two joint probability distributions, offering a balance between accuracy and computational cost:

$$\text{KL}(P\|Q) = \sum_{i} \sum_{j=1}^{m} P(i,j) \log \frac{P(i,j)}{Q(i,j)},$$

where $P$ and $Q$ are the true and predicted joint distributions.

**Wasserstein Distance** (Panaretos & Zemel, 2019) reflects the geometric distance between joint distributions but is computationally expensive:

$$\text{WD}(P,Q) = \inf_{\gamma \in \Pi(P,Q)} \mathbb{E}_{(X,Y) \sim \gamma}[\|X - Y\|],$$

where $P$ and $Q$ are distributions, $\Pi(P, Q)$ represents the set of joint distributions coupling $P$ and $Q$.

From empirical analysis, KL divergence achieves the best trade-off between accuracy and efficiency, as shown in **Figure 11**. We evaluate the effect of LCKD+$C^3$R by introducing different distance metrics. We select the standard MML benchmark dataset, FOOD 101, and dataset with missing modality, i.e., tumour core on BraTS (with F1 and T1 available), for evaluation. The results show that it provides higher accuracy in shorter training times compared to MSE and cross-entropy, while Wasserstein distance achieves comparable accuracy, its computational cost is significantly higher, making KL divergence the more practical choice.

From a theoretical perspective, to better use $C^3$R in MML models, the distance metric used in Theorem 3.5 should include: (i) the ability to capture subtle differences between distributions accurately, (ii) utility in ensuring stable and efficient convergence to the global optimum during optimization, (iii) applicability to a wide range of complex distributions, and (iv) computational efficiency. KL divergence excels in these aspects, as evidenced by three key features (Hershey & Olsen, 2007; Goldberger et al., 2003; Shlens, 2014). First, KL divergence is non-negative and equals zero only when two distributions are identical, aligning with intuitive notions of difference (Gong et al., 2021). This property ensures its stability in capturing subtle differences, fulfilling criteria (i) and (iv). Second, KL divergence is convex, increasing the likelihood of convergence to the global optimum rather than becoming trapped in local minima, especially in high-dimensional settings (Hershey & Olsen, 2007). This characteristic addresses criterion (ii). Finally, as an extension of information entropy, KL divergence effectively quantifies information loss and uncertainty, making it highly versatile for diverse applications (Goldberger et al., 2003), including self-rewarding learning tasks, thereby satisfying criterion (iii).

In contrast, alternative metrics exhibit significant limitations. MSE, rooted in Euclidean distance, is overly sensitive to outliers and disregards the non-negativity and normalization properties of probability distributions (Marmolin, 1986; Chicco et al., 2021; Lebanon, 2010), failing to meet criteria (i) and (iii). Cross-entropy, a specific case of KL divergence, struggles with continuous distributions and non-one-hot true distributions (De Boer et al., 2005; Botev et al., 2013), limiting its ability to precisely measure complex distributions (i and iii). While Wasserstein distance captures overall shape differences between distributions, its high computational cost and dependence on smoothness conditions make it unsuitable for high-dimensional problems (Panaretos & Zemel, 2019; Vallender, 1974), falling short of criterion (iv).

In summary, KL divergence strikes an optimal balance between theoretical robustness and computational feasibility, adhering to the properties for twin network. This balance enables better model generalization and reduced training costs, making it the preferred metric for practical applications.

