# OpenReview forum: "Towards the Causal Complete Cause of Multi-Modal Representation Learning"
_ICML.cc/2025/Conference — ICML 2025 poster_

### Official Review · Reviewer_eaJB · 2025-03-07

**Overall Recommendation:** 4

**Summary:**

The paper explores causal completeness in multi-modal representation learning, addressing issues where existing methods may capture unnecessary or insufficient information. It introduces the Causal Complete Cause (C3) framework, which ensures learned representations are both sufficient (contain all necessary information) and necessary (exclude irrelevant details). The authors propose a twin-network approach using instrumental variables and counterfactual modeling to estimate and enforce C3, leading to a new regularization method (C3R). Experimental results show that this approach improves robustness and accuracy, especially in scenarios with spurious correlations and missing modalities

**Claims And Evidence:**

Yes.

**Essential References Not Discussed:**

I can not certainly comment on this because I am not aware of the literature in multi-modal representation learning.

**Experimental Designs Or Analyses:**

Yes.

**Methods And Evaluation Criteria:**

Yes.

**Other Comments Or Suggestions:**

> Typos:

1. Line 78: Looks like there is a grammatical error in the sentence.
2. Line 156: Shouldn't it be "we construct a structural causal model"?
3. Lin2 121 (right): Should be "an SCM".

**Other Strengths And Weaknesses:**

> Strengths
1. The paper is written well and easy to understand.
2. Even though the idea of causal sufficiency and necessity is borrowed from literature, applying it to multi-model learning shows.

> Weaknesses
1. In theorem 3.2, the the term obtained after identification still contains interventional terms. This is in contrast to traditional identifiability where the final result is statistical estimand instead of an estimand containing interventional terms. It may be good to use another word instead of "identifiability".
2. Because the proposed method is used to improve the existing methods as shown in Table 1, what is the additional computational cost of the proposed method adds to the existing methods?
3. In real-world data, there is a high chance that hidden confounding exists between causal and spurious feature, how does the proposed method handle such scenarios?

> Minor issue:
1. It is crucial to highlight the difference between causal sufficiency assumption used in causality literature to avoid confusion.
2. How are the ideas related to the paper: https://arxiv.org/abs/2109.03795 ? The ideas of causal sufficiency and necessity of representations has been studied in the paper. It is crucial to discuss this paper in related works.

**Questions For Authors:**

Please see weaknesses above. Addressing those points is crucial for strengthening the paper.

**Relation To Broader Scientific Literature:**

I am not aware of the literature in multi-modal representation learning.

**Theoretical Claims:**

I only checked proof of theorem 3.2.

---

> ### Author Rebuttal · Authors · 2025-03-31
>
> We sincerely appreciate Reviewer eaJB's constructive feedback and the time and effort dedicated to the review process. We are also grateful for the recognition of our work and sincerely hope the following responses can eliminate the concerns.
> ## Response to W1
> We appreciate the suggestions and apologize for any misunderstandings. To distinguish it from traditional identifiability, we change the word to "Causal Identifiability." According to Pearl (2009) and Lee & Bareinboim (2020), causal identifiability refers to establishing the effect of intervening on a set of variables on another set, using observational or interventional distributions under given assumptions. It permits the inclusion of interventional terms if they are uniquely determined from the observational distribution (Shpitser, 2012). Based on these, Theorem 3.2 is derived. When the model satisfies local invertibility, we can uniquely recover the variable distribution from the observed data, thereby ensuring causal identifiability and estimation of $C^3$.
> ## Response to W2
> The experiments on computational cost in Appendix H.4 (Figure 7) indicate that introducing $C^3$R increases the computational cost by less than 1.3x compared to the original.
> ## Response to W3
> We provide an outline to explain how it "handles hidden confounding":
> - Theoretically: The "hidden confounding" issues correspond to the relaxation of the exogeneity assumption, which requires satisfying causal identifiability of $C^3$ in the environment that contains hidden confoundings. To achieve the relaxation, we introduce an instrumental variable $V$ to eliminate confounding effects, thus achieving causal identifiability regardless of whether hidden confounding exists (Section 3.2). We model $V$ using the improved self-attention mechanism, which employs alignment scores to capture causal factors $F_c$, theoretically constrain $Z$  to only contain $F_c$, and eliminate confounding effects (L220-257 and Appendix A.3).
> - Methodology: Based on theoretical results, we constrain the causal completeness of the representations with $C^3$ risk. The concept of causal completeness also means without "hidden confounding" in causality (Pearl, 2009). We propose $C^3$R to constrain it by reducing $C^3$ risk. It is achieved through a twin network: the real-world branch uses $V$ to adjust $Z$ so that it only contains $F_c$ for accurate prediction, achieving causal sufficiency and eliminating hidden confounding; the hypothetical branch employs gradient-based counterfactual modeling to further calibrate the causal factors, constraining causal necessity and further filtering out hidden confounding (L258-307 and Appendix D.4). Table 1-8 prove its effectiveness.
> ## Response to Minor 1
> We sincerely appreciate the suggestions and would like to provide an outline for illustration (further emphasized in Section 3.1):
> - Meaning of Causal Sufficiency: Our concept of causal sufficiency (Definition 3.1) is adapted from Definition 9.2.2 in Pearl (2009), i.e., “setting x would produce y in a situation where x and y are in fact absent” which is considered the original definition of causal sufficiency. It also aligns with another type of illustration in causality literature, e.g., “the absence of latent confounder”, as we utilize instrumental variable and twin network in Sections 3.3 and 4 to eliminate confounders, aiming to satisfy causal sufficiency.
> - Clarification on Assumptions: Previous works rely on an exogeneity assumption, assuming there are no confounders in environments. In contrast, we relax these assumptions to account for possible confounders in practice (L80–99 and Appendix D.3). It does not alter the meaning of causal sufficiency; it means that we do not assume that there is no confounding in the environment, but introduce instrumental variables to eliminate the impact of confounding to relax the assumption.
> ## Response to Minor 2
> We sincerely appreciate the suggestions and have carefully reviewed the mentioned paper. Below is a brief illustration of the differences (cited and supplemented in Section 7). Wang & Jordan aim to construct measures of non-spuriousness and disentanglement. Their exploitation of causal necessity and sufficiency concept is to align it with non-spuriousness and “invoke” the corresponding measure in Pearl (2009) to construct the measure of non-spuriousness. We focus on modeling the concept of causal sufficiency and necessity itself in MML. We relax the exogeneity and monotonicity assumptions that previous works depend on, including Wang & Jordan, and propose a new method to measure and constrain MML-specific causal completeness. Thus, although both works draw inspiration from Pearl (2009), they differ significantly in problem, theoretical framework, methodology, and experimental validation.
> ## Regarding Comments
> We appreciate the reviewer's suggestions and have polished the text accordingly, i.e., changing “with both” to “be both”, “conduct” to “construct”, and “a” to “an”.

---

> > ### Comment · Reviewer_eaJB · 2025-04-03
> >
> > I thank the authors for the response. I've read the responses and I will keep my positive score.

---

> > > ### Author Response · Authors · 2025-04-03
> > >
> > > Dear Reviewer eaJB,
> > >
> > > We sincerely appreciate your feedback which has greatly encouraged us. We would like to express our gratitude again for the time and effort you have dedicated to reviewing it, which helped us improve our work further.
> > >
> > > Best regards,
> > > The Authors

---

### Official Review · Reviewer_fkSa · 2025-03-11

**Overall Recommendation:** 4

**Summary:**

This paper addresses the problem of multi-modal representation learning from a causal perspective. It analyzes the insufficiency and redundancy of information across multiple modalities. The authors propose a novel concept termed Causal Complete Cause ($C^3$), supported by identifiability guarantees under weaker assumptions (non-exogeneity and non-monotonicity). A twin network is introduced to estimate the $C^3$ measurement, and extensive experiments are conducted to validate the proposed method.

**Claims And Evidence:**

Yes. This paper is mainly discussing the definition (explained in Section 2 & 3.1), identifiability (explained in Section 3.2), and measurement (explained in Section 3.3) of the novel causal complete cause ($C^3$).

**Essential References Not Discussed:**

In general, this paper has covered most related work. It would be more comprehensive if also comparing and discussing the following disentangled/causal representation learning papers, to name a few:

- Schölkopf et al. "Toward causal representation learning", Proceddings of IEEE, 2021.

- Brehmer et al. "Weakly supervised causal representation learning", NeurIPS, 2022.

- Ahuja et al. "Interventional causal representation learning", ICML, 2023.

- Yao et al. "Multi-view causal representation learning with partial observability", ICLR, 2024.

- Sun et al. "Causal representation learning from multimodal biological observations", ICLR 2025.

**Experimental Designs Or Analyses:**

Yes. The experimental design is sound and valid, the experimental evaluation is comprehensive, covering 17 baseline methods and 6 datasets across various tasks including scene recognition, image-text classification, and segmentation.

**Methods And Evaluation Criteria:**

Yes, the proposed method is the causal complete cause ($C^3$), which makes sense for multi-modal representation learning, and the evaluation criteria is mainly the classification accuracy, which is also reasonable.

**Other Comments Or Suggestions:**

There is no obvious typo so far.

**Other Strengths And Weaknesses:**

**Strengths:**

- This paper is well-written and clearly-organized.

- The $C^3$ concept is novel and intriguing. The authors provide a new measurement backed by identifiability guarantees, along with a thorough analysis of sufficiency and necessity.

- The experimental evaluation is comprehensive, covering 17 baseline methods and 6 datasets across various tasks including scene recognition, image-text classification, and segmentation.

**Weaknesses and Comments:**

- The example in Figure 1 remains somewhat ambiguous. The distinction between sufficient and necessary features in the image domain is unclear. Including an additional example that explicitly illustrates features that are both sufficient and necessary would be helpful. Additionally, the figure should more clearly reflect the elements of multi-modality.
- In Figure 2, both anti-causal (Y → X) and causal (X → Y) directions are discussed. However, in the absence of spurious correlations, either direction could represent the true data-generating process. The comparison between an anti-causal mechanism (true, without spurious correlation) and a causal mechanism (learned, with spurious correlation) may be misleading.
- Terminology clarification: The term causal sufficiency in Section 3 is typically used in causal inference to mean the absence of latent confounders. It would be clearer if the authors explicitly clarify their intended meaning in the context of this paper.
- In Theorem 3.2, the terms local invertibility and non-exogeneity are not clearly defined. Providing a brief explanation or formal definition immediately following the theorem would improve clarity.
- The derivations of Equations (3) and (4) are not clearly explained. A more detailed explanation or derivation would help the reader better understand the methodology.
- What is the intuition of choosing self-attention mechanism particularly for generating instrumental variable V?

**Questions For Authors:**

- Can the learned representations be mapped to interpretable latent concepts, or are they inherently abstract?
- How does the method perform in domains with real-world data complexities, such as missing data, or measurement error?

**Relation To Broader Scientific Literature:**

This paper presented the definition, identifiability, and measurement of $C^3$ with theoretical support, without exogeneity and monotonicity assumptions. This is a relaxed and quantifiable framework, compared to previous work.

**Theoretical Claims:**

Yes, the key theoretical claims that I checked include Theorem 3.2 (Identifiability under Non-Exogeneity), Theorem 3.3 (Identifiability under Non-Monotonicity and Non-Exogeneity), and Theorem 3.4 (Modeling Instrumental Variable V in MML).

- In Theorem 3.2, what do you mean by "local invertibility"?

- how to interpret that non-monotonicity in Theorem 3.3, compared to the monotonicity requirement Theorem 3.2? What is the key modification such that monotonicity is not required anymore.

- In Theorem 3.4, what is the intuition of choosing self-attention mechanism, instead of other methods?

---

> ### Author Rebuttal · Authors · 2025-03-31
>
> We sincerely appreciate the reviewer fkSa's constructive feedback and the time and effort dedicated to the review. We are grateful for the recognition of our work and sincerely hope the following responses can eliminate the concerns.
> ## Response to W1
> We sincerely appreciate the suggestions and have refined Fig.1 accordingly. The examples in Fig.1 are based on specific tasks and data conditions (L129-155). A sufficient and necessary feature can be "flat duck bill" as its presence indicates "duck" and every "duck" sample includes it. We also added the textual modality, e.g., "A duck has a flat bill and orange webbed feet standing with its wings folded" as suggested for better clarification.
> ## Response to W2
> We agree “in the absence...” but would like to kindly clarify that the SCMs in Fig.2 are conducted under different settings, aiming to illustrate potential confounding issues in MML instead of true data-generating direction (L129-155). Left is for how MML sample $X$ is generated based on causal generating mechanism; Right is to align with the practical MML process, where factors are typically coupled for predicting $Y$. We apologize for any ambiguity that may caused by the caption and have refined it accordingly.
> ## Response to W3
> We sincerely appreciate the suggestions and would like to clarify that our concept of causal sufficiency (Definition 3.1) is adapted from Definition 9.2.2 in Pearl (2009), i.e., “setting x would produce y where x and y are in fact absent”. It aligns with the mentioned “the absence of latent confounder”, as we utilize instrumental variable and twin network to eliminate confounders (Sections 3.3 and 4), aiming to satisfy causal sufficiency. We further emphasized it in Section 3.1 according to the valuable suggestion.
> ## Response to W4, Theoretical Claims 1&2
> - We have illustrated “local invertibility” in Proposition D.3 and will further elaborate on it immediately following Theorem 3.2 as suggested. Briefly, it states that the model can uniquely recover the distribution of exogenous variable $s$ from the conditional distribution of $Y$ given its parents $Pa(Y)$.
> - We have provided a brief explanation about non-exogeneity, e.g., $P(Y_{do(Z=c)}) \neq P(Y \mid Z=c)$ in L185-191 with detailed in Appendix D.3. We will further refine this immediately following the theorem.
> - Compared to Theorem 3.2, the non-monotonicity in Theorem 3.3 allows the effect of $Z$ on $Y$ to vary in direction or intensity. The key modification is introducing $V$ for piecewise estimation through integration (Eq.5).
> ## Response to W5
> We sincerely appreciate the suggestions and provide a brief derivation following Theorem 9.2.15 in (Pearl, 2009): $C^3(Z)$ can be decomposed into (i) when $Z\neq c$ occurs with probability $1-P(Z=c)$, its contribution reflects $C^3_{su}(Z)$ for $Z=c$; (ii) when $Z=c$ occurs with $P(Z=c)$, its contribution reflects $C^3_{ne}(Z)$ of $Z=\bar{c}$ on $Y$. By normalizing, we get $ C^3_{su}(Z) = \frac{C^3(Z)}{1-P(Z=c)}$ and $C^3_{ne}(Z) = \frac{C^3(Z)}{P(Z=c)}$ as Eq.3 and Eq.4. We will add it in Appendix A.1 for better understanding.
> ## Response to W6 & Theoretical Claim 3
> The intuition is twofold:
> - Aligning with the modeling objectives of $V$ (L220–248): The self-attention mechanism, dynamically measures the alignment score between modalities, can emphasize the important features for $Y$ across modalities (i.e., achieves higher scores on $F_c$) while downplaying $F_s$. Intuitively, using self-attention-based $V$ on $Z$ with distance penalties satisfies the goal that "constrain $Z$ only contains $F_c$".
> - Higher efficiency: Although custom-designed networks can also achieve the above goal, the self-attention mechanism is typically more lightweight without substantially increasing model complexity.
> ## Response to Q1
> The learned representations can "be mapped to interpretable latent concepts". Such a representation must both fully explain the decision (sufficiency) and be indispensable (necessity), revealing the most important factors behind predictions. For example, a causally complete representation captures all critical factors for classifying "happy", e.g., visual (upturned mouth corners, wide-open eyes) and audio cues (high pitch) where removing "upturned mouth corners" may result in "angry".
> ## Response to Q2
> The results in Table 2 (missing modality) and Table 8 (data noise) show that $C^3$R achieves stable performance improvements in domains with real-world data complexities.
> ## Regarding Essential References Not Discussed
> We appreciate the suggestion and carefully reviewed the papers (cited and supplemented in Section 7). Schölkopf et al. present a review for causal inference; the rest four mainly focus on identifiability under different settings and problems, e.g., weak supervision and partial observability. We focus on the causal sufficiency and necessity concept, aiming to explore and model it for MML, where the problem, theory, method, and experiments are all different.

---

### Official Review · Reviewer_ZspQ · 2025-03-13

**Overall Recommendation:** 3

**Summary:**

This paper proposes Causal Complete Cause Regularization (C³R) Risk, a metric that quantifies the likelihood that a learned representation is causally complete. A lower C³ Risk indicates that the representation satisfies both causal sufficiency and causal necessity, meaning that spurious correlations have been removed, and all essential causal factors are retained.

In conventional Multi-Modal Learning (MML) methods, two major issues arise. 1) Spurious correlations exist in sufficiency evaluation, leading to unreliable representations. To address this, the paper introduces Twin Network’s real-world branch, which leverages a self-attention-based instrumental variable to ensure that representations do not rely on non-causal factors, effectively mitigating spurious correlations. 2) For necessity evaluation, the challenge is that it requires observing label changes when a representation is absent, which is impossible in real-world data.

The proposed method overcomes this limitation using Gradient-Based Counterfactual Modeling in Twin Network’s hypothetical-world branch. Since directly removing a representation is not feasible, the model instead adjusts gradients in the loss function to guide the representation toward a counterfactual direction, generating an approximation of counterfactual representations.

**Claims And Evidence:**

Table 1 and Table 2 demonstrate the effectiveness of applying C³ Risk to various benchmark datasets, including NYU Depth V2, SUN RGBD, FOOD 101, MVSA, and BraTS. The results show that incorporating C³R improves performance across scene recognition, image-text classification, and segmentation tasks, particularly by enhancing both average and worst-case accuracy, thereby increasing model robustness.

Additionally, Figure 4's Ablation Study experimentally verifies the contribution of each component in the proposed loss function. The study confirms that these components play a crucial role in improving model performance by evaluating performance degradation when removing C³ Risk, the instrumental variable, and counterfactual representation modeling. The results indicate that the proposed loss function is effective in learning robust and causally complete representations.

**Essential References Not Discussed:**

Causal Mode Multiplexer: A Novel Framework for Unbiased Multispectral Pedestrian Detection, CVPR 2024

**Experimental Designs Or Analyses:**

The soundness and validity of the experimental designs were verified.

In Table 1, the proposed method was validated by comparing it with recent approaches across various MML benchmark datasets, including NYU Depth V2, SUN RGBD, FOOD 101, and MVSA, which cover different tasks. Additionally, the ablation study confirmed the effectiveness of each regularization term.

**Methods And Evaluation Criteria:**

The proposed C³R method and evaluation criteria are well-aligned with the goal of learning causally complete representations in Multi-Modal Learning (MML). The introduction of C³ Risk effectively quantifies causal sufficiency and necessity, while the Twin Network architecture provides a structured approach to estimating counterfactual effects. The use of instrumental variables further helps disentangle causal from spurious factors, which is essential for robust multimodal learning.

The evaluation strategy is comprehensive, covering six benchmark datasets spanning classification and segmentation tasks, including NYU Depth V2, SUN RGBD, FOOD 101, MVSA, and BraTS. The robustness analysis under Gaussian noise (image) and blank noise (text) strengthens the credibility of the results, ensuring that C³R improves both average and worst-case accuracy. The ablation study further confirms that each component of C³R contributes meaningfully to performance.

**Other Comments Or Suggestions:**

The paper claims that instrumental variables are used to mitigate spurious correlations, but there is a lack of quantitative experiments measuring how much spurious correlation has actually been reduced.

A more compelling validation would include quantitative and qualitative comparisons of representations before and after applying C³R on datasets with a high presence of spurious correlations.

**Other Strengths And Weaknesses:**

* Strengths:

The paper effectively identifies the limitations of existing causal representation learning research and introduces a novel approach that ensures both causal sufficiency and necessity, which is highly compelling.

Additionally, the use of Gradient-Based Counterfactual Modeling to disentangle causal factors and approximate counterfactual effects without generating new data is particularly interesting, as it allows for counterfactual reasoning without access to actual counterfactual samples.


* Weaknesses:

While the Twin Network generates counterfactual representations, there is a lack of direct validation on how well these estimated counterfactuals align with true counterfactual effects.

A comparison with manually curated counterfactual examples or other counterfactual modeling approaches would enhance the credibility of the proposed method.

**Questions For Authors:**

1. How can C³ Risk be directly verified as an indicator of causally complete representations? Specifically, what experimental approaches could be used to validate that a lower C³ Risk truly corresponds to representations that satisfy both causal sufficiency and necessity?

2. If you have conducted experiments using datasets with a high concentration of spurious correlations, could you provide details on the quantitative and qualitative comparisons of representations before and after applying C³R?

3. Are there any quantitative or qualitative methods to evaluate how similar the counterfactual representations generated through Gradient-Based Counterfactual Modeling using the Twin Network are to actual counterfactual representations?

**Relation To Broader Scientific Literature:**

The key contributions of this paper are closely related to existing research in multi-modal learning (MML).

By adopting the concepts of sufficiency and necessity from Pearl (2009), it introduces a novel approach—Causal Sufficiency and Causal Necessity. which significantly advances MML learning.

**Theoretical Claims:**

The proposed methods and evaluation criteria are well-aligned with the problem, as they have been quantitatively validated across various MML methods and diverse MML tasks.

---

> ### Author Rebuttal · Authors · 2025-03-31
>
> We sincerely appreciate Reviewer ZspQ's feedback and the time and effort dedicated to the review. We are also grateful for the recognition of our work and sincerely hope the following responses can eliminate the concerns.
> ## Response to W1 & Q3
> - The true counterfactual effect involves unobserved outcomes, making direct modeling difficult (Pearl, 2009). Existing work typically adopts the "minimal change" principle to estimate counterfactuals, which is proven to align with true counterfactual effects (Galles & Pearl, 1998; Kusner et al., 2017). For instance, Chapters III–V in Wachter et al. (2017) demonstrate that by making only minor adjustments to the treatment variable while preserving the distribution of other covariates, the counterfactual samples maintain the original data’s characteristics, ensuring accurate counterfactual effect estimates. Based on these results, we leverage gradient intervention to satisfy the "minimal change" principle for counterfactual effect estimation with theoretical guarantees (Appendix D.4). To assess whether the generated counterfactual data adhere to the principle for accurate estimation, we develop a distribution consistency test with Wasserstein distance $D_w$, i.e., whether the distribution of the covariates matches that of the original data. The lower $D_w$ shown below proves that our method satisfies the principle.
> - According to the suggestions, we conduct a toy experiment on LCKD and NYU Depth V2 to demonstrate credibility. We follow (Galles & Pearl, 1998) to make manually curated examples and select the recently proposed transport-based counterfactual modeling method (Lara et al., 2024) as another baseline. The table below shows the advantages of our method, i.e., superior accuracy and lowest computational cost.
> |Method|Acc|Calculation overhead|$D_w$|
> |-|-|-|-|
> |$C^3$R|77.6|$1\times$|0.9|
> |manually curated|77.4|$4.9\times$|0.7|
> |Transport-based|75.2|$2.3\times$|2.6|
> ## Response to Q1
> We have conducted experiments to validate "$C^3$ risk serves as an accurate indicator":
> - Appendix H.5: We calculate the heatmaps of samples using $C^3$ risk in Fig.7, where high-weight elements correspond to low $C^3$ risk. If low $C^3$ risk reflects causally sufficient and necessary factors, then reducing the weights of these elements would degrade model performance. Table 7 shows that reducing the weights of low $C^3$ risk elements indeed lowers performance, confirming that $C^3$ risk is an accurate indicator.
> - Section 6.2: We conduct experiments on MMLSynData, which includes four types of generated data, i.e., sufficient and necessary causes (SNC), sufficient but unnecessary causes (SC), necessary but insufficient causes (NC), and spurious correlations (SP). By evaluating the correlation between the representation learned by minimizing $C^3$ risk and SNC, we can validate whether low $C^3$ risk refers to causal sufficient and necessary causes. Figures 3 & 6 indicate that $C^3$R significantly increases the correlation between representations and SNC, proving the accuracy.
> - Section 6.2 & Appendix H: If $C^3$ risk can be the indicator, then $C^3$R should lead to performance improvements by minimizing $C^3$ risk. Tables 1-8 show that $C^3$R achieves significant performance gains across all baselines, proving effectiveness.
> ## Response to Q2 & Comments
> We have conducted experiments on datasets with "high spurious correlations" and provide the outline below:
> - Quantitative (Section 6.2, Appendices D.5 and H.3): We conduct experiments on MMLSynData, which contains SNC, SC, NC, and SP as mentioned in "Response to Q1 (2)". Different degrees of $D_{sp}$ are set to control the level of spurious correlations. By evaluating the correlation between the learned representation and SP/SNC, we can validate the effectiveness of the corresponding model in mitigating spurious correlations/learning causal complete representation. Figures 3 and 6 show that even with high spurious correlation ($D_{sp}=0.6$), $C^3$R markedly reduces the correlation with SP (0.4 -> 0.1) while significantly increasing the correlation with SNC (0.5 -> 0.9),  proving advantages.
> - Qualitative (Appendix H.5): As in "Response to Q1", the visualization shows that $C^3$R learns causally complete representations and effectively eliminates spurious correlations.
> ## Regarding Essential References Not Discussed
> We sincerely thank the suggestion and carefully reviewed the paper, finding differences with ours in problem, theory, method, experiments, etc. (cited and added in Section 7). Kim et al. aim to address modality bias in multispectral pedestrian detection tasks, e.g., "prediction on ROTX without thermal features". They conduct SCMs for analysis and propose CMM for the tasks. Differently, we focus on the concepts of causal sufficiency and necessity and are the first to explore them in MML. We propose a new theoretical framework and a plug-and-play method to learn causal complete representations, which can also be embedded in CMM.

---

### Decision · Program_Chairs · 2025-05-01

**Decision:**

Accept (poster)

**Comment:**

This paper addresses key limitations in Multi-Modal Learning (MML), where representations may be influenced by spurious correlations or lack causal relevance. It introduces the concept of Causal Complete Cause, C^3, requiring that learned representations be both causally sufficient and necessary. To operationalize this, the authors propose C^3 Risk, a metric quantifying the failure to meet causal completeness. They develop a Twin Network to estimate and minimize this risk. The resulting C³R regularization is a plug-and-play method that can be integrated into existing MML pipelines.

The reviewers pointed out the paper’s main weaknesses, which are mianly in clarity and technical precision. For example, the illustrative examples (e.g., Fig. 1) are ambiguous, with insufficiently clear distinctions between sufficient and necessary features, and a lack of explicit multi-modal elements. Some mechanisms of network components are not properly clarified.
Some key concepts, such as causal sufficiency, local invertibility, and non-exogeneity, are not clearly defined or aligned with standard causal inference terminology, which may cause confusion. Several important derivations lack sufficient explanation.

The authors have addressed the main concerns in the rebuttal, and all reviewers have converged toward positive recommendations, with two accepts and one weak accept. AC agrees and recommends accepting this paper based on the consensus from reviewers.

The authors are encouraged to revise the paper by clarifying the raised issues and incorporating relevant clarifications from the rebuttal.